# Deterministic PAC-Bayesian generalization bounds for deep networks via generalizing noise-resilience

**Vaishnavh Nagarajan**
Department of Computer Science
Carnegie Mellon University
Pittsburgh, PA
vaishnavh@cs.cmu.edu

**J. Zico Kolter**
Department of Computer Science
Carnegie Mellon University &
Bosch Center for AI
Pittsburgh, PA
zkolter@cs.cmu.edu

## Abstract

The ability of overparameterized deep networks to generalize well has been linked to the fact that stochastic gradient descent (SGD) finds solutions that lie in flat, wide minima in the training loss – minima where the output of the network is resilient to small random noise added to its parameters. So far this observation has been used to provide generalization guarantees only for neural networks whose parameters are either *stochastic* or *compressed*. In this work, we present a general PAC-Bayesian framework that leverages this observation to provide a bound on the original network learned – a network that is deterministic and uncompressed. What enables us to do this is a key novelty in our approach: our framework allows us to show that if on training data, the interactions between the weight matrices satisfy certain conditions that imply a wide training loss minimum, these conditions themselves *generalize* to the interactions between the matrices on test data, thereby implying a wide test loss minimum. We then apply our general framework in a setup where we assume that the pre-activation values of the network are not too small (although we assume this only on the training data). In this setup, we provide a generalization guarantee for the original (deterministic, uncompressed) network, that does not scale with product of the spectral norms of the weight matrices – a guarantee that would not have been possible with prior approaches.

## 1 Introduction

Modern deep neural networks contain millions of parameters and are trained on relatively few samples. Conventional wisdom in machine learning suggests that such models should massively overfit on the training data, as these models have the capacity to memorize even a randomly labeled dataset of similar size (Zhang et al., 2017; Neyshabur et al., 2015). Yet these models have achieved state-of-the-art generalization error on many real-world tasks. This observation has spurred an active line of research (Soudry et al., 2018; Brutzkus et al., 2018; Li & Liang, 2018) that has tried to understand what properties are possessed by stochastic gradient descent (SGD) training of deep networks that allows these networks to generalize well.

One particularly promising line of work in this area (Neyshabur et al., 2017; Arora et al., 2018) has been bounds that utilize the *noise-resilience* of deep networks on training data i.e., how much the training loss of the network changes with noise injected into the parameters, or roughly, how wide is the training loss minimum. While these have yielded generalization bounds that do not have a severe exponential dependence on depth (unlike other bounds that grow with the product of spectral norms of the weight matrices), these bounds are quite limited: they either apply to a *stochastic* version of the classifier (where the parameters are drawn from a distribution) or a *compressed* version of the classifier (where the parameters are modified and represented using fewer bits).

In this paper, we revisit the PAC-Bayesian analysis of deep networks in Neyshabur et al. (2017; 2018) and provide a general framework that allows one to use noise-resilience of the deep network

on training data to provide a bound on the original deterministic and uncompressed network. We achieve this by arguing that if on the training data, the interaction between the 'activated weight matrices' (weight matrices where the weights incoming from/outgoing to inactive units are zeroed out) satisfy certain conditions which results in a wide training loss minimum, these conditions themselves generalize to the weight matrix interactions on the test data.

After presenting this general PAC-Bayesian framework, we specialize it to the case of deep ReLU networks, showing that we can provide a generalization bound that accomplishes two goals simultaneously: i) it applies to the original network and ii) it does not scale exponentially with depth in terms of the products of the spectral norms of the weight matrices; instead our bound scales with more meaningful terms that capture the interactions between the weight matrices and do not have such a severe dependence on depth in practice. We note that all but one of these terms are indeed quite small on networks in practice. However, one particularly (empirically) large term that we use is the reciprocal of the magnitude of the network pre-activations on the training data (and so our bound would be small only in the scenario where the pre-activations are not too small). We emphasize that this drawback is more of a limitation in how we characterize noise-resilience through the specific conditions we chose for the ReLU network, rather than a drawback in our PAC-Bayesian framework itself. Our hope is that, since our technique is quite general and flexible, by carefully identifying the right set of conditions, in the future, one might be able to derive a similar generalization guarantee that is smaller in practice.

To the best of our knowledge, our approach of *generalizing* noise-resilience of deep networks from training data to test data in order to derive a bound on the *original* network that does not scale with products of spectral norms, has neither been considered nor accomplished so far, even in limited situations.

## 2 BACKGROUND AND RELATED WORK

One of the most important aspects of the generalization puzzle that has been studied is that of the flatness/width of the training loss at the minimum found by SGD. The general understanding is that flatter minima are correlated with better generalization behavior, and this should somehow help explain the generalization behavior (Hochreiter & Schmidhuber, 1997; Hinton & van Camp, 1993; Keskar et al., 2017). Flatness of the training loss minimum is also correlated with the observation that on training data, adding noise to the parameters of the network results only in little change in the output of the network – or in other words, the network is noise-resilient. Deep networks are known to be similarly resilient to noise injected into the inputs (Novak et al., 2018); but note that our theoretical analysis relies on resilience to parameter perturbations.

While some progress has been made in understanding the convergence and generalization behavior of SGD training of simple models like two-layered hidden neural networks under simple data distributions (Neyshabur et al., 2015; Soudry et al., 2018; Brutzkus et al., 2018; Li & Liang, 2018), all known generalization guarantees for SGD on deeper networks – through analyses that do not use noise-resilience properties of the networks – have strong exponential dependence on depth. In particular, these bounds scale either with the product of the spectral norms of the weight matrices (Neyshabur et al., 2018; Bartlett et al., 2017) or their Frobenius norms (Golowich et al., 2018). In practice, the weight matrices have a spectral norm that is as large as 2 or 3, and an even larger Frobenius norm that scales with $\sqrt{H}$ where $H$ is the width of the network i.e., maximum number of hidden units per layer. [1] Thus, the generalization bound scales as say, $2^D$ or $H^{D/2}$, where $D$ is the depth of the network.

At a high level, the reason these bounds suffer from such an exponential dependence on depth is that they effectively perform a worst case approximation of how the weight matrices interact with each other. For example, the product of the spectral norms arises from a naive approximation of the Lipschitz constant of the neural network, which would hold only when the singular values of the

---

[1]To understand why these values are of this order in magnitude, consider the initial matrix that is randomly initialized with independent entries with variance $1/\sqrt{H}$. It can be shown that the spectral norm of this matrix, with high probability, lies near its expected value, near 2 and the Frobenius norm near its expected value which is $\sqrt{H}$. Since SGD is observed not to move too far away from the initialization regardless of $H$ (Nagarajan & Kolter, 2017), these values are more or less preserved for the final weight matrices.

weight matrices all align with each other. However, in practice, for most inputs to the network, the interactions between the activated weight matrices are not as adverse.

By using noise-resilience of the networks, prior approaches (Arora et al., 2018; Neyshabur et al., 2017) have been able to derive bounds that replace the above worst-case approximation with smaller terms that realistically capture these interactions. However, these works are limited in critical ways. Arora et al. (2018) use noise-resilience of the network to *modify* and "compress" the parameter representation of the network, and derive a generalization bound on the compressed network. While this bound enjoys a better dependence on depth because its applies to a compressed network, the main drawback of this bound is that it does not apply on the original network. On the other hand, Neyshabur et al. (2017) take advantage of noise-resilience on training data by incorporating it within a PAC-Bayesian generalization bound (McAllester, 1999a). However, their final guarantee is only a bound on the expected test loss of a stochastic network.

In this work, we revisit the idea in Neyshabur et al. (2017), by pursuing the PAC-Bayesian framework (McAllester, 1999a) to answer this question. The standard PAC-Bayesian framework provides generalization bounds for the expected loss of a stochastic classifier, where the stochasticity typically corresponds to Gaussian noise injected into the parameters output by the learning algorithm. However, if the classifier is noise-resilient on both training and test data, one could extend the PAC-Bayesian bound to a standard generalization guarantee on the deterministic classifier.

Other works have used PAC-Bayesian bounds in different ways in the context of neural networks. Langford & Caruana (2001); Dziugaite & Roy (2017) optimize the stochasticity and/or the weights of the network in order to *numerically* compute good (i.e., non-vacuous) generalization bounds on the stochastic network. Neyshabur et al. (2018) derive generalization bounds on the original, deterministic network by working from the PAC-Bayesian bound on the stochastic network. However, as stated earlier, their work does not make use of noise resilience in the networks learned by SGD.

OUR CONTRIBUTIONS    The key contribution in our work is a general PAC-Bayesian framework for deriving generalization bounds while leveraging the noise resilience of a deep network. While our approach is applied to deep networks, we note that it is general enough to be applied to other classifiers.

In our framework, we consider a set of conditions that when satisfied by the network, makes the output of the network noise-resilient at a particular input datapoint. For example, these conditions could characterize the interactions between the activated weight matrices at a particular input. To provide a generalization guarantee, we assume that the learning algorithm has found weights such that these conditions hold for the weight interactions in the network *on training data* (which effectively implies a wide training loss minimum). Then, as a key step, we *generalize* these conditions over to the weight interactions on test data (which effectively implies a wide test loss minimum) [2]. Thus, with the guarantee that the classifier is noise-resilient both on training and test data, we derive a generalization bound on the test loss of the original network.

Finally, we apply our framework to a specific set up of ReLU based feedforward networks. In particular, we first instantiate the above abstract framework with a set of specific conditions, and then use the above framework to derive a bound on the original network. While very similar conditions have already been identified in prior work (Arora et al., 2018; Neyshabur et al., 2017) (see Appendix G for an extensive discussion of this), our contribution here is in showing how these conditions generalize from training to test data. Crucially, like these works, our bound does not have severe exponential dependence on depth in terms of products of spectral norms.

We note that in reality, all but one of our conditions on the network do hold on training data as necessitated by the framework. The strong, non-realistic condition we make is that the pre-activation values of the network are sufficiently large, although only on training data; however, in practice a small proportion of the pre-activation values can be arbitrarily small. Our generalization bound scales inversely with the smallest absolute value of the pre-activations on the training data, and hence in practice, our bound would be large.

---

[2]Note that we can not directly assume these conditions to hold on test data, as that would be 'cheating' from the perspective of a generalization guarantee.

Intuitively, we make this assumption to ensure that under sufficiently small parameter perturbations, the activation states of the units are guaranteed not to flip. It is worth noting that Arora et al. (2018); Neyshabur et al. (2017) too require similar, but more realistic assumptions about pre-activation values that effectively assume only a small proportion of units flip under noise. However, even under our stronger condition that no such units exist, it is not apparent how these approaches would yield a similar bound on the deterministic, uncompressed network without generalizing their conditions to test data. We hope that in the future our work could be developed further to accommodate the more realistic conditions from Arora et al. (2018); Neyshabur et al. (2017).

## 3   A GENERAL PAC-BAYESIAN FRAMEWORK

In this section, we present our general PAC-Bayesian framework that uses noise-resilience of the network to convert a PAC-Bayesian generalization bound on the stochastic classifier to a generalization bound on the deterministic classifier.

NOTATION.   Let $KL(\cdot\|\cdot)$ denote the KL-divergence. Let $\|\cdot\|, \|\cdot\|_\infty$ denote the $\ell_2$ norm and the $\ell_\infty$ norms of a vector, respectively. Let $\|\cdot\|_2, \|\cdot\|_F, \|\cdot\|_{2,\infty}$ denote the spectral norm, Frobenius norm and maximum row $\ell_2$ norm of a matrix, respectively. Consider a $K$-class learning task where the labeled datapoints $(\mathbf{x}, y)$ are drawn from an underlying distribution $\mathcal{D}$ over $\mathcal{X} \times \{1, 2, \cdots, K\}$ where $\mathcal{X} \in \mathbb{R}^N$. We consider a classifier parametrized by weights $\mathcal{W}$. For a given input $\mathbf{x}$ and class $k$, we denote the output of the classifier by $f(\mathbf{x}; \mathcal{W})[k]$. In our PAC-Bayesian analysis, we will use $\mathcal{U} \sim \mathcal{N}(0, \sigma^2)$ to denote parameters whose entries are sampled independently from a Gaussian, and $\mathcal{W} + \mathcal{U}$ to denote the entrywise addition of the two sets of parameters. We use $\|\mathcal{W}\|_F^2$ to denote $\sum_{d=1}^D \|W_d\|_F^2$. Given a training set $S$ of $m$ samples, we let $(\mathbf{x}, y) \sim S$ to denote uniform sampling from the set. Finally, for any $\gamma > 0$, let $\mathcal{L}_\gamma(f(\mathbf{x}; \mathcal{W}), y)$ denote a margin-based loss such that the loss is 0 only when $f(\mathbf{x}; \mathcal{W})[y] \geq \max_{j \neq y} f(\mathbf{x}; \mathcal{W})[j] + \gamma$, and 1 otherwise. Note that $\mathcal{L}_0$ corresponds to 0-1 error. See Appendix A for more notations.

TRADITIONAL PAC-BAYESIAN BOUNDS.   The PAC-Bayesian framework (McAllester, 1999a;b) allows us to derive generalization bounds for a *stochastic* classifier. Specifically, let $\tilde{\mathcal{W}}$ be a random variable in the parameter space whose distribution is learned based on training data $S$. Let $P$ be a prior distribution in the parameter space chosen independent of the training data. The PAC-Bayesian framework yields the following generalization bound on the 0-1 error of the stochastic classifier that holds with probability $1 - \delta$ over the draw of the training set $S$ of $m$ samples[3]:

$$\mathbb{E}_{\tilde{\mathcal{W}}}[\mathbb{E}_{(\mathbf{x},y)\sim\mathcal{D}}[\mathcal{L}_0(f(\mathbf{x}; \tilde{\mathcal{W}}), y)]] \leq \mathbb{E}_{\tilde{\mathcal{W}}}[\mathbb{E}_{(\mathbf{x},y)\sim S}[\mathcal{L}_0(f(\mathbf{x}; \tilde{\mathcal{W}}), y)]] + \tilde{\mathcal{O}}\left(\sqrt{KL(\tilde{\mathcal{W}}\|P)/m}\right)$$

Typically, and in the rest of this discussion, $\tilde{\mathcal{W}}$ is a Gaussian with covariance $\sigma^2 I$ for some $\sigma > 0$ centered at the weights $\mathcal{W}$ learned based on the training data. Furthermore, we will set $P$ to be a Gaussian with covariance $\sigma^2 I$ centered at the random initialization of the network like in Dziugaite & Roy (2017), instead of at the origin, like in Neyshabur et al. (2018). This is because the resulting KL-divergence – which depends on the distance between the means of the prior and the posterior – is known to be smaller, and to save a $\sqrt{H}$ factor in the bound (Nagarajan & Kolter, 2017).

### 3.1   OUR FRAMEWORK

To extend the above PAC-Bayesian bound to a standard generalization bound on a deterministic classifier $\mathcal{W}$, we need to replace the training and the test loss of the stochastic classifier with that of the original, deterministic classifier. However, in doing so, we will have to introduce extra terms in the upper bound to account for the perturbation suffered by the train and test loss under the Gaussian perturbation of the parameters. To tightly bound these two terms, we need that the network is noise-resilient on training and test data respectively. Our hope is that if the learning algorithm has found weights such that the network is noise-resilient on the training data, we can then generalize this noise-resilience over to test data as well, allowing us to better bound the excess terms.

---

[3]We use $\tilde{\mathcal{O}}(\cdot)$ to hide logarithmic factors.

We now discuss how noise-resilience is formalized in our framework through certain conditions on the weight matrices. Much of our discussion below is dedicated to how these conditions must be designed, as these details carry the key ideas behind how noise-resilience can be generalized from training to test data. We then present our main generalization bound and some intuition about our proof technique.

INPUT-DEPENDENT PROPERTIES OF WEIGHTS    Recall that, at a high level, the noise-resilience of a network corresponds to how little the network reacts to random parameter perturbations. Naturally, this would vary depending on the input. Hence, in our framework, we will analyze the noise-resilience of the network as a function of a given input. Specifically, we will characterize noise-resilience through conditions on how the weights of the model interact with each other for a given input. For example, in the next section, we will consider conditions of the form "the preactivation values of the hidden units in layer $d$, have magnitude larger than some small positive constant". The idea is that when these conditions involving the weights and the input are satisfied, if we add noise to the weights, the output of the classifier for that input will provably suffer only little perturbation. We will more generally refer to each scalar quantity involved in these conditions, such as each of the pre-activation values, as an *input-dependent property of the weights*.

We will now formulate these input-dependent properties and the conditions on them, for a generic classifier, and in the next section, we will see how they can be instantiated in the case of deep networks. Consider a classifier for which we can define $R$ different conditions, which when satisfied on a given input, will help us guarantee the classifier's noise-resilience at that input i.e., bound the output perturbation under random parameter perturbations (to get an idea of what $R$ corresponds to, in the case of deep networks, we will have a condition for each layer, and so $R$ will scale with depth). In particular, let the $r$th condition be a bound involving a particular set of input-dependent properties of the weights denoted by $\{\rho_{r,1}(\mathcal{W}, \mathbf{x}, y), \rho_{r,2}(\mathcal{W}, \mathbf{x}, y), \cdots, \}$ – here, each element $\rho_{r,l}(\mathcal{W}, \mathbf{x}, y)$ is a scalar value that depends on the weights and the input, just like pre-activation values[4]. Note that here the first subscript $l$ is the index of the element in the set, and the second subscript $r$ is the index of the set itself. Now for each of these properties, we will define a corresponding set of *positive* constants (that are independent of $\mathcal{W}, \mathbf{x}$ and $y$), denoted by $\{\Delta^{\star}_{r,1}, \Delta^{\star}_{r,2}, \cdots\}$, which we will use to specify our conditions. In particular, we say that the weights $\mathcal{W}$ satisfy the $r$th *condition* on the input $(\mathbf{x}, y)$ if[5]:

$$\forall l, \ \rho_{r,l}(\mathcal{W}, \mathbf{x}, y) > \Delta^{\star}_{r,l} \tag{1}$$

For convenience, we also define an additional $R+1$th set to be the singleton set containing the *margin* of the classifier on the input: $f(\mathbf{x}; \mathcal{W})[y] - \max_{j \neq y} f(\mathbf{x}; \mathcal{W})[j]$. Note that if this term is positive (negative) then the classification is (in)correct. We will also denote the constant $\Delta^{\star}_{R+1,1}$ as $\gamma_{\text{class}}$.

ORDERING OF THE SETS OF PROPERTIES    We now impose a crucial constraint on how these sets of properties depend on each other. Roughly speaking, we want that for a given input, *if the first $r-1$ sets of properties approximately satisfy the condition in Equation 1, then the properties in the $r$th set are noise-resilient* i.e., under random parameter perturbations, these properties do not suffer much perturbation. This kind of constraint would naturally hold for deep networks if we have chosen the properties carefully e.g., we will show that, for any given input, the perturbation in the pre-activation values of the $d$th layer is small as long as the absolute pre-activation values in the layers below $d-1$ are large, and a few other norm-bounds on the lower layer weights are satisfied.

We formalize the above requirement by defining expressions $\Delta_{r,l}(\sigma)$ that bound the perturbation in the properties $\rho_{r,l}$, in terms of the variance $\sigma^2$ of the parameter perturbations. For any $r \leq R+1$ and for any $(\mathbf{x}, y)$, our framework requires the following to hold:

---

[4]As we will see in the next section, most of these properties depend on only the unlabeled input $\mathbf{x}$ and not on $y$. But for the sake of convenience, we include $y$ in the formulation of the input-dependent property, and use the word input to refer to $\mathbf{x}$ or $(\mathbf{x}, y)$ depending on the context

[5]When we say $\forall l$ below, we refer to the set of all possible indices $l$ in the $r$th set, noting that different sets may have different cardinality.

if $\forall q < r, \forall l, \rho_{q,l}(\mathcal{W}, \mathbf{x}, y) > 0$ then

$$Pr_{\mathcal{U} \sim \mathcal{N}(0, \sigma^2 I)}\Big[\forall l \; |\rho_{r,l}(\mathcal{W} + \mathcal{U}, \mathbf{x}, y) - \rho_{r,l}(\mathcal{W}, \mathbf{x}, y)| > \frac{\Delta_{r,l}(\sigma)}{2} \quad \text{and}$$

$$\forall q < r, \forall l \; |\rho_{q,l}(\mathcal{W} + \mathcal{U}, \mathbf{x}, y) - \rho_{q,l}(\mathcal{W}, \mathbf{x}, y)| < \frac{\Delta_{q,l}(\sigma)}{2}\Big] \leq \frac{1}{(R+1)\sqrt{m}}. \tag{2}$$

Let us unpack the above constraint. First, although the above constraint must hold for all inputs $(\mathbf{x}, y)$, it effectively applies only to those inputs that satisfy the pre-condition of the if-then statement: namely, it applies only to inputs $(\mathbf{x}, y)$ that approximately satisfy the first $r - 1$ conditions in Equation 1 in that $\rho_{q,l}(\mathcal{W}, \mathbf{x}, y) > 0$ (instead of $\rho_{q,l}(\mathcal{W}, \mathbf{x}, y) > \Delta_{q,l}^\star$). Next, we discuss the second part of the above if-then statement which specifies a probability term that is required to be small for all such inputs. In words, the first event within the probability term above is the event that for a given random perturbation $\mathcal{U}$, the properties involved in the $r$th condition suffer a large perturbation. The second is the event that the properties involved in the first $r - 1$ conditions do *not* suffer much perturbation; but, given that these $r - 1$ conditions already hold approximately, this second event implies that these conditions are still preserved approximately under perturbation. In summary, our constraint requires the following: for any input on which the first $r - 1$ conditions hold, there should be very few parameter perturbations that significantly perturb the $r$th set of properties while preserving the first $r - 1$ conditions. When we instantiate the framework, we have to derive closed form expressions for the perturbation bounds $\Delta_{r,l}(\sigma)$ (in terms of only $\sigma$ and the constants $\Delta_{r,l}^\star$). As we will see, for ReLU networks, we will choose the properties in a way that this constraint naturally falls into place in a way that the perturbation bounds $\Delta_{r,l}(\sigma)$ do not grow with the product of spectral norms (Lemma E.1).

THEOREM STATEMENT    In this setup, we have the following 'margin-based' generalization guarantee on the original network. That is, we bound the 0-1 test error of the network by a margin-based error on the training data. Our generalization guarantee, which scales linearly with the number of conditions $R$, holds under the setting that the training algorithm always finds weights such that on the training data, the conditions in Equation 1 is satisfied for all $r = 1, \cdots, R$.

**Theorem 3.1.** *Let $\sigma^*$ be the maximum standard deviation of the Gaussian parameter perturbation such that the constraint in Equation 2 holds with $\Delta_{r,l}(\sigma^\star) \leq \Delta_{r,l}^\star \; \forall r \leq R + 1$ and $\forall l$. Then, for any $\delta > 0$, with probability $1 - \delta$ over the draw of samples $S$ from $\mathcal{D}^m$, for any $\mathcal{W}$ we have that, if $\mathcal{W}$ satisfies the conditions in Equation 1 for all $r \leq R$ and for all training examples $(\mathbf{x}, y) \in S$, then*

$$Pr_{(\mathbf{x},y) \sim \mathcal{D}}\left[\mathcal{L}_0(f(\mathbf{x}; \mathcal{W}), y)\right] \leq Pr_{(\mathbf{x},y) \sim S}\left[\mathcal{L}_{\gamma_{class}}(f(\mathbf{x}; \mathcal{W}), y)\right]$$

$$+ \tilde{\mathcal{O}}\left(R\sqrt{\frac{2KL(\mathcal{N}(\mathcal{W}, (\sigma^\star)^2 I)\|P) + \ln\frac{2mR}{\delta}}{m - 1}}\right)$$

The crux of our proof (in Appendix D) lies in generalizing the conditions of Equation 1 satisfied on the training data to test data *one after the other*, by proving that they are noise-resilient on both training and test data. Crucially, after we generalize the first $r - 1$ conditions from training data to test data (i.e., on most test and training data, the $r - 1$ conditions are satisfied), we will have from Equation 2 that the $r$th set of properties are noise-resilient on both training and test data. Using the noise-resilience of the $r$th set of properties on test/train data, we can generalize even the $r$th condition to test data.

We emphasize a key, fundamental tool that we present in Theorem C.1 to convert a generic PAC-Bayesian bound on a stochastic classifier, to a generalization bound on the deterministic classifier. Our technique is at a high level similar to approaches in London et al. (2016); McAllester (2003). In Section C.1, we argue how this technique is more powerful than other approaches in Neyshabur et al. (2018); Langford & Shawe-Taylor (2002); Herbrich & Graepel (2000) in leveraging the noise-resilience of a classifier. The high level argument is that, to convert the PAC-Bayesian bound, these latter works relied on a looser output perturbation bound, one that holds on all possible inputs, with high probability over all perturbations i.e., a bound on $\max_{\mathbf{x}} \|f(\mathbf{x}; \mathcal{W}) - f(\mathbf{x}; \mathcal{W} + \mathcal{U})\|_\infty$ w.h.p over draws of $\mathcal{U}$. In contrast, our technique relies on a subtly different but significantly tighter bound: a bound on the output perturbation that holds with high probability *given an input* i.e., a bound

on $\|f(\mathbf{x};\mathcal{W}) - f(\mathbf{x};\mathcal{W}+\mathcal{U})\|_\infty$ w.h.p over draws of $\mathcal{U}$ for each $\mathbf{x}$. When we do instantiate our framework as in the next section, this subtle difference is critical in being able to bound the output perturbation without suffering from a factor proportional to the product of the spectral norms of the weight matrices (which is the case in Neyshabur et al. (2018)).

## 4 APPLICATION OF OUR FRAMEWORK TO ReLU NETWORKS

NOTATION. In this section, we apply our framework to feedforward fully connected ReLU networks of depth $D$ (we care about $D > 2$) and width $H$ (which we will assume is larger than the input dimensionality $N$, to simplify our proofs) and derive a generalization bound on the original network that does not scale with the product of spectral norms of the weight matrices. Let $\phi(\cdot)$ denote the ReLU activation. We consider a network parameterized by $\mathcal{W} = (W_1, W_2, \cdots, W_D)$ such that the output of the network is computed as $f(\mathbf{x};\mathcal{W}) = W_D\phi(W_{D-1}\cdots\phi(W_1\mathbf{x}))$. We denote the value of the $h$th hidden unit on the $d$th layer before and after the activation by $g^d(\mathbf{x};\mathcal{W})[h]$ and $f^d(\mathbf{x};\mathcal{W})[h]$ respectively. We define $J^{d/d'}(\mathbf{x};\mathcal{W}) := \partial g^d(\mathbf{x};\mathcal{W})/\partial g^{d'}(\mathbf{x};\mathcal{W})$ to be the Jacobian of the pre-activations of layer $d$ with respect to the pre-activations of layer $d'$ for $d' \le d$ (each row in this Jacobian corresponds to a unit in layer $d$). In short, we will call this, Jacobian $d/d'$. Let $\mathcal{Z}$ denote the random initialization of the network.

Informally, we consider a setting where the learning algorithm satisfies the following conditions on the training data that make it noise-resilient on training data: a) the $\ell_2$ norm of the hidden layers are all small, b) the pre-activation values are all sufficiently large in magnitude, c) the Jacobian of any layer with respect to a lower layer, has rows with a small $\ell_2$ norm, and has a small spectral norm. We cast these conditions in the form of Equation 1 by appropriately defining the properties $\rho$'s and the margins $\Delta^\star$'s in the general framework. We note that these properties are quite similar to those already explored in Arora et al. (2018); Neyshabur et al. (2017); we provide more intuition about these properties, and how we cast them in our framework in Appendix E.1.

Having defined these properties, we first prove in Lemma E.1 in Appendix E a guarantee equivalent to the abstract inequality in Equation 2. Essentially, we show that under random perturbations of the parameters, the perturbation in the output of the network and the perturbation in the input-dependent properties involved in (a), (b), (c) themselves can all be bounded in terms of each other. Crucially, these perturbation bounds do not grow with the spectral norms of the network.

Having instantiated the framework as above, we then instantiate the bound provided by the framework. Our generalization bound scales with the bounds on the properties in (a) and (c) above as satisfied on the training data, and with the reciprocal of the property in (b) i.e., the smallest absolute value of the pre-activations on the training data. Additionally, our bound has an explicit dependence on the depth of the network, which arises from the fact that we generalize $R = \mathcal{O}(D)$ conditions. Most importantly, our bound does not have a dependence on the product of the spectral norms of the weight matrices.

**Theorem 4.1.** *(shorter version; see Appendix F for the complete statement) For any margin $\gamma_{class} > 0$, and any $\delta > 0$, with probability $1 - \delta$ over the draw of samples from $\mathcal{D}^m$, for any $\mathcal{W}$, we have that:*

$$Pr_{(\mathbf{x},y)\sim\mathcal{D}}\left[\mathcal{L}_0(f(\mathbf{x};\mathcal{W}),y)\right] \le Pr_{(\mathbf{x},y)\sim S}\left[\mathcal{L}_{\gamma_{class}}(f(\mathbf{x};\mathcal{W}),y)\right] + \tilde{\mathcal{O}}\left(D\sqrt{\|\mathcal{W} - \mathcal{Z}\|_F^2/((\sigma^\star)^2 m)}\right)$$

*Here $1/\sigma^\star$ equals $\tilde{\mathcal{O}}(\sqrt{H}\max\{\mathcal{B}_{layer\text{-}\ell_2}, \mathcal{B}_{preact}, \mathcal{B}_{output}, \mathcal{B}_{jac\text{-}row\text{-}\ell_2}, \mathcal{B}_{jac\text{-}spec}\})$, where*

$$\mathcal{B}_{layer\text{-}\ell_2} := \mathcal{O}\left(\max_{1\le d<D}\frac{\sum_{d'=1}^d\zeta_{d/d'}^\star\alpha_{d'-1}^\star}{\alpha_d^\star}\right), \mathcal{B}_{preact} := \mathcal{O}\left(\max_{1\le d<D}\frac{\sum_{d'=1}^d\zeta_{d/d'}^\star\alpha_{d'-1}^\star}{\sqrt{H}\gamma_d^\star}\right)$$

$$\mathcal{B}_{output} := \mathcal{O}\left(\frac{\sum_{d=1}^D\zeta_{D/d}^\star\alpha_{d-1}^\star}{\sqrt{H}\gamma_{class}}\right), \mathcal{B}_{jac\text{-}row\text{-}\ell_2} := \mathcal{O}\left(\max_{1\le d'<d<D}\frac{\zeta_{d-1/d'}^\star + \|W_d\|_{2,\infty}\sum_{d''=d'+1}^{d-1}\psi_{d-1/d''}^\star\zeta_{d''-1/d'}^\star}{\zeta_{d/d'}^\star}\right)$$

$$\mathcal{B}_{jac\text{-}spec} := \mathcal{O}\left(\max_{1\le d'<d<D}\frac{\psi_{d''-1/d'}^\star + \|W_d\|_2\sum_{d''=d'+1}^{d-1}\psi_{d-1/d''}^\star\zeta_{d''-1/d'}^\star}{\psi_{d/d'}^\star}\right)$$

*where, the terms $\alpha_d^\star, \gamma_d^\star$ etc., are norm-bounds that hold on all training data $(\mathbf{x}, y) \in S$ as follows: $\alpha_d^\star \geq \left\| f^d(\mathbf{x}; \mathcal{W}) \right\|$ (an upper bound on the $\ell_2$ norm of each hidden layer output), $\gamma_d^\star \leq \min_h \left| f^d(\mathbf{x}; \mathcal{W})[h] \right|$ (a lower bound on the absolute values of the pre-activations for each layer), $\zeta_{d/d'}^\star \geq \left\| J^{d/d'}(\mathbf{x}; \mathcal{W}) \right\|_{2,\infty}$ (an upper bound on the row $\ell_2$ norms of the Jacobian for each layer), $\psi_{d/d'}^\star \geq \left\| J^{d/d'}(\mathbf{x}; \mathcal{W}) \right\|_2$ (an upper bound on the spectral norm of the Jacobian for each layer).*

In Figure 1, we show how the terms in the bound vary for networks of varying depth with a small width of $H = 40$ on the MNIST dataset. We observe that $\mathcal{B}_{\text{layer-}\ell_2}, \mathcal{B}_{\text{output}}, \mathcal{B}_{\text{jac-row-}\ell_2}, \mathcal{B}_{\text{jac-spec}}$ typically lie in the range of $[10^0, 10^2]$ and scale with depth as $\propto 1.57^D$. In contrast, the equivalent term from Neyshabur et al. (2018) consisting of the product of spectral norms can be as large as $10^3$ or $10^5$ and scale with $D$ more severely as $2.15^D$.

The bottleneck in our bound is $\mathcal{B}_{\text{preact}}$, which scales inversely with the magnitude of the smallest absolute pre-activation value of the network. In practice, this term can be arbitrarily large, even though it does not depend on the product of spectral norms/depth. This is because some hidden units can have arbitrarily small absolute pre-activation values – although this is true only for a small proportion of these units.

To give an idea of the typical, non-pathological magnitude of the pre-activation values, we plot two other variations of $\mathcal{B}_{\text{preact}}$: a) $5\%$-$\mathcal{B}_{\text{preact}}$ which is calculated by ignoring $5\%$ of the training datapoints with the smallest absolute pre-activation values and b) median-$\mathcal{B}_{\text{preact}}$ which is calculated by ignoring half the hidden units in each layer with the smallest absolute pre-activation values for each input. We observe that median-$\mathcal{B}_{\text{preact}}$ is quite small (of the order of $10^2$), while $5\%$-$\mathcal{B}_{\text{preact}}$, while large (of the order of $10^4$), is still orders of magnitude smaller than $\mathcal{B}_{\text{preact}}$.

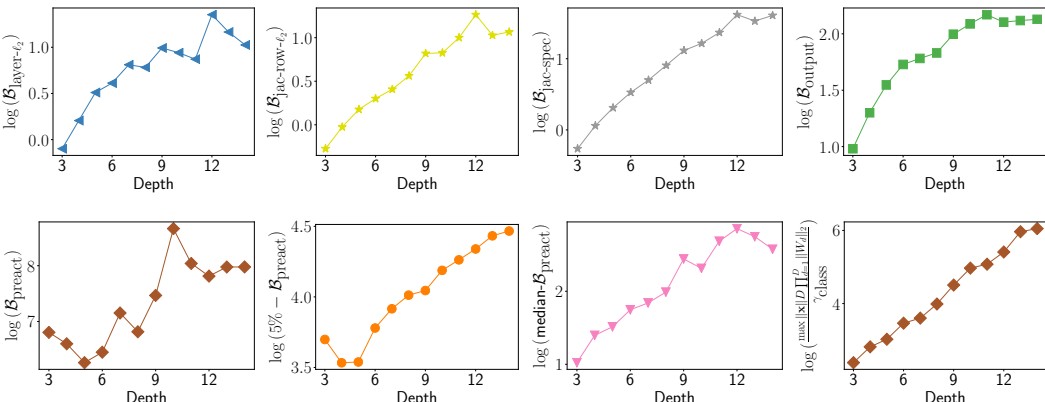

Figure 1: In the above figure, we plot the logarithm (to the base 10) of values of the terms occurring in the upper bound of $1/\sigma^\star$ for networks of varying depth, with $H = 40$. Additionally, we plot variations of $\mathcal{B}_{\text{preact}}$, namely $5\%$-$\mathcal{B}_{\text{preact}}$ and median-$\mathcal{B}_{\text{preact}}$ as discussed in the text. We also plot the equivalent term from Neyshabur et al. (2018) corresponding to $\max_{\mathbf{x}} \|\mathbf{x}\| D \prod_{d=1}^D \|W_d\|_2 / \gamma_{\text{class}}$. Note that if the slope of the $\log y$ vs $D$ graph is $c$, then the $y \propto (10^c)^D$.

In Figure 2 we show how our overall bound and existing product-of-spectral-norm-based bounds (Bartlett et al., 2017; Neyshabur et al., 2018) vary with depth. While our bound is orders of magnitude larger than prior bounds, the key point here is that our bound grows with depth as $1.57^D$ while prior bounds grow with depth as $2.15^D$ indicating that our bound should perform asymptotically better with respect to depth. Indeed, we verify that our bound obtains better values than the other existing bounds when $D = 28$ (see Figure 2 b). We also plot hypothetical variations of our bound replacing $\mathcal{B}_{\text{preact}}$ with $5\%$-$\mathcal{B}_{\text{preact}}$ (see "Ours-5%") and median-$\mathcal{B}_{\text{preact}}$ (see "Ours-Median") both of which perform orders of magnitude better than our actual bound (note that these two hypothetical bounds do not actually hold good). In fact for larger depth, the bound with $5\%$-$\mathcal{B}_{\text{preact}}$ performs better than all other bounds (including existing bounds). This indicates that the only bottleneck in our bound comes from the dependence on the smallest pre-activation magnitudes, and if this particular

dependence is addressed, our bound has the potential to achieve tighter guarantees for even smaller $D$ such as $D = 8$.

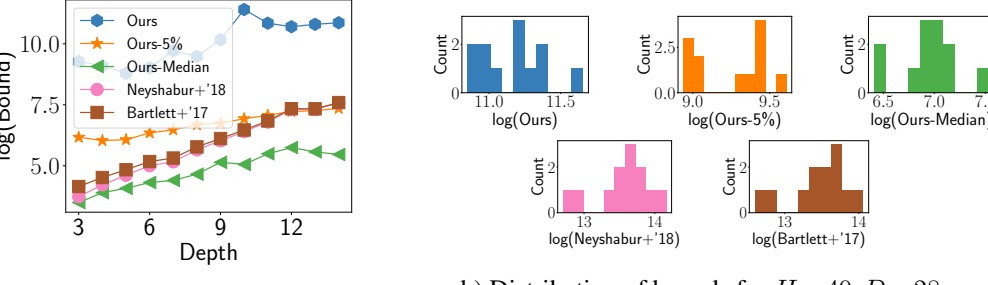

a) Bound vs. Depth

b) Distribution of bounds for $H = 40, D = 28$

Figure 2: In the left, we vary the depth of the network (fixing $H = 40$) and plot the logarithm of various generalization bounds ignoring the dependence on the training dataset size and a $\log(DH)$ factor in all of the considered bounds. Specifically, we consider our bound, the hypothetical versions of our bound involving $5\%$-$\mathcal{B}_{\text{preact}}$ and median-$\mathcal{B}_{\text{preact}}$ respectively, and the bounds from Neyshabur et al. (2018) $\frac{\max_{\mathbf{x}} \|\mathbf{x}\|_2 D\sqrt{H} \prod_{d=1}^{D} \|W_d\|_2}{\gamma_{\text{class}}} \cdot \sqrt{\sum_{d=1}^{D} \frac{\|W_d - Z_d\|_F^2}{\|W_d\|_2^2}}$ and Bartlett et al. (2017) $\frac{\max_{\mathbf{x}} \|\mathbf{x}\|_2 \prod_{d=1}^{D} \|W_d\|_2}{\gamma_{\text{class}}} \cdot$ $\left( \sum_{d=1}^{D} \left( \frac{\|W_d - Z_d\|_{2,1}}{\|W_d\|} \right)^{2/3} \right)^{3/2}$ both of which have been modified to include distance from initialization instead of distance from origin for a fair comparison. Observe the last two bounds have a plot with a larger slope than the other bounds indicating that they might potentially do worse for a sufficiently large $D$. Indeed, this can be observed from the plots on the right where we report the distribution of the logarithm of these bounds for $D = 28$ across 12 runs (although under training settings different from the experiments on the left; see Appendix F.3 for the exact details).

We refer the reader to Appendix F.3 for added discussion where we demonstrate how all the quantities in our bound vary with depth for $H = 1280$ (Figure 3, 4) and with width for $D = 8, 14$ (Figures 5 and 6).

Finally, as noted before, we emphasize that the dependence of our bound on the pre-activation values is a limitation in how we characterize noise-resilience through our conditions rather than a drawback in our general PAC-Bayesian framework itself. Specifically, using the assumed lower bound on the pre-activation magnitudes we can ensure that, under noise, the activation states of the units do not flip; then the noise propagates through the network in a tractable, "linear" manner. Improving this analysis is an important direction for future work. For example, one could modify our analysis to allow perturbations large enough to flip a small proportion of the activation states; one could potentially formulate such realistic conditions by drawing inspiration from the conditions in Neyshabur et al. (2017); Arora et al. (2018).

However, we note that even though these prior approaches made more realistic assumptions about the magnitudes of the pre-activation values, the key limitation in these approaches is that even under our non-realistic assumption, their approaches would yield bounds only on stochastic/compressed networks. *Generalizing noise-resilience from training data to test data* is crucial to extending these bounds to the original network, which we accomplish.

## 5   SUMMARY AND FUTURE WORK

In this work, we introduced a novel PAC-Bayesian framework for leveraging the noise-resilience of deep neural networks on training data, to derive a generalization bound on the original uncompressed, deterministic network. The main philosophy of our approach is to first generalize the noise-resilience from training data to test data using which we convert a PAC-Bayesian bound on a stochastic network to a standard margin-based generalization bound. We apply our approach to ReLU based networks and derive a bound that scales with terms that capture the interactions between the weight matrices better than the product of spectral norms.

For future work, the most important direction is that of removing the dependence on our strong assumption that the magnitude of the pre-activation values of the network are not too small on training data. More generally, a better understanding of the source of noise-resilience in deep ReLU networks would help in applying our framework more carefully in these settings, leading to tighter guarantees on the original network.

ACKNOWLEDGEMENTS.    Vaishnavh Nagarajan was partially supported by a grant from the Bosch Center for AI.

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

## Appendix

## A  Notations (continued)

We will use upper-case symbols to denote matrices, and lower-case bold-face symbols to denote vectors. In order to make the mathematical statements/derivations easier to read, if we want to emphasize a term, say $x$, we write, $\textcolor{red}{\boldsymbol{x}}$.

Recall that we consider a neural netork of depth $D$ (i.e., $D - 1$ hidden layers and one output layer) mapping from $\mathbb{R}^N \to \mathbb{R}^K$, where $K$ is the number of class labels in the learning task. The layers are fully connected with $H$ units in each hidden layer, and with ReLU activations $\phi(\cdot)$ on all the hidden units and linear activations on the output units. We denote the parameters of the network using the symbol $\mathcal{W}$, which in turn denotes a set of weight matrices $W_1, W_2, \cdots, W_D$. Here, $W_1 \in \mathbb{R}^{H \times N}$, and $W_D \in \mathbb{R}^{K \times H}$ and for all other layers $d \neq 1, D$, $W_d \in \mathbb{R}^{H \times H}$. We will use the notation $\mathcal{W}_d$ to denote the first $d$ weight matrices. We denote the vector of weights input to the $h$th unit on the $d$th layer (which corresponds to the $h$th row in $W_d$) as $\mathbf{w}_h^d$.

For any input $\mathbf{x} \in \mathbb{R}^N$, we denote the function computed by the network on that input as $f(\mathbf{x}; \mathcal{W}) = W_D \phi(W_{D-1} \cdots \phi(W_1 \mathbf{x}))$. For any $d = 1, \cdots, D - 1$, we denote the output of the $d$th hidden layer after the activation by $f^d(\mathbf{x}; \mathcal{W})$. We denote the corresponding pre-activation values for that layer by $g^d(\mathbf{x}; \mathcal{W})$. We denote the value of the $h$th hidden unit on the $d$th layer after and before the activation by $f^d(\mathbf{x}; \mathcal{W})[h]$ and $g^d(\mathbf{x}; \mathcal{W})[h]$ respectively. Note that for the output layer $d = D$, these two values are equal as we assume only a linear activation. For $d = 0$, we define $f^0(\mathbf{x}; \mathcal{W}) = \mathbf{x}$. As a result, we have the following recursions:

$$f^d(\mathbf{x}; \mathcal{W}) = \phi\left(g^d(\mathbf{x}; \mathcal{W})\right), d = 1, 2, \cdots, D - 1$$
$$f^D(\mathbf{x}; \mathcal{W}) = g^D(\mathbf{x}; \mathcal{W}) = f(\mathbf{x}; \mathcal{W}),$$
$$g^d(\mathbf{x}; \mathcal{W}) = W_d f^{d-1}(\mathbf{x}; \mathcal{W}), \forall d = 1, 2, \cdots, D$$
$$g^d(\mathbf{x}; \mathcal{W})[h] = \mathbf{w}_h^d \cdot f^{d-1}(\mathbf{x}; \mathcal{W})[h], \forall d = 1, 2, \cdots, D$$

For layers $d', d$ such that $d' \leq d$, let us define $J^{d/d'}(\mathbf{x}; \mathcal{W})$ to be the Jacobian corresponding to the pre-activation values of layer $d$ with respect to the pre-activation values of layer $d'$ on an input $\mathbf{x}$. That is,

$$J^{d/d'}(\mathbf{x}; \mathcal{W}) = \frac{\partial g^d(\mathbf{x}; \mathcal{W})}{\partial g^{d'}(\mathbf{x}; \mathcal{W})}$$

In other words, this corresponds to the product of the 'activated' portion of the matrices $W_{d'+1}, W_{d'+2}, \cdots, W_d$, where the weights corresponding to inactive inputs are zeroed out. In short, we will call this 'Jacobian $d/d'$'. Note that each row in this Jacobian corresponds to a unit on the $d$th layer, and each column corresponds to a unit on the $d'$th layer.

We will denote the parameters of a random initialization of the network by $\mathcal{Z} = (Z_1, Z_2, \cdots, Z_d)$. Let $\mathcal{D}$ be an underlying distribution over $\mathbb{R}^N \times \{1, 2, \cdots, K\}$ from which the data is drawn.

In our PAC-Bayesian analysis, we will use $\mathcal{U}$ to denote a set of $D$ weight matrices $U_1, U_2, \cdots, U_D$ whose entries are sampled independently from a Gaussian. Furthermore, we will use $\mathcal{U}_d$ to denote only the first $d$ of the randomly sampled weight matrices, and $\mathcal{W} + \mathcal{U}_d$ to denote a network where the $d$ random matrices are added to the first $d$ weight matrices in $\mathcal{W}$. Note that $\mathcal{W} + \mathcal{U}_0 = \mathcal{W}$. Thus, $f(\mathbf{x}; \mathcal{W} + \mathcal{U}_d)$ is the output of a network where the first $d$ weight matrices have been perturbed. In our analysis, we will also need to study a perturbed network where the hidden units are frozen to be at the activation state they were at before the perturbation; we will use the notation $\mathcal{W}[+\mathcal{U}_d]$ to denote the weights of such a network.

For our statements regarding probability of events, we will use $\wedge$, $\vee$, and $\neg$ to denote the intersection, union and complement of events (to disambiguate from the set operators).

# B    USEFUL LEMMAS

In this section, we present some standard results. The first two results below will be useful for our noise resilience analysis.

## HOEFFDING BOUND

**Lemma B.1.** *For $i = 1, 2, \cdots, n$, let $X_i$ be independent random variables sampled from a Gaussian with mean $\mu_i$ and variance $\sigma_i^2$. Then for all $t \geq 0$, we have:*

$$Pr\left[\sum_{i=1}^{n}(X_i - \mu_i) \geq t\right] \leq \exp\left(-\frac{t^2}{2\sum_{i=1}^{n}\sigma_i^2}\right).$$

*Or alternatively, for $\delta \in (0, 1]$*

$$Pr\left[\sum_{i=1}^{n}(X_i - \mu_i) \geq \sqrt{2\sum_{i=1}^{n}\sigma_i^2 \ln\frac{1}{\delta}}\right] \leq \delta$$

Note that an identical inequality holds good symmetrically for the event $\sum_{i=1}^{n} X_i - \mu_i \leq -t$, and so the probability that the event $|\sum_{i=1}^{n} X_i - \mu_i| > t$ holds, is at most twice the failure probability in the above inequalities.

## PRODUCT OF AN ENTRYWISE GAUSSIAN MATRIX AND A VECTOR

**Lemma B.2.** *Let $U$ be a $H_1 \times H_2$ matrix where each entry is sampled from $\mathcal{N}(0, \sigma^2)$. Let $\mathbf{x}$ be an arbitrary vector in $\mathbb{R}^{H_2}$. Then, $U\mathbf{x} \sim \mathcal{N}(0, \|\mathbf{x}\|_2^2 \sigma^2 I)$.*

*Proof.* $U\mathbf{x}$ is a random vector sampled from a multivariate Gaussian with mean $\mathbb{E}[U\mathbf{x}] = 0$ and co-variance $\mathbb{E}[U\mathbf{x}\mathbf{x}^T U^T]$. The $(i, j)$th entry in this covariance matrix is $\mathbb{E}[(\mathbf{u}_i^T \mathbf{x})(\mathbf{u}_j^T \mathbf{x})]$ where $\mathbf{u}_i$ and $\mathbf{u}_j$ are the $i$th and $j$th row in $U$. When $i = j$, $\mathbb{E}[(\mathbf{u}_i^T \mathbf{x})(\mathbf{u}_j^T \mathbf{x})] = \mathbb{E}[\|\mathbf{u}_i^T \mathbf{x}\|^2] = \sum_{h=1}^{H_2} \mathbb{E}[u_{ih}^2]x_h^2 = \sigma^2 \|\mathbf{x}\|_2^2$. When $i \neq j$, since $\mathbf{u}_i$ and $\mathbf{u}_j$ are independent random variables, we will have $\mathbb{E}[(\mathbf{u}_i^T \mathbf{x})(\mathbf{u}_j^T \mathbf{x})] = \sum_{h=1}^{H_2} \mathbb{E}[u_{ih}x_h] \sum_{h=1}^{H_2} \mathbb{E}[u_{jh}x_h] = 0$. $\qquad\square$

SPECTRAL NORM OF ENTRY-WISE GAUSSIAN MATRIX    The following result (Tropp, 2012) bounds the spectral norm of a matrix with Gaussian entries, with high probability:

**Lemma B.3.** *Let $U$ be a $H \times H$ matrix. Then,*

$$Pr_{U \sim \mathcal{N}(0, \sigma^2 I)}\left[\|U\|_2 > t\right] \leq 2H\exp(-t^2/2H\sigma^2)$$

*or alternatively, for any $\delta > 0$,*

$$Pr_{U \sim \mathcal{N}(0, \sigma^2 I)}\left[\|U\|_2 > \sigma\sqrt{2H\ln\frac{2H}{\delta}}\right] \leq 2H\exp(-t^2/2H\sigma^2)$$

KL DIVERGENCE OF GAUSSIANS.    We will use the following KL divergence equality to bound the generalization error in our PAC-Bayesian analyses.

**Lemma B.4.** *Let $P$ be the spherical Gaussian $\mathcal{N}(\boldsymbol{\mu}_1, \sigma^2 I)$ and $Q$ be the spherical Gaussian $\mathcal{N}(\boldsymbol{\mu}_2, \sigma^2 I)$. Then, the KL-divergence between $Q$ and $P$ is:*

$$KL(Q\|P) = \frac{\|\boldsymbol{\mu}_2 - \boldsymbol{\mu}_1\|^2}{2\sigma^2}$$

## C    PAC-BAYESIAN THEOREM

In this section, we will present our main PAC-Bayesian theorem that will guide our analysis of generalization in our framework. Concretely, our result extends the generalization bound provided by conventional PAC-Bayesian analysis (McAllester, 2003) – which is a generalization bound on the expected loss of a *distribution* of classifiers i.e., a stochastic classifier – to a generalization bound on a deterministic classifier. The way we reduce the PAC-Bayesian bound to a standard generalization bound, is different from the one pursued in previous works (Neyshabur et al., 2018; Langford & Shawe-Taylor, 2002).

The generalization bound that we state below is a bit more general than standard generalization bounds on deterministic networks. Typically, generalization bounds are on the classification error; however, as discussed in the main paper we will be dealing with generalizing multiple different conditions on the interactions between the weights of the network from the training data to test data. So to state a bound that is general enough, we consider a set of generic functions $\rho_r(\mathcal{W}, \mathbf{x}, \mathbf{y})$ for $r = 1, 2, \cdots R'$ (we use $R'$ to distinguish it from $R$, the number of conditions in the abstract classifier of Section 3.1). Each of these functions compute a scalar value that corresponds to some input-dependent property of the network with parameters $\mathcal{W}$ for the datapoint $(\mathbf{x}, y)$. As an example, this property could simply be the margin of the function on the $y$th class i.e., $f(\mathbf{x}; \mathcal{W})[y] - \max_{j \neq y} f(\mathbf{x}; \mathcal{W})[j]$.

**Theorem C.1.** *Let $P$ be a prior distribution over the parameter space that is chosen independent of the training dataset. Let $\mathcal{U}$ be a random variable sampled entrywise from $\mathcal{N}(0, \sigma^2)$. Let $\rho_r(\cdot, \cdot, \cdot)$ and $\Delta_r > 0$ for $r = 1, 2, \cdots R'$, be a set of input-dependent properties and their corresponding margins. We define the network $\mathcal{W}$ to be noise-resilient with respect to all these functions, at a given data point $(\mathbf{x}, y)$ if:*

$$Pr_{\mathcal{U} \sim \mathcal{N}(0, \sigma^2)} \left[ \exists r \ : \ |\rho_r(\mathcal{W}, \mathbf{x}, y) - \rho_r(\mathcal{W} + \mathcal{U}, \mathbf{x}, y)| > \frac{\Delta_r}{2} \right] \leq \frac{1}{\sqrt{m}}. \tag{3}$$

*Let $\mu_{\mathcal{D}}(\{(\rho_r, \Delta_r)\}_{r=1}^{R'}, \mathcal{W})$ denote the probability over the random draw of a point $(\mathbf{x}, y)$ drawn from $\mathcal{D}$, that the network with weights $\mathcal{W}$ is **not** noise-resilient at $(\mathbf{x}, y)$ according to Equation 3. That is, let $\mu_{\mathcal{D}}(\{(\rho_r, \Delta_r)\}_{r=1}^{R'}, \mathcal{W}) :=$*

$$Pr_{(\mathbf{x}, y) \sim \mathcal{D}} \left[ Pr_{\mathcal{U} \sim \mathcal{N}(0, \sigma^2)} \left[ \exists r \ : \ |\rho_r(\mathcal{W}, \mathbf{x}, y) - \rho_r(\mathcal{W} + \mathcal{U}, \mathbf{x}, y)| > \frac{\Delta_r}{2} \right] > \frac{1}{\sqrt{m}} \right]$$

*Similarly, let $\hat{\mu}_S(\{(\rho_r, \Delta_r)\}_{r=1}^{R'}, \mathcal{W})$ denote the fraction of data points $(\mathbf{x}, y)$ in a dataset $S$ for which the network is **not** noise-resilient according to Equation 3. Then for any $\delta$, with probability $1 - \delta$ over the draws of a sample set $S = \{(\mathbf{x}_i, y_i) \sim \mathcal{D} \,|\, i = 1, 2, \cdots, m\}$, for any $\mathcal{W}$ we have:*

$$Pr_{(\mathbf{x}, y) \sim \mathcal{D}} \left[ \exists r \ : \ \rho_r(\mathcal{W}, \mathbf{x}, y) < 0 \right] \leq \frac{1}{m} \sum_{(\mathbf{x}, y) \in S} \mathbf{1} \left[ \exists r \ : \ \rho_r(\mathcal{W}, \mathbf{x}, y) < \Delta_r \right] + \hat{\mu}_S(\{(\rho_r, \Delta_r)\}_{r=1}^{R'}, \mathcal{W})$$

$$+ \mu_{\mathcal{D}}(\{(\rho_r, \Delta_r)\}_{r=1}^{R'}, \mathcal{W}) + 2\sqrt{\frac{2KL(\mathcal{N}(\mathcal{W}, \sigma^2 I) \| P) + \ln \frac{2m}{\delta}}{m - 1}} + \frac{2}{\sqrt{m} - 1}.$$

The reader maybe curious about how one would bound the term $\mu_{\mathcal{D}}$ in the above bound, as this term corresponds to noise-resilience with respect to *test* data. This is precisely what we bound later when we generalize the noise-resilience-related conditions satisfied on train data over to test data.

### C.1    KEY ADVANTAGE OF OUR BOUND.

The above approach differs from previous approaches used by Neyshabur et al. (2018); Langford & Shawe-Taylor (2002) in how strong a noise-resilience we require of the classifier to provide the generalization guarantee. The stronger the noise-resilience requirement, the more price we have to pay when we jump from the PAC-Bayesian guarantee on the stochastic classifier to a guarantee on the deterministic classifier. We argue that *our noise-resilience requirement is a much milder condition and therefore promises tighter guarantees*. Our requirement is in fact philosophically similar to London et al. (2016); McAllester (2003), although technically different.

More concretely, to arrive at a reasonable generalization guarantee in our setup, we would need that $\mu_{\mathcal{D}}$ and $\hat{\mu}_S$ are both only as large as $\mathcal{O}(1/\sqrt{m})$. In other words, we would want the following for $(\mathbf{x}, y) \sim \mathcal{D}$ and for $(\mathbf{x}, y) \sim S$:

$$\Pr_{(\mathbf{x},y)}\left[\Pr_{\mathcal{U} \sim \mathcal{N}(0,\sigma^2)}\left[\exists r \; : \; |\rho_r(\mathcal{W}, \mathbf{x}, y) - \rho_r(\mathcal{W} + \mathcal{U}, \mathbf{x}, y)| > \frac{\Delta_r}{2}\right] > \frac{1}{\sqrt{m}}\right] = \mathcal{O}(1/\sqrt{m}).$$

Previous works require a noise resilience condition of the form that with high probability a particular perturbation does not perturb the classifier output on *any input*. For example, the noise-resilience condition used in Neyshabur et al. (2018) written in terms of our notations, would be:

$$\Pr_{\mathcal{U} \sim \mathcal{N}(0,\sigma^2)}\left[\textcolor{red}{\exists \mathbf{x}} \; : \; \exists r \; : \; |\rho_r(\mathcal{W}, \mathbf{x}, y) - \rho_r(\mathcal{W} + \mathcal{U}, \mathbf{x}, y)| > \frac{\Delta_r}{2}\right] \le \frac{1}{2}.$$

The main difference between the above two formulations is in what makes a particular perturbation (un)favorable for the classifier. In our case, we deem a perturbation unfavorable only after fixing the datapoint. However, in the earlier works, a perturbation is deemed unfavorable if it perturbs the classifier output sufficiently on *some* datapoint from the domain of the distribution. While this difference is subtle, the earlier approach would lead to a much more pessimistic analysis of these perturbations. In our analysis, this weakened noise resilience condition will be critical in analyzing the Gaussian perturbations more carefully than in Neyshabur et al. (2018) i.e., we can bound the perturbation in the classifier output more tightly by analyzing the Gaussian perturbation for a fixed input point.

Note that one way our noise resilience condition would seem stronger in that on a given datapoint we want less than $1/\sqrt{m}$ mass of the perturbations to be unfavorable for us, while in previous bounds, there can be as much as $1/2$ probability mass of perturbations that are unfavorable. In our analysis, this will only weaken our generalization bound by a $\ln \sqrt{m}$ factor in comparison to previous bounds (while we save other significant factors).

## C.2  PROOF FOR THEOREM C.1

*Proof.* The starting point of our proof is a standard PAC-Bayesian theorem McAllester (2003) which bounds the generalization error of a stochastic classifier. Let $P$ be a data-independent prior over the parameter space. Let $\mathcal{L}(\mathcal{W}, \mathbf{x}, y)$ be any loss function that takes as input the network parameter, and a datapoint $\mathbf{x}$ and its true label $y$ and outputs a value in $[0, 1]$. Then, we have that, with probability $1 - \delta$ over the draw of $S \sim \mathcal{D}^m$, for every distribution $Q$ over the parameter space, the following holds:

$$\mathbb{E}_{\tilde{\mathcal{W}} \sim Q}\left[\mathbb{E}_{(\mathbf{x},y) \sim \mathcal{D}}\left[\mathcal{L}(\tilde{\mathcal{W}}, \mathbf{x}, y)\right]\right] \le \mathbb{E}_{\tilde{\mathcal{W}} \sim Q}\left[\frac{1}{m}\sum_{(\mathbf{x},y) \in S}\mathcal{L}(\tilde{\mathcal{W}}, \mathbf{x}, y)\right] + 2\sqrt{\frac{2KL(Q\|P) + \ln\frac{2m}{\delta}}{m-1}} \quad (4)$$

In other words, the statement tells us that except for a $\delta$ proportion of bad draws of $m$ samples, the test loss of the stochastic classifier $\tilde{\mathcal{W}} \sim Q$ would be close to its train loss. This holds for every possible distribution $Q$, which allows us to cleverly choose $Q$ based on $S$. As is the convention, we choose $Q$ to be the distribution of the stochastic classifier picked from $\mathcal{N}(\mathcal{W}, \sigma^2 I)$ i.e., a Gaussian perturbation of the deterministic classifier $\mathcal{W}$.

RELATING TEST LOSS OF STOCHASTIC CLASSIFIER TO DETERMINISTIC CLASSIFIER.  Now our task is to bound the loss for the deterministic classifier $\mathcal{W}$, $\Pr_{(\mathbf{x},y) \sim \mathcal{D}}\left[\exists r \mid \rho_r(\mathcal{W}, \mathbf{x}, y) < 0\right]$. To this end, let us define the following margin-based variation of this loss for some $c \ge 0$:

$$\mathcal{L}_c(\mathcal{W}, \mathbf{x}, y) = \begin{cases} 1 & \exists r \; : \; \rho_r(\mathcal{W}, \mathbf{x}, y) < c\Delta_r \\ 0 & \text{otherwise,} \end{cases}$$

and so we have $\Pr_{(\mathbf{x},y) \sim \mathcal{D}}\left[\exists r \mid \rho_r(\mathcal{W}, \mathbf{x}, y) < 0\right] = \mathbb{E}_{(\mathbf{x},y) \sim \mathcal{D}}\left[\mathcal{L}_0(\mathcal{W}, \mathbf{x}, y)\right]$. First, we will bound the expected $\mathcal{L}_0$ of a deterministic classifier by the expected $\mathcal{L}_{1/2}$ of the stochastic classifier; then we will bound the test $\mathcal{L}_{1/2}$ of the stochastic classifier using the PAC-Bayesian bound.

We will split the expected loss of the deterministic classifier into an expectation over datapoints for which it is noise-resilient with respect to Gaussian noise and an expectation over the rest. To write this out, we define, for a datapoint $(\mathbf{x}, y)$, $\mathfrak{N}(\mathcal{W}, \mathbf{x}, y)$ to be the event that $\mathcal{W}$ is noise-resilient at $(\mathbf{x}, y)$ as defined in Equation 3 in the theorem statement:

$$
\begin{aligned}
\mathbb{E}_{(\mathbf{x},y)\sim\mathcal{D}}\left[\mathcal{L}_0(\mathcal{W},\mathbf{x},y)\right] = {} & \mathbb{E}_{(\mathbf{x},y)\sim\mathcal{D}}\left[\mathcal{L}_0(\mathcal{W},\mathbf{x},y)\big|\ \mathfrak{N}(\mathcal{W},\mathbf{x},y)\right]\Pr_{(\mathbf{x},y)\sim\mathcal{D}}\left[\mathfrak{N}(\mathcal{W},\mathbf{x},y)\right] \\
& + \underbrace{\mathbb{E}_{(\mathbf{x},y)\sim\mathcal{D}}\left[\mathcal{L}_0(\mathcal{W},\mathbf{x},y)\big|\ \neg\mathfrak{N}(\mathcal{W},\mathbf{x},y)\right]}_{\leq 1}\underbrace{\Pr_{(\mathbf{x},y)\sim\mathcal{D}}\left[\neg\mathfrak{N}(\mathcal{W},\mathbf{x},y)\right]}_{\mu_{\mathcal{D}}(\{(\rho_r,\Delta_r)\}_{r=1}^{R'},\mathcal{W})} \\
\leq {} & \mathbb{E}_{(\mathbf{x},y)\sim\mathcal{D}}\left[\mathcal{L}_0(\mathcal{W},\mathbf{x},y)\big|\ \mathfrak{N}(\mathcal{W},\mathbf{x},y)\right]\Pr_{(\mathbf{x},y)\sim\mathcal{D}}\left[\mathfrak{N}(\mathcal{W},\mathbf{x},y)\right] \\
& + \mu_{\mathcal{D}}(\{(\rho_r,\Delta_r)\}_{r=1}^{R'},\mathcal{W}) \qquad\qquad\qquad\qquad\qquad\qquad\qquad (5)
\end{aligned}
$$

To further continue the upper bound on the left hand side, we turn our attention to the stochastic classifier's loss on the noise-resilient part of the distribution $\mathcal{D}$ (we will lower bound this term in terms of the first term on the right hand side above). For simplicity of notations, we will write $\mathcal{D}'$ to denote the distribution $\mathcal{D}$ conditioned on $\mathfrak{N}(\mathcal{W}, \mathbf{x}, y)$. Also, let $\mathfrak{U}(\tilde{\mathcal{W}}, \mathbf{x}, y)$ be the favorable event that for a given data point $(\mathbf{x}, y)$ and a draw of the stochastic classifier, $\tilde{\mathcal{W}}$, it is the case that for every $r$, $|\rho_r(\mathcal{W}, \mathbf{x}, y) - \rho_r(\tilde{\mathcal{W}}, \mathbf{x}, y)| \leq \Delta_r/2$. Then, the stochastic classifier's loss $\mathcal{L}_{1/2}$ on $\mathcal{D}'$ is:

$$
\mathbb{E}_{\tilde{\mathcal{W}}\sim Q}\left[\mathbb{E}_{(\mathbf{x},y)\sim\mathcal{D}'}\left[\mathcal{L}_{1/2}(\tilde{\mathcal{W}},\mathbf{x},y)\right]\right] = \mathbb{E}_{(\mathbf{x},y)\sim\mathcal{D}'}\left[\mathbb{E}_{\tilde{\mathcal{W}}\sim Q}\left[\mathcal{L}_{1/2}(\tilde{\mathcal{W}},\mathbf{x},y)\right]\right]
$$

splitting the inner expectation over the favorable and unfavorable perturbations, and using linearity of expectations,

$$
\begin{aligned}
= {} & \mathbb{E}_{(\mathbf{x},y)\sim\mathcal{D}'}\left[\mathbb{E}_{\tilde{\mathcal{W}}\sim Q}\left[\mathcal{L}_{1/2}(\tilde{\mathcal{W}},\mathbf{x},y)\big|\ \mathfrak{U}(\tilde{\mathcal{W}},\mathbf{x},y)\right]\Pr_{\tilde{\mathcal{W}}\sim Q}\left[\mathfrak{U}(\tilde{\mathcal{W}},\mathbf{x},y)\right]\right] \\
& + \mathbb{E}_{(\mathbf{x},y)\sim\mathcal{D}'}\left[\mathbb{E}_{\tilde{\mathcal{W}}\sim Q}\left[\mathcal{L}_{1/2}(\tilde{\mathcal{W}},\mathbf{x},y)\big|\ \neg\mathfrak{U}(\tilde{\mathcal{W}},\mathbf{x},y)\right]\Pr_{\tilde{\mathcal{W}}\sim Q}\left[\neg\mathfrak{U}(\tilde{\mathcal{W}},\mathbf{x},y)\right]\right]
\end{aligned}
$$

to lower bound this, we simply ignore the second term (which is positive)

$$
\geq \mathbb{E}_{(\mathbf{x},y)\sim\mathcal{D}'}\left[\mathbb{E}_{\tilde{\mathcal{W}}\sim Q}\left[\mathcal{L}_{1/2}(\tilde{\mathcal{W}},\mathbf{x},y)\big|\ \mathfrak{U}(\tilde{\mathcal{W}},\mathbf{x},y)\right]\Pr_{\tilde{\mathcal{W}}\sim Q}\left[\mathfrak{U}(\tilde{\mathcal{W}},\mathbf{x},y)\right]\right].
$$

Next, we use the following fact: if $\mathcal{L}_{1/2}(\tilde{\mathcal{W}}, \mathbf{x}, y) = 0$, then for all $r$, $\rho_r(\tilde{\mathcal{W}}, x, y) \geq \Delta_r/2$ and if $\tilde{\mathcal{W}}$ is a favorable perturbation of $\mathcal{W}$, then for all $r$, $\rho_r(\mathcal{W}, x, y) \geq \rho_r(\tilde{\mathcal{W}}, \mathbf{x}, y) - \Delta_r/2 > 0$ i.e., $\mathcal{L}_{1/2}(\tilde{\mathcal{W}}, \mathbf{x}, y) = 0$ implies $\mathcal{L}_0(\mathcal{W}, \mathbf{x}, y) = 0$. Hence if $\tilde{\mathcal{W}}$ is a favorable perturbation then, $\mathcal{L}_{1/2}(\tilde{\mathcal{W}}, \mathbf{x}, y) \geq \mathcal{L}_0(\mathcal{W}, \mathbf{x}, y)$. Therefore, we can lower bound the above expression by replacing the stochastic classifier with the deterministic classifier (and thus ridding ourselves of the expectation over $Q$):

$$
\geq \mathbb{E}_{(\mathbf{x},y)\sim\mathcal{D}'}\left[\mathcal{L}_0(\mathcal{W},\mathbf{x},y)\Pr_{\tilde{\mathcal{W}}\sim Q}\left[\mathfrak{U}(\tilde{\mathcal{W}},\mathbf{x},y)\right]\right].
$$

Since the favorable perturbations for a fixed datapoint drawn from $\mathcal{D}'$ have sufficiently high probability (that is, $\Pr_{\tilde{\mathcal{W}}\sim Q}\left[\mathfrak{U}(\tilde{\mathcal{W}},\mathbf{x},y)\right] \geq 1 - 1/\sqrt{m}$), we have:

$$
\geq \left(1 - \frac{1}{\sqrt{m}}\right)\mathbb{E}_{(\mathbf{x},y)\sim\mathcal{D}'}\left[\mathcal{L}_0(\mathcal{W},\mathbf{x},y)\right].
$$

Thus, we have a lower bound on the stochastic classifier's loss that is in terms of the deterministic classifier's loss on the noise-resilient datapoints. Rearranging it, we get an upper bound on the latter:

$$\mathbb{E}_{(\mathbf{x},y)\sim\mathcal{D}'}\left[\mathcal{L}_0(\mathcal{W},\mathbf{x},y)\right] \leq \frac{1}{\left(1-\frac{1}{\sqrt{m}}\right)}\mathbb{E}_{\tilde{\mathcal{W}}\sim Q}\left[\mathbb{E}_{(\mathbf{x},y)\sim\mathcal{D}'}\left[\mathcal{L}_{1/2}(\tilde{\mathcal{W}},\mathbf{x},y)\right]\right]$$

$$\leq \left(1+\frac{1}{\sqrt{m}-1}\right)\mathbb{E}_{\tilde{\mathcal{W}}\sim Q}\left[\mathbb{E}_{(\mathbf{x},y)\sim\mathcal{D}'}\left[\mathcal{L}_{1/2}(\tilde{\mathcal{W}},\mathbf{x},y)\right]\right]$$

$$\leq \mathbb{E}_{\tilde{\mathcal{W}}\sim Q}\left[\mathbb{E}_{(\mathbf{x},y)\sim\mathcal{D}'}\left[\mathcal{L}_{1/2}(\tilde{\mathcal{W}},\mathbf{x},y)\right]\right]$$

$$+\frac{1}{\sqrt{m}-1}\underbrace{\mathbb{E}_{\tilde{\mathcal{W}}\sim Q}\left[\mathbb{E}_{(\mathbf{x},y)\sim\mathcal{D}'}\left[\mathcal{L}_{1/2}(\tilde{\mathcal{W}},\mathbf{x},y)\right]\right]}_{\leq 1}$$

$$\leq \mathbb{E}_{\tilde{\mathcal{W}}\sim Q}\left[\mathbb{E}_{(\mathbf{x},y)\sim\mathcal{D}'}\left[\mathcal{L}_{1/2}(\tilde{\mathcal{W}},\mathbf{x},y)\right]\right]+\frac{1}{\sqrt{m}-1}$$

Thus, we have an upper bound on the expected loss of the deterministic classifier $\mathcal{W}$ on the noise-resilient part of the distribution. Plugging this back in the first term of the upper bound on the deterministic classifier's loss on the whole distribution $\mathcal{D}$ in Equation 5 we get :

$$\mathbb{E}_{(\mathbf{x},y)\sim\mathcal{D}}\left[\mathcal{L}_0(\mathcal{W},\mathbf{x},y)\right] \leq \left(\mathbb{E}_{\tilde{\mathcal{W}}\sim Q}\left[\mathbb{E}_{(\mathbf{x},y)\sim\mathcal{D}'}\left[\mathcal{L}_{1/2}(\tilde{\mathcal{W}},\mathbf{x},y)\right]\right]+\frac{1}{\sqrt{m}-1}\right)\Pr_{(\mathbf{x},y)\sim\mathcal{D}}\left[\mathfrak{N}(\mathcal{W},\mathbf{x},y)\right]+$$

$$\mu_{\mathcal{D}}(\{(\rho_r,\Delta_r)\}_{r=1}^{R'},\mathcal{W})$$

rearranging, we get:

$$\leq \left(\mathbb{E}_{\tilde{\mathcal{W}}\sim Q}\left[\mathbb{E}_{(\mathbf{x},y)\sim\mathcal{D}'}\left[\mathcal{L}_{1/2}(\tilde{\mathcal{W}},\mathbf{x},y)\right]\right]\right)\Pr_{(\mathbf{x},y)\sim\mathcal{D}}\left[\mathfrak{N}(\mathcal{W},\mathbf{x},y)\right]+$$

$$\mu_{\mathcal{D}}(\{(\rho_r,\Delta_r)\}_{r=1}^{R'},\mathcal{W})+\frac{1}{\sqrt{m}-1}\underbrace{\Pr_{(\mathbf{x},y)\sim\mathcal{D}}\left[\mathfrak{N}(\mathcal{W},\mathbf{x},y)\right]}_{\leq 1}$$

rewriting the expectation over $\mathcal{D}'$ explicitly as an expectation over $\mathcal{D}$ conditioned on $\mathfrak{N}(\mathcal{W},\mathbf{x},y)$, we get:

$$\leq \left(\mathbb{E}_{\tilde{\mathcal{W}}\sim Q}\left[\mathbb{E}_{(\mathbf{x},y)\sim\mathcal{D}}\left[\mathcal{L}_{1/2}(\tilde{\mathcal{W}},\mathbf{x},y)\middle|\ \mathfrak{N}(\mathcal{W},\mathbf{x},y)\right]\Pr_{(\mathbf{x},y)\sim\mathcal{D}}\left[\mathfrak{N}(\mathcal{W},\mathbf{x},y)\right]\right]\right)+$$

$$\mu_{\mathcal{D}}(\{(\rho_r,\Delta_r)\}_{r=1}^{R'},\mathcal{W})+\frac{1}{\sqrt{m}-1}$$

the first term above is essentially an expectation of a loss over the distribution $\mathcal{D}$ with the loss set to be zero over the non-noise-resilient datapoints and set to be $\mathcal{L}_{1/2}$ over the noise-resilient datapoints; thus we can upper bound it with the expectation of the $\mathcal{L}_{1/2}$ loss over the whole distribution $\mathcal{D}$:

$$\leq \mathbb{E}_{\tilde{\mathcal{W}}\sim Q}\left[\mathbb{E}_{(\mathbf{x},y)\sim\mathcal{D}}\left[\mathcal{L}_{1/2}(\tilde{\mathcal{W}},\mathbf{x},y)\right]\right]+\cdot\mu_{\mathcal{D}}(\{(\rho_r,\Delta_r)\}_{r=1}^{R'},\mathcal{W})+\frac{1}{\sqrt{m}-1} \tag{6}$$

Now observe that we can upper bound the first term here using the PAC-Bayesian bound by plugging in $\mathcal{L}_{1/2}$ for the generic $\mathcal{L}$ in Equation 4; however, the bound would still be in terms of the stochastic classifier's train error. To get the generalization bound we seek, which involves the deterministic classifier's train error, we need to take one final step mirroring these tricks on the train loss.

RELATING THE STOCHASTIC CLASSIFIER'S TRAIN LOSS TO DETERMINISTIC CLASSIFIER'S TRAIN LOSS. Our analysis here is almost identical to the previous analysis. Instead of working with the distribution $\mathcal{D}$ and $\mathcal{D}'$ we will work with the training data set $S$ and a subset of it $S'$ for which noise resilience property is satisfied by $\mathcal{W}$. Below, to make the presentation neater, we use $(\mathbf{x},y)\sim S$ to denote uniform sampling from $S$.

First, we upper bound the stochastic classifier's train loss ($\mathcal{L}_{1/2}$) as follows:

$$\mathbb{E}_{\tilde{\mathcal{W}} \sim Q} \left[ \mathbb{E}_{(\mathbf{x},y) \sim S} \left[ \mathcal{L}_{1/2}(\tilde{\mathcal{W}}, \mathbf{x}, y) \right] \right] = \mathbb{E}_{(\mathbf{x},y) \sim S} \left[ \mathbb{E}_{\tilde{\mathcal{W}} \sim Q} \left[ \mathcal{L}_{1/2}(\tilde{\mathcal{W}}, \mathbf{x}, y) \right] \right]$$

splitting over the noise-resilient points $S'$ ($(\mathbf{x}, y) \in S$ for which $\mathfrak{N}(\mathcal{W}, \mathbf{x}, y)$ holds) like in Equation 5, we can upper bound as:

$$\leq \mathbb{E}_{(\mathbf{x},y) \sim S'} \left[ \mathbb{E}_{\tilde{\mathcal{W}} \sim Q} \left[ \mathcal{L}_{1/2}(\tilde{\mathcal{W}}, \mathbf{x}, y) \right] \right] \Pr_{(\mathbf{x},y) \sim S} \left[ (\mathbf{x}, y) \in S' \right]$$
$$+ \hat{\mu}_S(\{(\rho_r, \Delta_r)\}_{r=1}^{R'}, \mathcal{W}) \tag{7}$$

We can upper bound the first term by first splitting it over the favorable and unfavorable perturbations like we did before:

$$\mathbb{E}_{(\mathbf{x},y) \sim S'} \left[ \mathbb{E}_{\tilde{\mathcal{W}} \sim Q} \left[ \mathcal{L}_{1/2}(\tilde{\mathcal{W}}, \mathbf{x}, y) \right] \right]$$
$$= \mathbb{E}_{(\mathbf{x},y) \sim S'} \left[ \mathbb{E}_{\tilde{\mathcal{W}} \sim Q} \left[ \mathcal{L}_{1/2}(\tilde{\mathcal{W}}, \mathbf{x}, y) \middle| \mathfrak{U}(\tilde{\mathcal{W}}, \mathbf{x}, y) \right] \Pr_{\tilde{\mathcal{W}} \sim Q} \left[ \mathfrak{U}(\tilde{\mathcal{W}}, \mathbf{x}, y) \right] \right]$$
$$+ \mathbb{E}_{(\mathbf{x},y) \sim S'} \left[ \mathbb{E}_{\tilde{\mathcal{W}} \sim Q} \left[ \mathcal{L}_{1/2}(\tilde{\mathcal{W}}, \mathbf{x}, y) \middle| \neg\mathfrak{U}(\tilde{\mathcal{W}}, \mathbf{x}, y) \right] \Pr_{\tilde{\mathcal{W}} \sim Q} \left[ \neg\mathfrak{U}(\tilde{\mathcal{W}}, \mathbf{x}, y) \right] \right]$$

To upper bound this, we apply a similar argument. First, if $\mathcal{L}_{1/2}(\tilde{\mathcal{W}}, \mathbf{x}, y) = 1$, then $\exists r$ such that $\rho_r(\tilde{\mathcal{W}}, x, y) < \Delta_r/2$ and if $\tilde{\mathcal{W}}$ is a favorable perturbation then for that value of $r$, $\rho_r(\mathcal{W}, x, y) < \rho_r(\tilde{\mathcal{W}}, \mathbf{x}, y) + \Delta_r/2 < \Delta_r$. Thus if $\tilde{\mathcal{W}}$ is a favorable perturbation then, $\mathcal{L}_1(\mathcal{W}, \mathbf{x}, y) = 1$ whenever $\mathcal{L}_{1/2}(\tilde{\mathcal{W}}, \mathbf{x}, y) = 1$ i.e., $\mathcal{L}_{1/2}(\tilde{\mathcal{W}}, \mathbf{x}, y) \leq \mathcal{L}_1(\mathcal{W}, \mathbf{x}, y)$. Next, we use the fact that the unfavorable perturbations for a fixed datapoint drawn from $S'$ have sufficiently low probability i.e., $\Pr_{\tilde{\mathcal{W}} \sim Q} \left[ \neg\mathfrak{U}(\tilde{\mathcal{W}}, \mathbf{x}, y) \right] \leq 1/\sqrt{m}$. Then, we get the following upper bound on the above equations, by replacing the stochastic classifier with the deterministic classifier (and thus ignoring the expectation over $Q$):

$$\leq \mathbb{E}_{(\mathbf{x},y) \sim S'} \left[ \mathbb{E}_{\tilde{\mathcal{W}} \sim Q} \left[ \mathcal{L}_1(\mathcal{W}, \mathbf{x}, y) \middle| \mathfrak{U}(\tilde{\mathcal{W}}, \mathbf{x}, y) \right] \underbrace{\Pr_{\tilde{\mathcal{W}} \sim Q} \left[ \mathfrak{U}(\tilde{\mathcal{W}}, \mathbf{x}, y) \right]}_{\leq 1} \right]$$

$$+ \mathbb{E}_{(\mathbf{x},y) \sim S'} \left[ \underbrace{\mathbb{E}_{\tilde{\mathcal{W}} \sim Q} \left[ \mathcal{L}_{1/2}(\tilde{\mathcal{W}}, \mathbf{x}, y) \middle| \neg\mathfrak{U}(\tilde{\mathcal{W}}, \mathbf{x}, y) \right]}_{\leq 1} \frac{1}{\sqrt{m}} \right]$$

$$\leq \mathbb{E}_{(\mathbf{x},y) \sim S'} \left[ \mathcal{L}_1(\mathcal{W}, \mathbf{x}, y) \right] + \frac{1}{\sqrt{m}}$$

Plugging this back in the first term of Equation 7, we get:

$$\mathbb{E}_{\tilde{\mathcal{W}} \sim Q} \left[ \mathbb{E}_{(\mathbf{x},y) \sim S} \left[ \mathcal{L}_{1/2}(\tilde{\mathcal{W}}, \mathbf{x}, y) \right] \right] \le \left( \mathbb{E}_{(\mathbf{x},y) \sim S'} \left[ \mathcal{L}_1(\mathcal{W}, \mathbf{x}, y) \right] + \frac{1}{\sqrt{m}} \right) \Pr_{(\mathbf{x},y) \sim S} \left[ (\mathbf{x}, y) \in S' \right]$$

$$+ \hat{\mu}_S(\{(\rho_r, \Delta_r)\}_{r=1}^{R'}, \mathcal{W})$$

$$\le \mathbb{E}_{(\mathbf{x},y) \sim S'} \left[ \mathcal{L}_1(\mathcal{W}, \mathbf{x}, y) \right] \Pr_{(\mathbf{x},y) \sim S} \left[ (\mathbf{x}, y) \in S' \right]$$

$$+ \hat{\mu}_S(\{(\rho_r, \Delta_r)\}_{r=1}^{R'}, \mathcal{W}) + \frac{1}{\sqrt{m}} \underbrace{\Pr_{(\mathbf{x},y) \sim S} \left[ (\mathbf{x}, y) \in S' \right]}_{\le 1}$$

$$\le \mathbb{E}_{(\mathbf{x},y) \sim S'} \left[ \mathcal{L}_1(\mathcal{W}, \mathbf{x}, y) \right] \Pr_{(\mathbf{x},y) \sim S} \left[ (\mathbf{x}, y) \in S' \right] + \frac{1}{\sqrt{m}}$$

$$+ \hat{\mu}_S(\{(\rho_r, \Delta_r)\}_{r=1}^{R'}, \mathcal{W})$$

since the first term is effectively the expectation of a loss over the whole distribution with the loss set to be zero on the non-noise-resilient points and set to $\mathcal{L}_1$ over the rest, we can upper bound it by setting the loss to be $\mathcal{L}_1$ over the whole distribution:

$$\le \mathbb{E}_{(\mathbf{x},y) \sim S} \left[ \mathcal{L}_1(\mathcal{W}, \mathbf{x}, y) \right] + \hat{\mu}_S(\{(\rho_r, \Delta_r)\}_{r=1}^{R'}, \mathcal{W}) + \frac{1}{\sqrt{m}}$$

Applying the above upper bound and the bound in Equation 6 into the PAC-Bayesian result of Equation 4 yields our result (Note that combining these equations would produce the term $\frac{1}{\sqrt{m}} + \frac{1}{\sqrt{m-1}}$ which is at most $\frac{2}{\sqrt{m-1}}$, which we reflect in the final bound. ).

$\square$

## D    PROOF FOR THEOREM 3.1

In this section, we present the proof for the abstract generalization guarantee presented in Section 3. Our proof is based on the following recursive inequality that we demonstrate for all $r \le R$ (we will prove a similar, but slightly different inequality for $r = R + 1$):

$$\Pr_{(\mathbf{x},y) \sim \mathcal{D}} \left[ \exists q {\le} r, \exists l \; \rho_{q,l}(\mathcal{W}, \mathbf{x}, y) < 0 \right] \le \Pr_{(\mathbf{x},y) \sim \mathcal{D}} \left[ \exists q {<} r, \exists l \; \rho_{q,l}(\mathcal{W}, \mathbf{x}, y) < 0 \right]$$

$$+ \underbrace{\tilde{\mathcal{O}} \left( \sqrt{\frac{2KL(\mathcal{N}(\mathcal{W}, \sigma^2 I) \| P)}{m - 1}} \right)}_{\text{generalization error for condition } r} \qquad (8)$$

Recall that the $r$th condition in Equation 1 is that $\forall l, \rho_{r,l} > \Delta_{r,l}^{\star}$. Above, we bound the probability mass of test points such that any one of the first $r$ conditions in Equation 1 is not even approximately satisfied, in terms of the probability mass of points where one of the first $r - 1$ conditions is not even approximately satisfied, and a term that corresponds to how much error there can be in generalizing the $r$th condition from the training data.

Our proof crucially relies on Theorem C.1. This theorem provides an upper bound on the proportion of test data that fail to satisfy a set of conditions, in terms of four quantities. The first quantity is the proportion of training data that do not satisfy the conditions; the second and third quantities, which we will in short refer to as $\hat{\mu}_S$ and $\hat{\mu}_\mathcal{D}$, correspond to the proportion of training and test data on which the properties involved in the conditions are not noise-resilient. The fourth quantity is the generalization error.

First, we consider the base case when $r = 1$, and apply the PAC-Bayes-based guarantee from Theorem C.1 on the first set of properties $\{\rho_{1,1}, \rho_{1,2}, \cdots\}$ and their corresponding constants

$\{\Delta^\star_{1,1}, \Delta^\star_{1,2}, \cdots\}$. First we have from our assumption (in the main theorem statement) that on all the training data, the condition $\rho_{1,l}(\mathcal{W}, \mathbf{x}, y) > \Delta^\star_{1,l}$ is satisfied for all possible $l$. Thus, the first term in the upper bound in Theorem C.1 is zero. Next, we can show that the terms $\hat{\mu}_S$ and $\hat{\mu}_\mathcal{D}$ would be zero too. This follows from the fact that the constraint in Equation 2 holds in this framework. Specifically, applying this equation for $r = 1$, for $\sigma = \sigma^\star$, we get that for all possible $(\mathbf{x}, y)$ the following inequality holds:

$$\Pr_{\mathcal{U} \sim \mathcal{N}(0, (\sigma^\star)^2 I)} \left[ \forall l \ \ |\rho_{1,l}(\mathcal{W} + \mathcal{U}, \mathbf{x}, y) - \rho_{l,1}(\mathcal{W}, \mathbf{x}, y)| > \frac{\Delta_{1,l}(\sigma^\star)}{2} \right] \le \frac{1}{R\sqrt{m}}.$$

Since, $\sigma^\star$ was chosen such that $\Delta_1(\sigma^\star) \le \Delta^\star_1$, we have:

$$\Pr_{\mathcal{U}} \left[ \forall l \ \ |\rho_{1,l}(\mathcal{W} + \mathcal{U}, \mathbf{x}, y) - \rho_{l,1}(\mathcal{W}, \mathbf{x}, y)| > \frac{\Delta^\star_1}{2} \right] \le \frac{1}{R\sqrt{m}}.$$

Effectively this establishes that the noise-resilience requirement of Equation 3 in Theorem C.1 holds on all possible inputs, thus proving our claim that the terms $\hat{\mu}_S$ and $\hat{\mu}_\mathcal{D}$ would be zero. Thus, we will get that

$$\Pr_{(\mathbf{x},y) \sim \mathcal{D}} \left[ \exists l \ \rho_{1,l}(\mathcal{W}, \mathbf{x}, y) < 0 \right] \le \tilde{\mathcal{O}} \left( \sqrt{\frac{2KL(\mathcal{N}(\mathcal{W}, \sigma^2 I) \| P)}{m-1}} \right)$$

which proves the recursion statement for the base case.

To prove the recursion for some arbitrary $r \le R$, we again apply the PAC-Bayes-based guarantee from Theorem C.1, but on the union of the first $r$ sets of properties. Again, we will have that the first term in the guarantee would be zero, since the corresponding conditions are satisfied on the training data. Now, to bound the proportion of bad points $\hat{\mu}_S$ and $\hat{\mu}_\mathcal{D}$, we make the following claim:

> the network is noise-resilient as per Equation 3 in Theorem C.1 for any input that satisfies the $r-1$ conditions approximately i.e., $\forall q \le r-1$ and $\forall l, \rho_{q,l}(\mathcal{W}, \mathbf{x}, y) > 0$.

The above claim can be used to prove Equation 8 as follows. Since all the conditions are assumed to be satisfied by a margin on the training data, this claim immediately implies that $\hat{\mu}_S$ is zero. Similarly, this claim implies that for the test data, we can bound $\mu_\mathcal{D}$ in terms of $\Pr_{(\mathbf{x},y) \sim \mathcal{D}} \left[ \exists q < r \ \exists l \ \rho_{q,l}(\mathcal{W}, \mathbf{x}, y) < 0 \right]$, thus giving rise to the recursion in Equation 8.

Now, to prove our claim, consider an input $(\mathbf{x}, y)$ such that $\rho_{q,l}(\mathcal{W}, \mathbf{x}, y) > 0$ for $q = 1, 2, \cdots, r-1$ and for all possible $l$. First from the assumption in our theorem statement that $\Delta_{q,l}(\sigma^\star) \le \Delta^\star_{q,l}$, we have the following upper bound on the proportion of parameter perturbations under which any of the properties in the first $r$ sets suffer a large perturbation:

$$\Pr_{\mathcal{U}} \left[ \exists q \le r \ \exists l \ : \ |\rho_{q,l}(\mathcal{W}, \mathbf{x}, y) - \rho_{q,l}(\mathcal{W} + \mathcal{U}, \mathbf{x}, y)| > \frac{\Delta^\star_{q,l}}{2} \right]$$

$$\le \Pr_{\mathcal{U}} \left[ \exists q \le r \ \exists l \ : \ |\rho_{q,l}(\mathcal{W}, \mathbf{x}, y) - \rho_{q,l}(\mathcal{W} + \mathcal{U}, \mathbf{x}, y)| > \frac{\Delta_{q,l}(\sigma^\star)}{2} \right]$$

Now, this can be expanded as a summation over $q = 1, 2, \cdots, r$ as:

$$\le \sum_{q=1}^{r} Pr \Big[ \exists l \ |\rho_{q,l}(\mathcal{W} + \mathcal{U}, \mathbf{x}, y) - \rho_{q,l}(\mathcal{W}, \mathbf{x}, y)| > \frac{\Delta_q(\sigma^\star)}{2} \wedge$$

$$\forall q' < q, \ \exists l \ |\rho_{q',l}(\mathcal{W}, \mathbf{x}, y) - \rho_{q',l}(\mathcal{W} + \mathcal{U}, \mathbf{x}, y)| < \frac{\Delta_{q'}(\sigma^\star)}{2} \Big]$$

and because $(\mathbf{x}, y)$ satisfies $\rho_{q,l}(\mathcal{W}, \mathbf{x}, y) > 0$ for $q = 1, 2, \cdots, r - 1$ and for all possible $l$, by the constraint assumed in Equation 2, we have:

$$\leq \sum_{q=1}^{r} \frac{1}{(R+1)\sqrt{m}} \leq \frac{1}{\sqrt{m}}$$

Thus, we have that $(\mathbf{x}, y)$ satisfies the noise-resilience condition from Equation 3 in Theorem C.1 if it also satisfies $\rho_{q,l}(\mathcal{W}, \mathbf{x}, y) > 0$ for $q = 1, 2, \cdots, r - 1$ and for all possible $l$. This proves our claim, and hence in turn proves the recursion in Equation 8.

Finally, we can apply a similar argument for the $R + 1$th set of input-dependent properties (which is a singleton set consisting of the margin of the network) with a small change since the first term in the guarantee from Theorem C.1 is not explicitly assumed to be zero; we will get an inequality in terms of the number of training points that are not classified correctly by a margin, giving rise to the margin-based bound:

$$\Pr_{(\mathbf{x},y)\sim\mathcal{D}} \left[ \exists q \leq R{+}1 \; \exists l, \rho_{q,l}(\mathcal{W}, \mathbf{x}, y) < 0 \right] \leq \frac{1}{m} \sum_{(\mathbf{x},y)\in S} \mathbf{1}\left[ \rho_{R+1,1}(\mathcal{W}, \mathbf{x}, y) < \Delta_{R,1} \right]$$
$$+ \Pr_{(\mathbf{x},y)\sim\mathcal{D}} \left[ \exists q \leq R \; \exists l \; \rho_{q,l}(\mathcal{W}, \mathbf{x}, y) < 0 \right]$$
$$+ \tilde{\mathcal{O}}\left( \sqrt{\frac{2KL(\mathcal{N}(\mathcal{W}, \sigma^2 I)\|P)}{m-1}} \right)$$

Note that in the first term on the right hand side, $\rho_{R+1,1}(\mathcal{W}, \mathbf{x}, y)$ corresponds to the margin of the classifier on $(\mathbf{x}, y)$. Now, by using the fact that the test error is upper bounded by the left hand side in the above equation, applying the recursion on the right hand side $R + 1$ times, we get our final result.

# E   PERTURBATION BOUNDS

## E.1   OVERVIEW OF THE BOUNDS.

In this section, we will quantify the noise resilience of a network in different aspects. Each of our bounds has the following structure: we fix an input point $(\mathbf{x}, y)$, and then say that with high probability over a Gaussian perturbation of the network's parameters, a particular input-dependent property of the network (say the output of the network, or the pre-activation value of a particular unit $h$ at a particular layer $d$, or say the Frobenius norm sof its active weight matrices), changes only by a small magnitude proportional to the variance $\sigma^2$ of the Gaussian perturbation of the parameters.

A key feature of our bounds is that they do not involve the product of the spectral norm of the weight matrices and hence save us an exponential factor in the final generalization bound. Instead, the bound in the perturbation of a particular property will be in terms of i) the magnitude of the some 'preceding' properties (typically, these are properties of the lower layers) of the network, and ii) how those preceding properties themselves respond to perturbations. For example, an upper bound in the perturbation of the $d$th layer's output would involve the $\ell_2$ norm of the lower layers $d' < d$, and how much they would blow up under these perturbations.

## E.2   SOME NOTATIONS.

To formulate our lemma statement succinctly, we design a notation wherein we define a set of 'tolerance parameters' which we will use to denote the extent of perturbation suffered by a particular property of the network.

Let $\hat{\mathscr{C}}$ denote a 'set' (more on what we mean by a set below) of positive tolerance values, consisting of the following elements:

1. $\hat{\alpha}_d$, for each layer $d = 1, \cdots, D-1$ (a tolerance value for the $\ell_2$ norm of the output of layer $d$)

2. $\hat{\gamma}_d$ for each layer $d = 1, \cdots, D$. (a tolerance value for the magnitude of the pre-activations of layer $d$)

3. $\hat{\zeta}_{d/d'}$ for each layer $d = 1, 2, \cdots, D$, and $d' = 1, \cdots, d$ (a tolerance value for the $\ell_2$ norm of each row of the Jacobians at layer $d$)

4. $\hat{\psi}_{d/d'}$ for each layer $d = 1, 2, \cdots, D$, and $d' = 1, \cdots, d$ (a tolerance value for the spectral norm of the Jacobians at layer $d$)

**Notes about (abuse of) notation:**

- We call $\hat{\mathscr{C}}$ a 'set' to denote a group of related constants into a single symbol. Each element in this set has a particular semantic associated with it, unlike the standard notation of a set, and so when we refer to, say $\hat{\zeta}_{d/d'} \in \hat{\mathscr{C}}$, we are indexing into the set to pick a particular element.

- We will use the subscript $\hat{\mathscr{C}}_d$ to index into a subset of only those tolerance values corresponding to layers from $1$ until $d$.

Next we define two events. The first event formulates the scenario that for a given input, a particular perturbation of the weights until layer $d$ brought about very little change in the properties of these layers (within some tolerance levels). The second event formulates the scenario that the perturbation did not flip the activation states of the network.

**Definition E.1.** *Given an input* $\mathbf{x}$*, and an arbitrary set of constants* $\hat{\mathscr{C}}'$*, for any perturbation* $\mathcal{U}$ *of* $\mathcal{W}$*, we denote by* PERT-BOUND$(\mathcal{W}+\mathcal{U}, \hat{\mathscr{C}}', \mathbf{x})$ *the event that:*

- *for each* $\hat{\alpha}'_d \in \hat{\mathscr{C}}'$*, the perturbation in the* $\ell_2$ *norm of layer* $d$ *activations is bounded as* $\left| \left\| f^d(\mathbf{x}; \mathcal{W}) \right\| - \left\| f^d(\mathbf{x}; \mathcal{W}+\mathcal{U}) \right\| \right| \leq \hat{\alpha}'_d$*.*

- *for each* $\hat{\gamma}'_d \in \hat{\mathscr{C}}'$*, the maximum perturbation in the preactivation of hidden units on layer* $d$ *is bounded as* $\max_h \left| f^d(\mathbf{x}; \mathcal{W})[h] - f^d(\mathbf{x}; \mathcal{W}+\mathcal{U})[h] \right| \leq \hat{\gamma}'_d$*.*

- *for each* $\hat{\zeta}'_{d/d'} \in \hat{\mathscr{C}}'$*, the maximum perturbation in the* $\ell_2$ *norm of a row of the Jacobian* $d/d'$ *is bounded as* $\max_h \left| \left\| J^{d/d'}(\mathbf{x}; \mathcal{W})[h] \right\| - \left\| J^{d/d'}(\mathbf{x}; \mathcal{W}+\mathcal{U})[h] \right\| \right| \leq \hat{\zeta}'_{d/d'}$*.*

- *for each* $\hat{\psi}'_{d/d'} \in \hat{\mathscr{C}}'$*, the perturbation in the spectral norm of the Jacobian* $d/d'$ *is bounded as* $\left| \left\| J^{d/d'}(\mathbf{x}; \mathcal{W}) \right\|_2 - \left\| J^{d/d'}(\mathbf{x}; \mathcal{W}+\mathcal{U}) \right\|_2 \right| \leq \hat{\psi}'_{d/d'}$*.*

NOTE: If we supply only a subset of $\hat{\mathscr{C}}$ (say $\hat{\mathscr{C}}_d$ instead of the whole of $\hat{\mathscr{C}}$) to the above event, PERT-BOUND$(\mathcal{W}+\mathcal{U}, \cdot, \mathbf{x})$, then it would denote the event that the perturbations suffered by only that subset of properties is within the respective tolerance values.

Next, we define the event that the perturbations do not affect the activation states of the network.

**Definition E.2.** *For any perturbation* $\mathcal{U}$ *of the matrices* $\mathcal{W}$*, let* UNCHANGED-ACTS$_d(\mathcal{W}+\mathcal{U}, \mathbf{x})$ *denote the event that none of the activation states of the first* $d$ *layers change on perturbation.*

### E.3 MAIN LEMMA.

Our results here are styled similar to the equations required by Equation 2 presented in the main paper. For a given input point and for a particular property of the network, roughly, we bound the probability that a perturbation affects the value of the property while none of the preceding preceding properties themselves are perturbed beyond a certain tolerance level.

**Lemma E.1.** *Let* $\hat{\mathscr{C}}$ *be a set of constants (that denote the amount of perturbation in the properties preceding a considered property). For any* $\hat{\delta} > 0$*, define* $\hat{\mathscr{C}}'$ *(which is a bound on the perturbation of*

*a considered property) in terms of $\hat{\mathscr{C}}$ and the perturbation parameter $\sigma$, for all $d = 1, 2, \cdots, D$ and for all $d' = d - 1, \cdots, 1$ as follows:*

$$\hat{\alpha}'_d := \sigma \sum_{d'=1}^{d} \left\| J^{d/d'}(\mathbf{x}; \mathcal{W}) \right\|_F \left( \|f^{d'-1}(\mathbf{x}; \mathcal{W})\| + \hat{\alpha}_{d'-1} \right) \sqrt{2 \ln \frac{2DH}{\hat{\delta}}}$$

$$\hat{\gamma}'_d := \sigma \sum_{d'=1}^{d} \left\| J^{d/d'}(\mathbf{x}; \mathcal{W}) \right\|_{2,\infty} \left( \|f^{d'-1}(\mathbf{x}; \mathcal{W})\| + \hat{\alpha}_{d'-1} \right) \sqrt{2 \ln \frac{2DH}{\hat{\delta}}}$$

$$\hat{\zeta}'_{d/d'} := \sigma \left( \left\| J^{d-1/d'}(\mathbf{x}; \mathcal{W}) \right\|_F + \hat{\zeta}_{d-1/d'} \sqrt{H} \right) \sqrt{4 \ln \frac{DH}{\hat{\delta}}}$$

$$+ \sigma \sum_{d''=d'+1}^{d-1} \|W_d\|_{2,\infty} \left\| J^{d-1/d''}(\mathbf{x}; \mathcal{W}) \right\|_2 \left( \left\| J^{d''-1/d'}(\mathbf{x}; \mathcal{W}) \right\|_F + \hat{\zeta}_{d''-1/d'} \sqrt{H} \right) \sqrt{4 \ln \frac{DH}{\hat{\delta}}}$$

$$\hat{\psi}'_{d/d'} := \sigma \sqrt{H} \left( \left\| J^{d-1/d'}(\mathbf{x}; \mathcal{W}) \right\|_2 + \hat{\psi}_{d-1/d'} \right) \sqrt{2 \ln \frac{2DH}{\hat{\delta}}}$$

$$+ \sigma \sqrt{H} \sum_{d''=d'+1}^{d-1} \|W_d\|_2 \left\| J^{d-1/d''}(\mathbf{x}; \mathcal{W}) \right\|_2 \left( \left\| J^{d''-1/d'}(\mathbf{x}; \mathcal{W}) \right\|_2 + \hat{\psi}_{d''-1/d'} \right) \sqrt{2 \ln \frac{2DH}{\hat{\delta}}}$$

$$\hat{\alpha}'_0 = 0$$

$$\hat{\zeta}'_{d/d} := 0$$

$$\hat{\psi}'_{d/d} := 0$$

*Let $\mathcal{U}_d$ be sampled entrywise from $\mathcal{N}(0, \sigma^2)$ for any $d$. Then, the following statements hold good:*

**1. Bound on perturbation of of $\ell_2$ norm of the output of layer $d$.** *For all $d = 1, 2, \cdots, D$,*

$$Pr_{\mathcal{U}} \Big[ \neg \text{PERT-BOUND}(\mathcal{W} + \mathcal{U}, \{\hat{\alpha}'_d\}, \mathbf{x}) \wedge$$

$$\text{PERT-BOUND}(\mathcal{W} + \mathcal{U}, \hat{\mathscr{C}}_{d-1} \bigcup \{\hat{\zeta}_{d/d'}\}_{d'=1}^{d}, \mathbf{x}) \wedge \text{UNCHANGED-ACTS}_{d-1}(\mathcal{W} + \mathcal{U}, \mathbf{x}) \Big] \leq \hat{\delta}$$

**2. Bound on perturbation of pre-activations at layer $d$.** *For all $d = 1, 2, \cdots, D$,*

$$Pr_{\mathcal{U}} \Big[ \neg \text{PERT-BOUND}(\mathcal{W} + \mathcal{U}, \{\hat{\gamma}'_d\}, \mathbf{x}) \wedge$$

$$\text{PERT-BOUND}(\mathcal{W} + \mathcal{U}, \hat{\mathscr{C}}_{d-1} \bigcup \{\hat{\zeta}_{d/d'}\}_{d'=1}^{d} \bigcup \{\hat{\alpha}_d\}, \mathbf{x}) \wedge \text{UNCHANGED-ACTS}_{d-1}(\mathcal{W} + \mathcal{U}, \mathbf{x}) \Big] \leq \hat{\delta}$$

**3. Bound on perturbation of $\ell_2$ norm on the rows of the Jacobians $d/d'$.**

$$Pr_{\mathcal{U}} \Big[ \neg \text{PERT-BOUND}(\mathcal{W} + \mathcal{U}, \{\hat{\zeta}'_{d/d'}\}_{d'=1}^{d}, \mathbf{x}) \wedge$$

$$\text{PERT-BOUND}(\mathcal{W} + \mathcal{U}, \hat{\mathscr{C}}_{d-1}, \mathbf{x}) \wedge \text{UNCHANGED-ACTS}_{d-1}(\mathcal{W} + \mathcal{U}, \mathbf{x}) \Big] \leq \hat{\delta}$$

**4. Bound on perturbation of spectral norm of the Jacobians $d/d'$.**

$$Pr_{\mathcal{U}} \Big[ \neg \text{PERT-BOUND}(\mathcal{W} + \mathcal{U}, \{\hat{\psi}'_{d/d'}\}_{d'=1}^{d}, \mathbf{x}) \wedge$$

$$\text{PERT-BOUND}(\mathcal{W} + \mathcal{U}, \hat{\mathscr{C}}_{d-1}, \mathbf{x}) \wedge \text{UNCHANGED-ACTS}_{d-1}(\mathcal{W} + \mathcal{U}, \mathbf{x}) \Big] \leq \hat{\delta}$$

*Proof.* For the most part of this discussion, we will consider a perturbed network where all the hidden units are frozen to be at the same activation state as they were at, before the perturbation. We will denote the weights of such a network by $\mathcal{W}[+\mathcal{U}]$ and its output at the $d$th layer by $f^d(\mathbf{x}; \mathcal{W}[+\mathcal{U}])$. By having the activations states frozen, the Gaussian perturbations propagate linearly through the activations, effectively remaining as Gaussian perturbations; then, we can enjoy the well-established properties of the Gaussian even after they propagate.

PERTURBATION BOUND ON THE $\ell_2$ NORM OF LAYER $d$. We bound the change in the $\ell_2$ norm of the $d$th layer's output by applying a triangle inequality[6] after splitting it into a sum of vectors. Each summand here (which we define as $\mathbf{v}_{d'}$ for each $d' \leq d$) is the difference in the $d$th layer output on introducing noise in weight matrix $d'$ after having introduced noise into all the first $d' - 1$ weight matrices.

$$\left| \left\| f^d \left( \mathbf{x}; \mathcal{W}[+\mathcal{U}_d] \right) \right\| - \left\| f^d \left( \mathbf{x}; \mathcal{W} \right) \right\| \right| \leq \left\| f^d \left( \mathbf{x}; \mathcal{W}[+\mathcal{U}_d] \right) - f^d \left( \mathbf{x}; \mathcal{W} \right) \right\|$$

because the activations are ReLU, we can replace this with the perturbation of the pre-activation

$$\leq \left\| g^d \left( \mathbf{x}; \mathcal{W}[+\mathcal{U}_d] \right) - g^d \left( \mathbf{x}; \mathcal{W} \right) \right\|$$

$$\leq \left\| \sum_{d'=1}^{d} \left( \underbrace{g^d \left( \mathbf{x}; \mathcal{W}[+\mathcal{U}_{d'}] \right) - g^d \left( \mathbf{x}; \mathcal{W}[+\mathcal{U}_{d'-1}] \right)}_{:=\mathbf{v}_{d'}} \right) \right\|$$

$$\leq \sum_{d'=1}^{d} \left\| \mathbf{v}_{d'} \right\| = \sum_{d'=1}^{d} \sqrt{\sum_h v_{d',h}^2} \tag{9}$$

Here, $v_{d',h}$ is the perturbation in the preactivation of hidden unit $h$ on layer $d'$, brought about by perturbation of the $d'$th weight matrix in a network where only the first $d' - 1$ weight matrices have already been perturbed.

Now, for each $h$, we bound $v_{d',h}$ in Equation 9. Since the activations have been frozen we can rewrite each $v_{d',h}$ as the product of the $h$th row of the unperturbed network's Jacobian $d/d'$, followed by only the perturbation matrix $U_{d'}$, and then the output of the layer $d' - 1$. Concretely, we have[7][8]:

$$v_{d',h} = \overbrace{J^{d/d'} \left( \mathbf{x}; \mathcal{W} \right)[h]}^{1 \times H_{d'}} \underbrace{\overbrace{U_{d'}}^{H_{d'} \times H_{d'-1}} \overbrace{f^{d'-1} \left( \mathbf{x}; \mathcal{W}[+\mathcal{U}_{d'-1}] \right)}^{H_{d'-1} \times 1}}_{\text{spherical Gaussian}}$$

What do these random variables $v_{d',h}$ look like? Conditioned on $\mathcal{U}_{d'-1}$, the second part of our expansion of $v_{d',h}$, namely, $U_{d'} f^{d'-1} \left( \mathbf{x}; \mathcal{W}[+\mathcal{U}_{d'-1}] \right)$ is a multivariate spherical Gaussian (see Lemma B.2) of the form $\mathcal{N}(0, \sigma^2 \| f^{d'-1} \left( \mathbf{x}; \mathcal{W}[+\mathcal{U}_{d'-1}] \right) \|^2 I)$. As a result, conditioned on $\mathcal{U}_{d'-1}$, $v_{d',h}$ is a univariate Gaussian $\mathcal{N}(0, \sigma^2 \| J^{d/d'} \left( \mathbf{x}; \mathcal{W} \right)[h] \|^2 \| f^{d'-1} \left( \mathbf{x}; \mathcal{W}[+\mathcal{U}_{d'-1}] \right) \|^2)$.

Then, we can apply a standard Gaussian tail bound (see Lemma B.1) to conclude that with probability $1 - \hat{\delta}/DH$ over the draws of $U_{d'}$ (conditioned on any $\mathcal{U}_{d'-1}$), $v_{d',h}$ is bounded as:

$$|v_{d',h}| \leq \sigma \left\| J^{d/d'} \left( \mathbf{x}; \mathcal{W} \right)[h] \right\| \| f^{d'-1} \left( \mathbf{x}; \mathcal{W}[+\mathcal{U}_{d'-1}] \right) \| \sqrt{2 \ln \frac{2DH}{\hat{\delta}}}. \tag{10}$$

Then, by a union bound over all the hidden units on layer $d$, and for each $d'$, we have that with probability $1 - \hat{\delta}$, Equation 9 is upper bounded as:

---

[6]Specifically, for two vectors $\mathbf{a}, \mathbf{b}$, we have from triangle inequality that $\|\mathbf{b}\| \leq \|\mathbf{a}\| + \|\mathbf{b} - \mathbf{a}\|$ and $\|\mathbf{a}\| \leq \|\mathbf{b}\| + \|\mathbf{a} - \mathbf{b}\|$. As a result of this, we have: $-\|\mathbf{a} - \mathbf{b}\| \leq \|\mathbf{a}\| - \|\mathbf{b}\| \leq \|\mathbf{a} - \mathbf{b}\|$. We use this inequality in our proof.

[7]Below, we have used $H_d$ to denote the number of units on the $d$th layer (and this equals $H$ for the hidden units and $K$ for the output layer).

[8]Note that the succinct formula below holds good even for the corner case $d' = d$, where the first Jacobian-row term becomes a vector with zeros on all but the $h$th entry and therefore only the $h$th row of the perturbation matrix $U_{d'}$ will participate in the expression of $v_{d',h}$.

$$\sum_{d'} \sqrt{\sum_{h} v_{d',h}^2} \le \sum_{d'=1}^{d} \sigma \left\| J^{d/d'}(\mathbf{x};\mathcal{W}) \right\|_F \| f^{d'-1}(\mathbf{x};\mathcal{W}[+\mathcal{U}_{d'-1}]) \| \sqrt{2\ln\frac{2DH}{\hat{\delta}}}. \tag{11}$$

Using this we prove the probability bound in the lemma statement. To simplify notations, let us denote $\hat{\mathscr{C}}_{d-1} \cup \{\hat{\zeta}_{d/d'}\}_{d'=1}^{d}$ by $\hat{\mathscr{C}}_{\text{prev}}$. Furthermore, we will drop redundant symbols in the arguments of the events we have defined. Then, recall that we want to upper bound the following probability (we ignore the arguments $\mathcal{W}+\mathcal{U}$ and $\mathbf{x}$ for brevity):

$$\Pr\left[ (\neg\textsf{PERT-BOUND}(\{\hat{\alpha}_d'\})) \land \textsf{PERT-BOUND}(\hat{\mathscr{C}}_{\text{prev}}) \land \textsf{UNCHANGED-ACTS}_{d-1} \right]$$

Recall that Equation 11 is a bound on the perturbation of the $\ell_2$ norm of the $d$th layer's output when the activation states are explicitly frozen. If the perturbation we randomly draw happens to satisfy $\textsf{UNCHANGED-ACTS}_{d-1}$ then the bound in Equation 11 holds good even in the case where the activation states are not explicitly frozen. Furthermore, when $\textsf{PERT-BOUND}(\hat{\mathscr{C}}_{\text{prev}})$ holds, the bound in Equation 11 can be upper-bounded by $\hat{\alpha}_d'$ as defined in the lemma statement, because under $\textsf{PERT-BOUND}(\hat{\mathscr{C}}_{\text{prev}})$, the middle term in Equation 11 can be upper bounded using triangle inequality as $\| f^{d'-1}(\mathbf{x};\mathcal{W}[+\mathcal{U}_{d'-1}]) \| \le \| f^{d'-1}(\mathbf{x};\mathcal{W}) \| + \hat{\alpha}_{d'-1}$. Hence, the event above happens only for the perturbations for which Equation 11 fails and hence we have that the above probability term is upper bounded by $\hat{\delta}$.

PERTURBATION BOUND ON THE PREACTIVATION VALUES OF LAYER $d$.   Following the same analysis as above, the bound we are seeking here is essentially $\max_h \sum_{d'=1}^{d} |v_{d',h}|$. The bound follows similarly from Equation 10.

PERTURBATION BOUND ON THE $\ell_2$ NORM OF THE ROWS OF THE JACOBIAN $d/d'$.   We split this term like we did in the previous subsection, and apply triangle equality as follows:

$$\max_{h} \left| \left\| J^{d/d'}(\mathbf{x};\mathcal{W})[h] \right\| - \left\| J^{d/d'}(\mathbf{x};\mathcal{W}[+\mathcal{U}_d])[h] \right\| \right|$$

$$\le \max_{h} \left\| J^{d/d'}(\mathbf{x};\mathcal{W})[h] - J^{d/d'}(\mathbf{x};\mathcal{W}[+\mathcal{U}_d])[h] \right\|$$

$$\le \left\| \sum_{d''=1}^{d} \left( \underbrace{J^{d/d'}(\mathbf{x};\mathcal{W}[+\mathcal{U}_{d''}])[h] - J^{d/d'}(\mathbf{x};\mathcal{W}[+\mathcal{U}_{d''-1}])[h]}_{:=\mathbf{y}_h^{d''}} \right) \right\|$$

$$\le \max_{h} \sum_{d''=1}^{d} \left\| \mathbf{y}_h^{d''} \right\| = \max_{h} \sum_{d''=1}^{d} \sqrt{\sum_{h'} (y_{d'',h,h'})^2} \tag{12}$$

Here, we have defined $\mathbf{y}_h^{d''}$ to be the vector that corresponds to the difference in the $h$th row of the Jacobian $d/d'$ brought about by perturbing the $d''$th weight matrix, given that the first $d''-1$ matrices have already been perturbed. We use $h$ to iterate over the units in the $d$th layer and $h'$ to iterate over the units in the $d'$th layer.

Now, under the frozen activation states, when we perturb the weight matrices from 1 uptil $d'$, since these matrices are not involved in the Jacobian $d/d'$, fortunately, the Jacobian $d/d'$ is not perturbed (as the set of active weights in $d/d'$ are the same when we perturb $\mathcal{W}$ as $\mathcal{W}[+\mathcal{U}_{d'}]$). So, we will only need to bound $y_{d'',h,h'}$ for $d'' > d'$.

What does the distribution of $y_{d'',h,h'}$ look like for $d'' > d'$? We can expand[9] $y_{d'',h,h'}$ as the product of i) the $h$th row of the Jacobian $d/d''$ ii) the perturbation matrix $\mathcal{U}_{d''}$ and iii) the $h'$th column of the Jacobian $d'/d''-1$ for the perturbed network:

---

[9]Again, note that the below succinct formula works even for corner cases like $d'' = d'$ or $d'' = d$.

$$y_{d'',h,h'} = \overbrace{J^{d/d''}(\mathbf{x};\mathcal{W})[h]}^{1 \times H_{d''}} \underbrace{\overbrace{U_{d''}}^{H_{d''} \times H_{d''-1}} \overbrace{J^{d''-1/d'}(\mathbf{x};\mathcal{W}[+\mathcal{U}_{d''-1}])[:,h']}^{H_{d''-1} \times 1}}_{\text{spherical Gaussian}}$$

Conditioned on $\mathcal{U}_{d''-1}$, the second part of this expansion, namely, $U_{d''} J^{d''-1/d'}(\mathbf{x};\mathcal{W}[+\mathcal{U}_{d''-1}])[:,h']$ is a multivariate spherical Gaussian (see Lemma B.2) of the form $\mathcal{N}(0, \sigma^2 \|J^{d''-1/d'}(\mathbf{x};\mathcal{W}[+\mathcal{U}_{d''-1}])[:,h']\|^2 I)$. As a result, conditioned on $\mathcal{U}_{d''-1}$, $y_{d'',h,h'}$ is a univariate Gaussian $\mathcal{N}(0, \sigma^2 \|J^{d/d''}(\mathbf{x};\mathcal{W})[h]\|^2 \|J^{d''-1/d'}(\mathbf{x};\mathcal{W}[+\mathcal{U}_{d''-1}])[:,h']\|^2)$.

Then, by applying a standard Gaussian tail bound we have that with probability $1 - \frac{\hat{\delta}}{D^2 H^2}$ over the draws of $U_{d''}$ conditioned on $\mathcal{U}_{d''-1}$, each of these quantities is bounded as:

$$|y_{d'',h,h'}| \le \sigma \|J^{d/d''}(\mathbf{x};\mathcal{W})[h]\| \|J^{d''-1/d'}(\mathbf{x};\mathcal{W}[+\mathcal{U}_{d''-1}])[:,h']\| \sqrt{2 \ln \frac{D^2 H^2}{\hat{\delta}}} \qquad (13)$$

We simplify the bound on the right hand side a bit further so that it does not involve any Jacobian of layer $d$. Specifically, when $d'' < d$, $\|J^{d/d''}(\mathbf{x};\mathcal{W})[h]\|$ can be written as the product of the spectral norm of the Jacobian $d' - 1/d''$ and the $\ell_2$ norm of the $h$th row of Jacobian $d - 1/d$. Here, the latter can be upper bounded by the $\ell_2$ norm of the $h$th row of $W_d$ since the Jacobian (for a ReLU network) is essentially $W_d$ but with some columns zerod out. When $d = d''$, $\|J^{d/d'}(\mathbf{x};\mathcal{W})[h]\|$ is essentially 1 as the Jacobian is merely the identity matrix. Thus, we have:

$$|y_{d'',h,h'}| \le \begin{cases} \sigma \|\mathbf{w}_h^d\| \|J^{d-1/d''}(\mathbf{x};\mathcal{W})\|_2 \|J^{d''-1/d'}(\mathbf{x};\mathcal{W}[+\mathcal{U}_{d''-1}])[:,h']\| \sqrt{4 \ln \frac{DH}{\hat{\delta}}} & d'' < d \\ \sigma \|J^{d''-1/d'}(\mathbf{x};\mathcal{W}[+\mathcal{U}_{d''-1}])[:,h']\| \sqrt{4 \ln \frac{DH}{\hat{\delta}}} & d'' = d \end{cases}$$

By a union bound on all $d''$, we then get that with probability $1 - \frac{\hat{\delta}}{D}$ over the draws of $\mathcal{U}_d$, we can upper bound Equation 12 as:

$$\max_h \sum_{d''=1}^{d} \sqrt{\sum_{h'} (y_{d'',h,h'})^2} \le \sigma \|J^{d-1/d'}(\mathbf{x};\mathcal{W}[+\mathcal{U}_{d''-1}])\|_F \sqrt{4 \ln \frac{DH}{\hat{\delta}}} +$$

$$\sum_{d''=d'+1}^{d-1} \sigma \max_h \|\mathbf{w}_h^d\| \|J^{d-1/d''}(\mathbf{x};\mathcal{W})\|_2 \|J^{d''-1/d'}(\mathbf{x};\mathcal{W}[+\mathcal{U}_{d''-1}])\|_F \sqrt{4 \ln \frac{DH}{\hat{\delta}}}$$

By again applying a union bound for all $d'$, we get the above bound to hold simultaneously for all $d'$ with probability at least $1 - \hat{\delta}$. Then, by a similar argument as in the case of the perturbation bound on the output of each layer, we get the result of the lemma.

PERTURBATION BOUND ON THE SPECTRAL NORM OF THE JACOBIAN $d/d'$. Again, we split this term and apply triangle equality as follows:

$$\left| \left\|J^{d/d'}(\mathbf{x};\mathcal{W})\right\|_2 - \left\|J^{d/d'}(\mathbf{x};\mathcal{W}[+\mathcal{U}_d])\right\|_2 \right|$$

$$\le \left\|J^{d/d'}(\mathbf{x};\mathcal{W}) - J^{d/d'}(\mathbf{x};\mathcal{W}[+\mathcal{U}_d])\right\|_2$$

$$\le \left\| \sum_{d''=1}^{d} \left( \underbrace{J^{d/d'}(\mathbf{x};\mathcal{W}[+\mathcal{U}_{d''}]) - J^{d/d'}(\mathbf{x};\mathcal{W}[+\mathcal{U}_{d''-1}])}_{:=Y_{d''}} \right) \right\|_2$$

$$\le \sum_{d''=1}^{d} \|Y_{d''}\|_2 \qquad (14)$$

Here, we have defined $Y_{d''}$ to be the matrix that corresponds to the difference in the Jacobian $d/d'$ brought about by perturbaing the the $d''$th weight matrix, given that the first $d'' - 1$ matrices have already been perturbed.

As argued before, under the frozen activation states, when we perturb the weight matrices from $1$ uptil $d'$, since these matrices are not involved in the Jacobian $d/d'$, fortunately, the Jacobian $d/d'$ is not perturbed (as the set of active weights in $d/d'$ are the same when we perturb $\mathcal{W}$ as $\mathcal{W}[+\mathcal{U}_{d'}]$). So, we will only need to bound $Y_{d''}$ for $d'' > d'$.

Recall that we can exapnd $Y_{d''}$ for $d'' > d'$, $y_{d'',h,h'}$ as the product of i) Jacobian $d/d''$ ii) the perturbation matrix $\mathcal{U}_{d''}$ and iii) the Jacobian $d'/d'' - 1$ for the perturbed network[10]:

$$Y_{d''} = \overbrace{J^{d/d''}(\mathbf{x}; \mathcal{W})}^{H_d \times H_{d''}} \underbrace{\overbrace{U_{d''}}^{H_{d''} \times H_{d''-1}} \overbrace{J^{d''-1/d'}(\mathbf{x}; \mathcal{W}[+\mathcal{U}_{d''-1}])}^{H_{d''-1} \times H_{d'}}}_{\text{spherical Gaussian}}$$

Now, the spectral norm of $Y_{d''}$ is at most the products of the spectral norms of each of these three matrices. Using Lemma B.3, the spectral norm of the middle term $U_{d''}$ can be bounded by $\sigma\sqrt{2H \ln \frac{2DH}{\hat{\delta}}}$ with high probability $1 - \frac{\hat{\delta}}{D}$ over the draws of $U_{d''}$. [11]

We will also decompose the spectral norm of the first term so that our final bound does not involve any Jacobian of the $d$th layer. When $d'' = d$, this term has spectral norm $1$ because the Jacobian $d/d$ is essentially the identity matrix. When $d'' < d$, we have that $\left\| J^{d/d''}(\mathbf{x}; \mathcal{W}) \right\|_2 \leq \left\| J^{d/d-1}(\mathbf{x}; \mathcal{W}) \right\|_2 \left\| J^{d-1/d''}(\mathbf{x}; \mathcal{W}) \right\|_2$. Furthermore, since, for a ReLU network, $J^{d/d-1}(\mathbf{x}; \mathcal{W})$ is effectively $W_d$ with some columns zerod out, the spectral norm of the Jacobian is upper bounded by the spectral norm $W_d$.

Putting all these together, we have that with probability $1 - \frac{\hat{\delta}}{D}$ over the draws of $U_{d''}$, the following holds good:

$$\|Y_{d''}\|_2 \leq \begin{cases} \sigma \|W_d\|_2 \left\| J^{d-1/d''}(\mathbf{x}; \mathcal{W}) \right\|_2 \left\| J^{d''-1/d'}(\mathbf{x}; \mathcal{W}[+\mathcal{U}_{d''-1}]) \right\|_2 \sqrt{2H \ln \frac{2DH}{\hat{\delta}}} & d'' < d \\ \sigma \left\| J^{d''-1/d'}(\mathbf{x}; \mathcal{W}[+\mathcal{U}_{d''-1}]) \right\|_2 \sqrt{2H \ln \frac{2DH}{\hat{\delta}}} & d'' = d \end{cases}$$

By a union bound, we then get that with probability $1 - \hat{\delta}$ over the draws of $\mathcal{U}_d$, we can upper bound Equation 14 as:

$$\sum_{d''=1}^{d} \|Y_{d'}\|_2 \leq \sigma \left\| J^{d''-1/d'}(\mathbf{x}; \mathcal{W}[+\mathcal{U}_{d''-1}]) \right\|_2 \sqrt{2H \ln \frac{2DH}{\hat{\delta}}}$$

$$+ \sigma \sum_{d''=d'+1}^{d-1} \|W_d\|_2 \left\| J^{d-1/d''}(\mathbf{x}; \mathcal{W}) \right\|_2 \left\| J^{d''-1/d'}(\mathbf{x}; \mathcal{W}[+\mathcal{U}_{d''-1}]) \right\|_2 \sqrt{2H \ln \frac{2DH}{\hat{\delta}}}$$

Note that the above bound simultaneously holds over all $d'$ (without the application of a union bound). Finally we get the result of the lemma by a similar argument as in the case of the perturbation bound on the output of each layer.

$\square$

---

[10]Again, note that the below succinct formula works even for corner cases like $d'' = d'$ or $d'' = d$.

[11]Although Lemma B.3 applies only to the case where $U_{d''}$ is a $H \times H$ matrix, it can be easily extended to the corner cases when $d'' = 1$ or $d'' = D$. When $d'' = 1$, $U_{d''}$ would be a $H \times N$ matrix, where $H > N$; one could imagine adding more random columns to this matrix, and applying Lemma B.3. Since adding columns does not reduce the spectral norm, the bound on the larger matrix would apply on the original matrix too. A similar argument would apply to $d'' = D$, where the matrix would be $K \times H$.

# F    PROOF FOR THEOREM 4.1

Below, we present our main result for this section, a generalization bound on a class of networks that is based on certain norm bounds on the *training data*. We provide a more intuitive presentation of these bounds after the proof in Appendix F.3.

**Theorem.  4.1** *For any $\delta > 0$, with probability $1 - \delta$ over the draw of samples $S \sim \mathcal{D}^m$, for any $\mathcal{W}$, we have that:*

$$Pr_{(\mathbf{x},y)\sim\mathcal{D}}\left[\mathcal{L}_0(f(\mathbf{x};\mathcal{W}),y)\right] \leq Pr_{(\mathbf{x},y)\sim S}\left[\mathcal{L}_{\gamma_{class}}(f(\mathbf{x};\mathcal{W}),y)\right] +$$

$$+ \mathcal{O}\left(D \cdot \sqrt{\frac{1}{m-1} \cdot \left(2\sum_{d=1}^{D}\|W_d - Z_d\|^2 \frac{1}{(\sigma^\star)^2} + \ln\frac{Dm}{\delta}\right)}\right)$$

*where $\frac{1}{\sigma^\star}$ is $\mathcal{O}\left(\sqrt{H} \cdot \sqrt{\ln\left(DH\sqrt{m}\right)} \times \max\{\mathcal{B}_{layer\text{-}\ell_2}, \mathcal{B}_{preact}, \mathcal{B}_{jac\text{-}row\text{-}\ell_2}, \mathcal{B}_{jac\text{-}spec}, \mathcal{B}_{output}\}\right)$, where:*

$$\mathcal{B}_{layer\text{-}\ell_2} := \mathcal{O}\left(\max_{1\leq d<D} \frac{\sum_{d'=1}^{d} \zeta_{d/d'}^\star \alpha_{d'-1}^\star}{\alpha_d^\star}\right)$$

$$\mathcal{B}_{preact} := \mathcal{O}\left(\max_{1\leq d<D} \frac{\sum_{d'=1}^{d} \zeta_{d/d'}^\star \alpha_{d'-1}^\star}{\sqrt{H}\gamma_d^\star}\right)$$

$$\mathcal{B}_{jac\text{-}row\text{-}\ell_2} := \mathcal{O}\left(\max_{1\leq d<D}\max_{1\leq d'<d\leq D} \frac{\zeta_{d-1/d'}^\star + \|W_d\|_{2,\infty}\sum_{d''=d'+1}^{d-1} \psi_{d-1/d''}^{\star 2}\zeta_{d''-1/d'}^\star}{\zeta_{d/d'}^\star}\right)$$

$$\mathcal{B}_{jac\text{-}spec} := \mathcal{O}\left(\max_{1\leq d<D}\max_{1\leq d'<d\leq D} \frac{\psi_{d-1/d'}^\star + \|W_d\|_{2}\sum_{d''=d'+1}^{d-1} \psi_{d-1/d''}^\star\psi_{d''-1/d'}^\star}{\psi_{d/d'}^\star}\right)$$

$$\mathcal{B}_{output} := \mathcal{O}\left(\frac{\sum_{d=1}^{D} \zeta_{D/d}^\star \alpha_{d-1}^\star}{\sqrt{H}\gamma_{class}}\right)$$

*where,*
$\alpha_d^\star := \max\left(\max_{(\mathbf{x},y)\in S}\left\|f^d(\mathbf{x};\mathcal{W})\right\|, 1\right)$ *is an upper bound on the $\ell_2$ norm of the output of each hidden layer $d = 0, 1, \cdots, D-1$ on the training data. Note that for layer 0, this would correspond to the $\ell_2$ norm of the input.*

$\gamma_d^\star := \min_{(\mathbf{x},y)\in S}\min_h \left|f^d(\mathbf{x};\mathcal{W})[h]\right|$ *is a lower bound on the absolute values of the pre-activations for each layer $d = 1, \cdots, D$ on the training data.*

$\zeta_{d/d'}^\star := \max\left(\max_{(\mathbf{x},y)\in S}\left\|J^{d/d'}(\mathbf{x};\mathcal{W})\right\|_{2,\infty}, 1\right)$ *is an upper bound on the row $\ell_2$ norms of the Jacobian for each layer $d = 1, 2, \cdots, D$, and $d' = 1, \cdots, d$ on the training data.*

$\psi_{d/d'}^\star := \max\left(\max_{(\mathbf{x},y)\in S}\left\|J^{d/d'}(\mathbf{x};\mathcal{W})\right\|_{2}, 1\right)$ *is an upper bound on the spectral norm of the Jacobian for each layer $d = 1, 2, \cdots, D$, and $d' = 1, \cdots, d$ on the training data.*

## F.1    NOTATIONS.

To make the presentation of our proof cleaner, we will set up some notations. First, we use $\mathscr{C}^\star$ to denote the 'set' of constants related to the norm bounds on training set defined in the Theorem above. (Here we use the term set loosely, like we noted in the previous section.) Based on these

training set related constants, we also define $\mathscr{C}^\dagger$ to be the following constants corresponding to weaker norm-bounds related to the *test data*:

1. $\alpha_d^\dagger := \mathbf{2}\alpha_d^\star$, for each hidden layer $d = 0, 1, \cdots, D-1$, (we will use this to bound $\ell_2$ norms of the outputs of the layers of the network on a test input)

2. $\gamma_d^\dagger := \gamma_d^\star / \mathbf{2}$ for each layer $d = 1, \cdots, D$, (we will use this to bound magnitudes of the preactivations values of the network on a test input).

3. $\zeta_{d/d'}^\dagger := \mathbf{2}\zeta_{d/d'}^\star$ for each layer $d = 1, 2, \cdots, D$, and $d' = 1, \cdots, d$ (we will use this to bound $\ell_2$ norms of rows in the Jacobians of the network for a test input)

4. $\psi_{d/d'}^\dagger := \mathbf{2}\psi_{d/d'}^\star$ for each layer $d = 1, 2, \cdots, D$, and $d' = 1, \cdots, d$ (we will use this to bound spectral norms of the Jacobians of the network for a test input)

**Note about (abuse of) notation:** We reiterate a point about our notation which we also made in Appendix E. We call $\mathscr{C}^\star$ and $\mathscr{C}^\dagger$ a 'set' to denote a group of related constants by a single symbol. Each element in this set has a particular semantic associated with it, unlike the standard notation of a set.

Now, for any given set of constants $\mathscr{C}$, for a particular weight configuration $\mathcal{W}$, and for a given input $\mathbf{x}$, we define the following event which holds when the network satisfies certain norm-bounds defined by the constants $\mathscr{C}$ (that are favorable for noise-resilience).

**Definition F.1.** *For a set of constants $\mathscr{C}$, for network parameters $\mathcal{W}$ and for any input $\mathbf{x}$, we define* NORM-BOUND$(\mathcal{W}, \mathscr{C}, \mathbf{x})$ *to be the event that all the following hold good:*

1. *for all $\alpha_d \in \mathscr{C}$, $\left\| f^d(\mathbf{x}; \mathcal{W}) \right\| < \alpha_d$ (Output of the layer does not have too large an $\ell_2$ norm).*

2. *for all $\gamma_d \in \mathscr{C}$, $\min_h \left| f^d(\mathbf{x}; \mathcal{W})[h] \right| > \gamma_d$. (Pre-activation values are not too small).*

3. *for all $\zeta_{d/d'} \in \mathscr{C}$, $\max_h \left\| J^{d/d'}(\mathbf{x}; \mathcal{W})[h] \right\| < \zeta_{d/d'}$ (Rows of Jacobian do not have too large an $\ell_2$ norm).*

4. *for all $\psi_{d/d'} \in \mathscr{C}$, $\left\| J^{d/d'}(\mathbf{x}; \mathcal{W}) \right\|_2 < \psi_{d/d'}$ (Jacobian does not have too large a spectral norm).*

NOTE: (Similar to a note under Definition E.1) A subtle point in the above definition (which we will make use of, to state our theorems) is that if we supply only a subset of $\mathscr{C}$ to the above event, NORM-BOUND$(\mathcal{W}, \cdot, \mathbf{x})$ then it would denote the event that only those subset of properties satsify the respective norm bounds.

## F.2  PROOF FOR THEOREM 4.1

*Proof.* To apply the framework, we will have to first define and order the input-dependent properties $\rho$ and the corresponding margins $\Delta^\star$ used in Theorem 3.1. We will define these properties in terms of the following functions: $\left\| f^d(\mathbf{x}; \mathcal{W}) \right\|$, $f^d(\mathbf{x}; \mathcal{W})[h]$, $\left\| J^{d/d'}(\mathbf{x}; \mathcal{W})[h] \right\|$ and $\left\| J^{d/d'}(\mathbf{x}; \mathcal{W}) \right\|_2$. Following this definition, we will create an ordered grouping of these properties.

**Definition F.2.** *For ReLU networks, we enumerate the input-dependent properties (on the left below) and their corresponding margins (on the right below) denoted with a superscript $\Delta$:*

$$
\begin{aligned}
2\alpha_d^\star - \left\| f^d(\mathbf{x}; \mathcal{W}) \right\| \qquad & \alpha_d^\vartriangle := \alpha_d^\star && \text{for } d = 0, 1, 2, \cdots, D-1 \\
\left| f^d(\mathbf{x}; \mathcal{W})[h] \right| - \tfrac{\gamma_d^\star}{2} \qquad & \gamma_d^\vartriangle := \tfrac{\gamma_d^\star}{2} && \text{for } d = 1, 2, \cdots, D-1 \text{ for all possible } h \\
2\zeta_{d/d'}^\star - \left\| J^{d/d'}(\mathbf{x}; \mathcal{W})[h] \right\| \qquad & \zeta_{d/d'}^\vartriangle := \zeta_{d/d'}^\star && \text{for } d = 1, \cdots, D \ \ d' = 1, \cdots, d-1 \text{ for all possible } h \\
2\psi_{d/d'}^\star - \left\| J^{d/d'}(\mathbf{x}; \mathcal{W}) \right\|_2 \qquad & \psi_{d/d'}^\vartriangle := \psi_{d/d'}^\star && \text{for } d = 1, \cdots, D \ \ d' = 1, \cdots, d-1
\end{aligned}
$$

*and for the output layer $D$:*

$$
f(\mathbf{x}; \mathcal{W})[y] - \max_{j \neq y} f(\mathbf{x}; \mathcal{W})[j] \qquad\qquad \gamma_D^\vartriangle := \gamma_{class}
$$

We will use the notation $\mathscr{C}^{\triangle}$ to denote the sets of all margin terms defined on the right side above, and $\mathscr{C}^{\triangle}/2$ to denote the values in that set divided by 2.

ON THE CHOICE OF THE ABOVE FUNCTIONS AND MARGIN VALUES. Recall that for a specific input-dependent property $\rho$ and its margin $\Delta^{\star}$, the condition in Equation 1 requires that $\rho(\mathcal{W}, \mathbf{x}, y) > \Delta^{\star}$. When we generalize these conditions in Theorem C.1, we will assume that these are satisfied on the training data, and we show that on the test data the approximate version of these conditions, namely $\rho(\mathcal{W}, \mathbf{x}, y) > 0$ hold. Below, we show what these conditions and their approximate versions translate to, in terms of norm-bounds on $\left\| f^d(\mathbf{x}; \mathcal{W}) \right\|$, $f^d(\mathbf{x}; \mathcal{W})[h]$, $\left\| J^{d/d'}(\mathbf{x}; \mathcal{W})[h] \right\|$ and $\left\| J^{d/d'}(\mathbf{x}; \mathcal{W}) \right\|_2$; we encapsulate our statements in the following fact for easy reference later in our proof.

**Fact F.1.** *When $\rho, \Delta^{\star}$ correspond to $2\alpha_d^{\star} - \left\| f^d(\mathbf{x}; \mathcal{W}) \right\|$ and $\alpha_d^{\triangle}$, the conditions translate to upper bounds on the $\ell_2$ norm of the layer as:*

$$\rho(\mathcal{W}, \mathbf{x}, y) > \Delta^{\star} \equiv \left\| f^d(\mathbf{x}; \mathcal{W}) \right\| < \alpha_d^{\star}$$
$$\rho(\mathcal{W}, \mathbf{x}, y) > 0 \equiv \left\| f^d(\mathbf{x}; \mathcal{W}) \right\| < 2\alpha_d^{\star} = \alpha_d^{\dagger}$$

*When $\rho, \Delta^{\star}$ correspond to $\left| f^d(\mathbf{x}; \mathcal{W})[h] \right| - \frac{\gamma_d^{\star}}{2}$ and $\gamma_d^{\triangle} := \frac{\gamma_d^{\star}}{2}$, then the conditions translate to lower bounds on the pre-activation values as:*

$$\rho(\mathcal{W}, \mathbf{x}, y) > \Delta^{\star} \equiv \left| f^d(\mathbf{x}; \mathcal{W})[h] \right| > \gamma_d^{\star}$$
$$\rho(\mathcal{W}, \mathbf{x}, y) > 0 \equiv \left| f^d(\mathbf{x}; \mathcal{W})[h] \right| > \gamma_d^{\star}/2 = \gamma_d^{\dagger}$$

*When $\rho, \Delta^{\star}$ correspond to $2\zeta_{d/d'}^{\star} - \left\| J^{d/d'}(\mathbf{x}; \mathcal{W})[h] \right\|$ and $\zeta_{d/d'}^{\triangle}$, the conditions translate to upper bounds on the row $\ell_2$ norm of the Jacobian as:*

$$\rho(\mathcal{W}, \mathbf{x}, y) > \Delta^{\star} \equiv \left\| J^{d/d'}(\mathbf{x}; \mathcal{W})[h] \right\| < \zeta_{d/d'}^{\star}$$
$$\rho(\mathcal{W}, \mathbf{x}, y) > 0 \equiv \left\| J^{d/d'}(\mathbf{x}; \mathcal{W})[h] \right\| < 2\zeta_{d/d'}^{\star} = \zeta_{d/d'}^{\dagger}$$

*When $\rho, \Delta^{\star}$ correspond to $2\psi_{d/d'}^{\star} - \left\| J^{d/d'}(\mathbf{x}; \mathcal{W}) \right\|_2$ and $\psi_{d/d'}^{\triangle}$, the conditions translate to upper bounds on the spectral norms of the Jacobian as:*

$$\rho(\mathcal{W}, \mathbf{x}, y) > \Delta^{\star} \equiv \left\| J^{d/d'}(\mathbf{x}; \mathcal{W}) \right\|_2 < \psi_{d/d'}^{\star}$$
$$\rho(\mathcal{W}, \mathbf{x}, y) > 0 \equiv \left\| J^{d/d'}(\mathbf{x}; \mathcal{W}) \right\|_2 > 2\psi_{d/d'}^{\star} = \psi_{d/d'}^{\dagger}$$

*When $\rho, \Delta^{\star}$ correspond to $f(\mathbf{x}; \mathcal{W})[y] - \max_{j \neq y} f(\mathbf{x}; \mathcal{W})[j]$ and $\gamma_D^{\triangle}$, the conditions translate to lower bounds on the margin:*

$$\rho(\mathcal{W}, \mathbf{x}, y) > \Delta^{\star} \equiv f(\mathbf{x}; \mathcal{W})[y] - \max_{j \neq y} f(\mathbf{x}; \mathcal{W})[j] > 0$$
$$\rho(\mathcal{W}, \mathbf{x}, y) > 0 \equiv f(\mathbf{x}; \mathcal{W})[y] - \max_{j \neq y} f(\mathbf{x}; \mathcal{W})[j] > \gamma_{class}$$

GROUPING AND ORDERING THE PROPERTIES. Now to apply the abstract generalization bound in Theorem 3.1, recall that we need to come up with an ordered grouping of the functions above such that we can realize the constraint given in Equation 2. Specifically, this constraint effectively required that, for a given input, the perturbation in the properties grouped in a particular set be small, given that all the properties in the preceding sets satisfy the corresponding conditions on them. To this end, we make use of Lemma E.1 where we have proven perturbation bounds relevant to the properties we have defined above. Our lemma also naturally induces dependencies between these properties in a way that they can be ordered as required by our framework.

The order in which we traverse the properties is as follows, as dictated by Lemma E.1. We will go from layer 0 uptil $D$. For a particular layer $d$, we will first group the properties corresponding to the

spectral norms of the Jacobians of that layer whose corresponding margins are $\{\psi_{d/d'}^{\triangle}\}_{d'=1}^{d}$. Next, we will group the row $\ell_2$ norms of the Jacobians of layer $d$, whose corresponding margins are $\{\zeta_{d/d'}^{\triangle}\}_{d'=1}^{d}$. Followed by this, we will have a singleton set of the layer output's $\ell_2$ norm whose corresponding margin is $\alpha_d^{\triangle}$. We then will group the pre-activations of layer $d$, each of which has the corresponding margin $\gamma_d^{\triangle}$. For the output layer, instead of the pre-activations or the output $\ell_2$ norm, we will consider the margin-based property we have defined above. [12] [13] Observe that the number of sets $R$ that we have created in this manner, is at most $4D$ since there are at most $4$ sets of properties in each layer.

PROVING CONSTRAINT IN EQUATION 2. Recall the constraint in Equation 2 that is required by our framework. For any $r$, the $r$th set of properties need to satisfy the following statement:

if $\forall q < r, \forall l, \rho_{q,l}(\mathcal{W}, \mathbf{x}, y) > 0$ then

$$Pr_{\mathcal{U} \sim \mathcal{N}(0, \sigma^2 I)}\Big[ \forall l \ |\rho_{r,l}(\mathcal{W} + \mathcal{U}, \mathbf{x}, y) - \rho_{r,l}(\mathcal{W}, \mathbf{x}, y)| > \frac{\Delta_{r,l}(\sigma)}{2} \quad \text{and}$$

$$\forall q < r, \forall l \ |\rho_{q,l}(\mathcal{W} + \mathcal{U}, \mathbf{x}, y) - \rho_{q,l}(\mathcal{W}, \mathbf{x}, y)| < \frac{\Delta_{q,l}(\sigma)}{2} \Big] \leq \frac{1}{(R+1)\sqrt{m}}. \quad (2)$$

Furthermore, we want the perturbation bounds $\Delta_{r,l}(\sigma)$ to satisfy $\Delta_{r,l}(\sigma^\star) \leq \Delta_{r,l}^\star$, where $\sigma^\star$ is the standard deviation of the parameter perturbation chosen in the PAC-Bayesian analysis.

The next step in our proof is to show that our choice of $\sigma^\star$, and the input-dependent properties, all satisfy the above requirements. To do this, we instantiate Lemma E.1 with $\sigma = \sigma^\star$ as in Theorem 4.1 (choosing appropriate constants), $\hat{\delta} = \frac{1}{4D\sqrt{m}}$ and $\hat{\mathscr{C}} = \mathscr{C}^{\triangle}/2$. Then, it can be verified that the values of the perturbation bounds in $\hat{\mathscr{C}}'$ in Lemma E.1 can be upper bounded by the corresponding value in $\mathscr{C}^{\triangle}/2$. In other words, we have that for our chosen value of $\sigma$, the perturbations in all the properties and the output of the network can be bounded by the constants specified in $\mathscr{C}^{\triangle}/2$. Succinctly, let us say:

$$\hat{\mathscr{C}}' \leq \mathscr{C}^{\triangle}/2 \quad (15)$$

Given that these perturbation bounds hold for our chosen value of $\sigma$, we will focus on showing that a constraint of the form Equation 2 holds for the row $\ell_2$ norms of the Jacobians $d/d'$ for all $d' < d$. A similar approach would apply for the other properties.

First, we note that the sets of properties preceding the ones corresponding to the row $\ell_2$ norms of Jacobian $d/d'$, consists of all the properties upto layer $d - 1$. Therefore, the precondition for Equation 2 which is of the form $\rho(\mathcal{W}, \mathbf{x}, y) > 0$ for all the previous properties $\rho$, translates to norm bound on these properties involving the constants $\mathscr{C}_{d-1}^{\dagger}$ as discussed in Fact F.1. Succinctly, these norm bounds can be expressed as NORM-BOUND$(\mathcal{W} + \mathcal{U}, \mathscr{C}_{d-1}^{\dagger}, \mathbf{x})$.

Given that these norm bounds hold for a particular $\mathbf{x}$, our goal is to argue that the rest of the constraint in Equation 2 holds. To do this, we first argue that given these norm bounds, if PERT-BOUND$(\mathcal{W} + \mathcal{U}, \mathscr{C}_{d-1}^{\triangle}/2, \mathbf{x})$ holds, then so does UNCHANGED-ACTS$_{d-1}(\mathcal{W} + \mathcal{U}, \mathbf{x})$. This is because, the event PERT-BOUND$(\mathcal{W} + \mathcal{U}, \mathscr{C}_{d-1}^{\triangle}/2, \mathbf{x})$ implies that the pre-activation values of layer $d - 1$ suffer a perturbation of at most $\gamma_{d-1}^{\triangle}/2 = \gamma_{d-1}^\star/4$ i.e., $\max_h |f^{d-1}(\mathbf{x}; \mathcal{W} + \mathcal{U})[h] - f^{d-1}(\mathbf{x}; \mathcal{W})[h]| \leq \gamma_{d-1}^\star/4$. However, since NORM-BOUND$(\mathcal{W}, \mathscr{C}_{d-1}^{\dagger}, \mathbf{x})$ holds, we have that the preactivation values of this layer have a magnitude of at least $\gamma_{d-1}^{\dagger} = \gamma_{d-1}^\star/2$ before perturbation i.e., $\min_h |f^{d-1}(\mathbf{x}; \mathcal{W})[h]| \geq \gamma_{d-1}^\star/2$. From these two equations, we have that the hidden units even at layer $d-1$ of the network do not change their activation state (i.e., the sign of the pre-activation does not change) under this perturbation. We can similarly argue for the layers below $d - 1$, thus proving that UNCHANGED-ACTS$_{d-1}(\mathcal{W} + \mathcal{U}, \mathbf{x})$ holds under PERT-BOUND$(\mathcal{W} + \mathcal{U}, \mathscr{C}_{d-1}^{\triangle}/2, \mathbf{x})$.

Then, from the above discussion on the activation states, and from Equation 15, we have that Lemma E.1 boils down to the following inequality, when we plug $\sigma = \sigma^\star$:

---

[12] For layer 0, the only property that we have defined is the $\ell_2$ norm of the input.

[13] Note that the Jacobian for $d/d$ is nothing but an identity matrix regardless of the input datapoint; thus we do not need any generalization analysis to bound its value on a test datapoint. Hence, we ignore it in our analysis, as can be seen from the list of properties that we have defined.

$$Pr_{\mathcal{U}}\Big[\neg\textsc{Pert-Bound}(\mathcal{W}+\mathcal{U}, \{\zeta_{d/d'}^{\triangle}/2\}_{d'=1}^{d-1}, \mathbf{x}) \wedge \textsc{Pert-Bound}(\mathcal{W}+\mathcal{U}, \mathscr{C}_{d-1}^{\triangle}/2, \mathbf{x})\Big] \leq \frac{1}{4D\sqrt{m}}$$

First note that this inequality has the same form as the constraint required by Equation 2 in our framework. Specifically, in place of the generic perturbation bound $\Delta(\sigma^\star)$, we have $\zeta_{d/d'}^{\triangle}$. Furthermore, recall that our abstract generalization theorem in Theorem 3.1 required that the perturbation bound $\Delta_{r,l}(\sigma^\star)$ be smaller than the corresponding margin $\Delta_{r,l}^\star$. Since the margin here is $\zeta_{d/d'}^{\triangle}$, this is indeed the case. Through identical arguments for the other sets of input-dependent properties that we have defined, we can similarly show how the constraint in Equation 2 holds.

Thus, the input-dependent properties we have devised satisfy all the requirements of our framework, allowing us to apply Theorem 3.1, with $R \leq 4D$. Here, we use a prior centered at the random initialization $\mathcal{Z}$; Lemma B.4 helps simplify the KL-divergence term between the posterior centered at $\mathcal{W}$ and the prior at the random initialization $\mathcal{Z}$.

COVERING ARGUMENT.   To complete our proof, we need to take one more final step. First note that the guarantee in Theorem 3.1 requires that both $\sigma^\star$ and the margin constants $\Delta_{r,l}^\star$ in Equation 1 are all chosen before drawing the training dataset. Thus, to apply this bound in practice, one would have to train the network on multiple independent draws of the training dataset (roughly $\mathcal{O}(1/\delta)$ many draws), and then compute norm-bounds on the input-dependent properties across all these runs, and then choose the largest $\sigma^\star$ based on all these norm-bounds. We emphasize that theoretically speaking, this sort of a bound is still a valid generalization bound that essentially applies to a restricted, norm-bounded class of neural networks. Indeed, the hope is that the implicit bias of stochastic gradient descent ensures that the networks it learns do satisfy these norm-bounds on the input-dependent properties across most draws of the training dataset.

But for practical purposes, one may not be able to empirically determine norm-bounds that hold on $1 - \delta$ of the training set draws, and one might want to get a generalization bound based on norm-bounds that hold on just a single draw. We take this final step in our proof in order to derive such a generalization bound. We do this via the standard theoretical trick of 'covering' the space of all possible norm-bounds. That is, consider the set of $\leq 4D^2$ different constants in $\mathscr{C}^\star$ (that bound the different norms), based on which we choose $\sigma^\star$. We will create a 'grid' of constants (independent of the training data) such that for any particular run of the algorithm, we can find a point on this grid (that corresponds to a configuration of the constants) for which the norm-bounds still hold for that run. These bounds will be looser, but only by a constant multiplicative factor. This will ensure that the bound resulting from choosing $\sigma^\star$ based on this point on the grid, is only a constant factor looser than choosing $\sigma^\star$ based on the actual norm-bounds for that training set. Then, we will instantiate Theorem 3.1 for all the points on the grid, and apply a union bound over all of these to get our final bound.

We create the grid based as follows. Observe that the bound we get from $\mathcal{B}_{\text{output}}$ is at least as large as $\alpha_{d-1}^\star/(\sqrt{H}\gamma_{\text{class}})$. Then, for any value of $\alpha_{d-1}^\star = \Omega\left(\sqrt{H}\gamma_{\text{class}}\sqrt{m}\right)$, we will choose a value of $1/\sigma^\star$ that is $\Omega\left(\sqrt{m}\right)$, rendering the final bound vacuous. Also note that $\alpha_{d-1}^\star \geq 1$. Thus, we will focus on the interval $[1, \mathcal{O}\left(\sqrt{H}\gamma_{\text{class}}\sqrt{m}\right)]$, and grid it based on the points $1, 2, 4, 8, \cdots, \mathcal{O}\left(\sqrt{H}\gamma_{\text{class}}\sqrt{m}\right)$. Observe that any value of $\alpha_{d-1}^\star$ can be approximated by one of these points within a multiplicative factor of 2. Furthermore, this gives rise to at most $\mathcal{O}\left(\log_2 \sqrt{H}\gamma_{\text{class}}\sqrt{m}\right)$ many points on this grid. Next, for a given point on this grid, by examining $\mathcal{B}_{\text{layer-}\ell_2}$ and $\mathcal{B}_{\text{output}}$, we can similarly argue how the range of values of $\zeta_{d/d'}^\star$ is limited between 1 and a polynomial in terms of $H$ and $\gamma_{\text{class}}$; this range of values can similarly be split into a grid. Then, by examining $\mathcal{B}_{\text{preact}}$, we can arrive at a similar grid for the quantity $1/\gamma_d^\star$; by examining $\mathcal{B}_{\text{jac-row-}\ell_2}$, we can get a grid for $\psi_{d/d'}^\star$ too. In this manner, we can grid the space of all possible configurations of the constants into at most $(poly(H, D, m, \gamma_{\text{class}}))^{4D^2}$ many points (since there are not more than $4D^2$ different constants).

For any given run, we can pick a point from this grid such that the norm-bounds are loose only by a constant multiplicative factor. Finally, we apply Theorem 3.1 for each of these grids by setting the failure probability to be $\delta/(poly(H, D, m, \gamma_{\text{class}}))^{4D^2}$, and then combine them via a union bound.

Note, that the resulting bound would have a $\sqrt{\log \frac{(poly(H,D,m,\gamma_{\text{class}}))^{4D^2}}{\delta}}/\sqrt{m}$ term, that would only result in a $\sqrt{\frac{D^2 \log poly(H,D,m,\gamma_{\text{class}})}{m}}$ term that does not affect our bound in an asymptotic sense.

$\square$

### F.3 ADDITIONAL EXPERIMENTS.

In this section, we provide more detailed demonstration of the dependence of the terms in our bound on the depth/width of the network. In all the experiments, including the ones in the main paper (except the one in Figure 2 (b)) we use SGD with learning rate $0.1$ and mini-batch size $64$. We train the network on a subset of $4096$ random training examples from the MNIST dataset to minimize cross entropy loss. We stop training when we classify at least $0.99$ of the data perfectly, with a margin of $\gamma_{\text{class}} = 10$. In Figure 2 (b) where we train networks of depth $D = 28$, the above training algorithm is quite unstable. Instead, we use Adam with a learning rate of $10^{-5}$ until the network achieves an accuracy of $0.95$ on the training dataset. Finally, we note that all logarithmic transformations in our plots are to the base $10$.

In Figure 3 we show how the norm-bounds on the input-dependent properties of the network do not scale as large as the product of spectral norms.

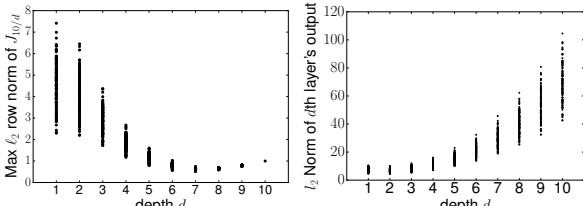

Figure 3: In all these plots, we train a network with $D = 11$, $H = 1280$. **Left**: Each black point corresponds to the maximum row $\ell_2$ norm of the Jacobian $10/d$. Observe that for any $d$, these quantities are nowhere near as large as a naive upper bound that would roughly scale as $\prod_{d'=d}^{10} \|W_d\|_2 = 2^{10-d}$. **Right**: Each black point corresponds to a particular training example $\mathbf{x}$, and has $y$-value equal to the $\ell_2$ norm of the output of layer $d$ for that datapoint. A naive upper bound on this value would be $\|\mathbf{x}\| \prod_{d'=1}^{d} \|W_d\|_2 \approx 10 \cdot 2^d$, which would be at least $100$ times larger than the observed value for $d = 10$.

For the remaining experiments in this section, we will present a slightly looser bound than the one presented in our main result, motivated by the fact that computing our actual bound is expensive as it involves computing spectral norms of $\Theta(D^2)$ Jacobians on $m$ training datapoints. We note that even this looser bound does not have a dependence on the product of spectral norms, and has similar overall dependence on the depth.

Specifically, we will consider a bound that is based on a slightly modified noise-resilience analysis. Recall that in Lemma E.1, when we considered the perturbation in the row $\ell_2$ norm Jacobian $d/d'$, we bounded Equation 13 in terms of the spectral norms of the Jacobians. Instead of taking this route, if we retained the bound in Equation 13, we will get a slightly different upper bound on the perturbation of the Jacobian row $\ell_2$ norm as:

$$\hat{\zeta}'_{d/d'} := \sigma \sum_{d''=d'+1}^{d} \left\| J^{d/d''}(\mathbf{x}; \mathcal{W}) \right\|_F \left( \left\| J^{d''-1/d'}(\mathbf{x}; \mathcal{W}) \right\|_F + \hat{\zeta}_{d''-1/d'} \sqrt{H} \right) \sqrt{2 \ln \frac{D^2 H^2}{\hat{\delta}}}$$

By using this bound in our analysis, we can ignore the spectral norm terms $\psi^\star_{d/d'}$ and derive a generalization bound that does not involve these terms. However, we would now have $\mathcal{O}(D^2)$ conditions instead of $\mathcal{O}(D)$. This is because, the perturbation bound for the row norms of Jacobian $d/d'$ now depends on the row norms of Jacobian $d/d''$, for all $d'' > d'$. Thus, the row $\ell_2$ norms of these Jacobians must be split into separate sets of properties, and the bound on them generalized one after the other (instead of grouped into one set and generalized all at one go as before). This would

give us a similar generalization bound that is looser by a factor of $D$, does not involve $\mathcal{B}_{\text{jac-spec}}$, and where $\mathcal{B}_{\text{jac-row-}\ell_2}$ is redefined as:

$$\mathcal{B}_{\text{jac-row-}\ell_2} := \mathcal{O}\left(\max_{1 \le d < D} \max_{1 \le d' < d \le D} \frac{\sum_{d''=d'+1}^{d} \zeta^{\star}_{d-1/d''} \zeta^{\star}_{d''-1/d'}}{\zeta^{\star}_{d/d'}}\right)$$

All other terms remain the same. In the rest of the discussion, we plot this generalization bound that is looser by a $D$ factor, but still does not depend on the product of the spectral norms. In Figure 4 we show how the quantities in this bound and the bound itself varies with depth, for a network of $H = 1280$, wider than what we considered in Figure 1.

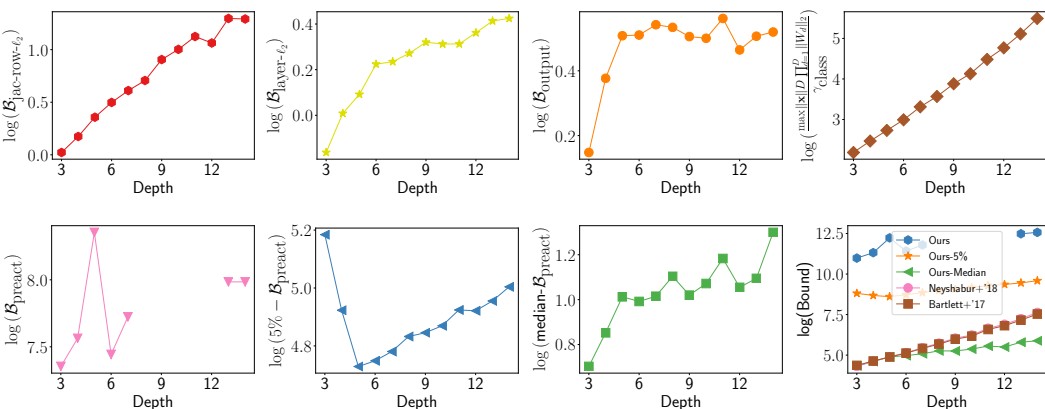

Figure 4: In the above figure, we plot the logarithm (to the base 10) of values of the terms in our bound and in existing bounds for $H = 1280$. Again, we observe that $\mathcal{B}_{\text{jac-row-}\ell_2}, \mathcal{B}_{\text{layer-}\ell_2}, \mathcal{B}_{\text{output}}$ typically lie in the range of $[10^0, 10^2]$. In contrast, the equivalent term from Neyshabur et al. (2018) consisting of the product of spectral norms can be as large as $10^5$ for $D = 10$. Unfortunately, for large $H$, due to numerical precision issues, the smallest pre-activation value is rounded off to zero and hence $\mathcal{B}_{\text{preact}}$ becomes undefined in such situations. However, as noted before, the hypothetical variations $5\%$-$\mathcal{B}_{\text{preact}}$ and median-$\mathcal{B}_{\text{preact}}$ are bounded better and achieve significantly smaller values. Finally, observe that our overall bound and all its hypothetical variations have a smaller slope than previous bounds.

In Figure 5 and Figure 6 we show log-log (note that here even the $x$-axis has been transformed logarithmically) plots of all the quantities for networks of varying width and $D = 8$ and $D = 14$ respectively. Here, we observe that $\mathcal{B}_{\text{jac-row-}\ell_2}$ is width-independent. On the other hand $\mathcal{B}_{\text{layer-}\ell_2}$ and the product-of-spectral-norm term mildly decrease with width; $\mathcal{B}_{\text{output}}$ *decreases* with width at the rate of $1/\sqrt{H}$.

As far as the term $\mathcal{B}_{\text{preact}}$ is concerned, recall from our discussion in the main paper that the minimum pre-activation value $\gamma^{\star}_d$ of the network tends to be quite small in practice (and can be rounded to zero due to precision issues). Therefore the term $\mathcal{B}_{\text{preact}}$ can be arbitrarily large and exhibit considerable variance across different widths/depths and different training runs. On the other hand, interestingly, the hypothetical variation median-$\mathcal{B}_{\text{preact}}$ *decreases* with width at the rate of $1/\sqrt{H}$, while $5\%$-$\mathcal{B}_{\text{preact}}$ increases with a $\sqrt{H}$ dependence on width.

Theoretically speaking, as far as the width-dependence is concerned, the best-case scenario for $\mathcal{B}_{\text{preact}}$ can be realized when the preactivation values of each layer (which has a total $\ell_2$ norm that is width-independent in practice) are equally spread out across the hidden units. Then we will have that the smallest pre-activation value to be as large as $\Omega(1/\sqrt{H})$.

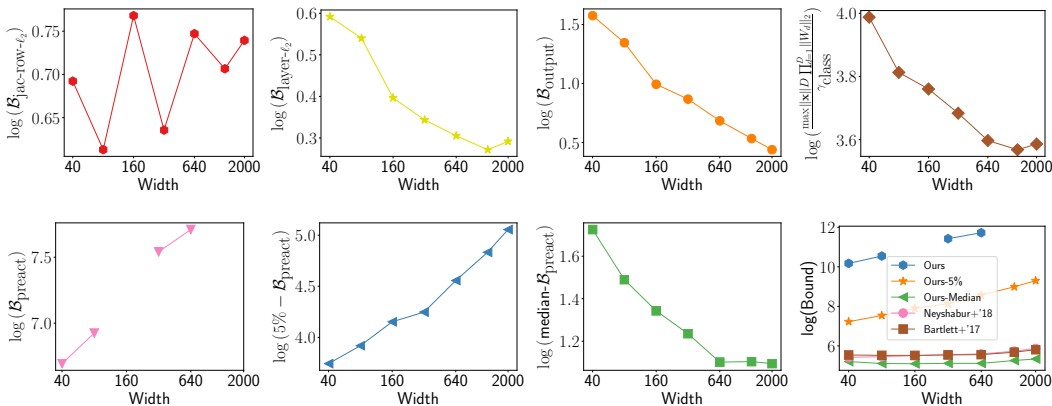

Figure 5: Log-log plots of various terms for $D = 8$ and varying $H$. Note that if the slope of the $\log y$ vs $\log H$ plot is $c$, then $y \propto H^c$.

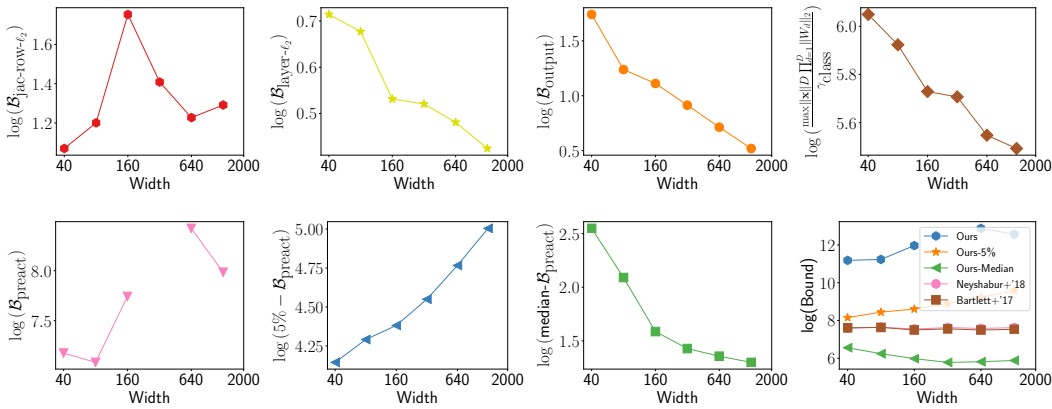

Figure 6: Log-log plots of various terms for $D = 14$ and varying $H$.

# G COMPARISON OF OUR NOISE-RESILIENCE CONDITIONS WITH RELATED WORK

Recall from the discussion in the introduction in the main paper that, prior works (Neyshabur et al., 2017; Arora et al., 2018) have also characterized noise resilience in terms of conditions on the interactions between the activated weight matrices. Below, we discuss the conditions assumed by these works, which parallel the conditions we have studied in our paper (such as the bounded $\ell_2$ norm in each layer).

There are two main high level similarities between the conditions studied across these works. First, these conditions – all of which characterize the interactions between the activated weights matrices in the network – are assumed only for the training inputs; such an assumption implies noise-resilience of the network on training inputs. Second, there are two kinds of conditions assumed. The first kind allows one to bound the propagation of noise through the network under the assumption that the activation states do not flip; the second kind allows one to bound the extent to which the activation states do flip.

CONDITIONS IN NEYSHABUR ET AL. (2017)  Using noise-resilience conditions assumed about the network on the training data, Neyshabur et al. (2017) derive a PAC-Bayes based generalization bound on a stochastic network. The first condition in Neyshabur et al. (2017) characterizes how the Jacobians of different parts of the network interact with each other. Specifically, consider layers $d, d'$ and $d''$ such that $d'' \le d' \le d$. Then, consider the Jacobian of layer $d'$ with respect to layer $d''$ and the

Jacobian of layer $d$ with respect to $d'$. Then, they require that $\left\|J^{d/d'}(\mathbf{x};\mathcal{W})\right\|_F \left\|J^{d'-1/d''}(\mathbf{x};\mathcal{W})\right\|_F = \mathcal{O}(\left\|J^{d/d''}(\mathbf{x};\mathcal{W})\right\|_F)$. This specific condition allows one to bound how the noise injected into the parameters propagate through the network under the assumption that the activation states do not flip. In our paper, we pick an orthogonal approach by assuming an upper bound on the Jacobian $\ell_2$ norms and the layer output norms, which allows us to bound the propagation of noise under unchanged activation states.

The second condition in Neyshabur et al. (2017) is that under a noise of variance $\sigma^2$, the number of units that flip their activation state in a particular layer must be bounded as $\mathcal{O}(H\sigma)$ i.e., smaller the noise, the smaller the proportion of units that flip their activation state. This condition is similar to (although milder than) our lower bounds on the magnitudes of the pre-activation values (which allow us to pick a sufficiently large noise that does not flip the activation states).

Note that a bound on the Jacobian norms corresponds to a bound on the weights input to the active units in the network. However, since Neyshabur et al. (2017) allow a few units to flip activation states, they additionally require a bound on the weights input to the inactive units too. Specifically, for every layer, the maximum row $\ell_2$ norm of the weight matrix $W_i$ is upper bounded in terms of the Frobenius norm of the Jacobian $J^{d/d-1}(\mathbf{x};\mathcal{W})$.

CONDITIONS IN ARORA ET AL. (2018)    In contrast to our work and Neyshabur et al. (2017), Arora et al. (2018) use their assumed noise-resilience conditions to derive a bound on a compressed network. Another small technical difference here is that, the kind of noise analysed here is Gaussian noise injected into the activations of each layer of the network (and not exactly the weights).

The first condition here characterizes the interaction between the Jacobian of layer $d$ with respect to $d'$ and the output of layer $d'$. Specifically, this is a lower bound on the so-called 'interlayer cushion', which is evaluated as

$$\frac{\left\|J^{d/d'}(\mathbf{x};\mathcal{W})f^{d'}(\mathbf{x};\mathcal{W})\right\|}{\left\|J^{d/d'}(\mathbf{x};\mathcal{W})\right\|_F \left\|f^{d'}(\mathbf{x};\mathcal{W})\right\|}$$

Essentially when the interlayer cushion is sufficiently large, it means that the output of layer $d'$ is well-aligned with the larger singular directions of the Jacobian matrix above it; as a result it can be shown that noise injected at/below layer $d'$ diminishes as it propagates through the weights above layer $d'$, assuming the activation states do not flip. Again, our analysis is technically orthogonal to this style of analysis as we bound the propogation of the noise under unchanged activation states assuming that the norms of the Jacobians and the layer outputs are bounded.

Another important condition in Arora et al. (2018) is that of "interlayer smoothness" which effectively captures how far the set of activation states between two layers, say $d'$ and $d$, flip under noise. Roughly speaking, the assumption made here is that when noise is injected into layer $d'$, there is not much difference between a) the output of the $d$th layer with the activation states of the units in layers $d'$ until $d$ frozen at their original state and b) the output of the $d$th layer with the activation states of the units in layers $d'$ to $d$ allowed to flip under the noise. As stated before, this condition is a relaxed version of our condition that essentially implies that none of the activation states flip.

