# OpenReview forum: "Deterministic PAC-Bayesian generalization bounds for deep networks via generalizing noise-resilience"
_ICLR.cc/2019/Conference_

### Official Review · AnonReviewer3 · 2018-11-03
**PAC-Bayesian generalization bounds of deep neural networks based on the noise-resilience analysis**

**Rating:** 5
**Confidence:** 4

**Review:**

The authors demonstrate the generalization bound for deep neural networks using the PAC-Bayesian approach. They adopt the idea of noise resilience in the analysis and obtain a result that has improved dependence in terms of the network dimensions, but involves parameters (e.g., pre-activation) that may be large potentially.

My major concern is also regarding the dependence on the pre-activation that can be very large in practice. This is also shown in the numerical experiments. Therefore, the overall generalization bound can be larger than existing results, though the later have stronger dependence on the network sizes. By examining the analysis for the main result, it seems to me that the reason the authors can induce weaker dependence on network sizes is essentially they involved the pre-activation parameters. This can be viewed as a trade-off how strong the generalization bound depend on the network sizes and other related parameters (like the pre-activation here) rather than strictly tighten the error bound from a more refined/structured way. I also suggest that the authors provide the comparison of their bound and existing ones to see the quantitative difference of the results.

Regarding the noise resilience, it is not clear to where the noise resilience shows up from the analysis or the result. From the proof of the main result, the analysis seems to be standard as in the PAC-Bayesian analysis, which is based on bounding the difference of the network before and after injecting randomness into the parameters. The difference with respect to the previous result due to the different way of bounding such a gap, where the Jacobian, the pre-activation and function output pop up. But this does not explain how well a network can tolerate the noise, either in the parameter space of the data space. This is different with the previous analysis based on the noise resilience, such as [1]. So, the title and the way the authors explain as noise resilience is somewhat misleading. More detailed explanation will help.

[1] Arora et al. Stronger generalization bounds for deep nets via a compression approach.

---

> ### Author Response · Authors · 2018-11-08
> **Our contribution is a new refined/structured way to "generalize noise-resilience", not to explain noise-resilience**
>
> Thanks for you comments! In this response, we'll address the second half of your comment and explain the contributions of the paper, which we believe has been misunderstood.
>
> We first note that our contribution is not just about getting rid of the dependence on the products of the spectral norms of the weight matrices; our contribution is also that we arrive at such a bound on the *original network* and not just a compressed network/stochastic network. While compression-based bounds  like [1] or other PAC-Bayes based bounds like  [2,3]  numerically evaluate to smaller values, and provide a partial answer for why deep networks generalize well, these bounds are not on the original network learned by SGD.  An extremely important and **non-trivial** piece of the puzzle is to extend the benefits of these bounds (or at least some of its benefits -- in this case the lack of a product-of-spectral-norm dependence) over to the original network.
>
>
> We do this by presenting a structured and novel technique which "generalizes noise-resilience" presented in Section 3. Thus we disagree with the observation that our bound does not "strictly tighten the error bound from a more refined/structured way." Below we describe what we mean by "generalizing noise-resilience", in effect justifying our title, and also clarifying what exactly our contribution is.
>
> Like in [1,2], we model noise-resilience in terms of certain "conditions". For example, [1] assume  conditions like "the interlayer smoothness of the network is sufficiently large on training data". We assume similar conditions (e.g., "the output of each layer has small l2 norm on the training data") this allows us to bound the output perturbation of the network without incurring a product-of-spectral-norm dependence. Crucially, our theory and the theory in [1,2] assume these conditions to hold only **on training data**.
>
> With reference to your comment:
>     "The difference with ... previous result due to the different way of bounding such a gap... But this does not explain how well a network can tolerate the noise":
>
>  While there are technical differences in how these conditions are formulated in [1,2] vs. our work, and how the perturbation in the output is bounded in terms of these conditions, the exact formulation of the conditions is NOT our key contribution. As mentioned in Page 3 under our contributions, our conditions are in fact philosophically similar to those in [1] and [2] and at a high level essentially characterize how the activated parts of the weight matrices in the network interact with each other. We strongly emphasize the following points:
>
>
> =====> The novelty in our paper is NOT primarily about explaining why a network is noise-resilient (on training data).
>
> =====> Our main contribution, when compared to [1] or [2], is that we take a step beyond these existing approaches and present an approach to how conditions assumed about the network on the training data *can be generalized to test data*.  This step is crucial and allows us to claim that the network is noise-resilient on test data as well.
>
>
>  The key reason [1,2] were not able to present product-of-spectral-norm independent bounds on the original network (but only on a modified network) was that they did not generalize these conditions about the behavior of the network from the training data to test data.
>
> To achieve this, we present a structured approach that iterates through the layers and generalizes these conditions one after the other, in a specific order. It requires a lot of care to not incur product-of-spectral-norm dependency (or other extra dependencies on the width) while generalizing any of these multiple O(depth^2) conditions.  Besides, to generalize each condition, we require a particular style of reducing PAC-Bayesian bounds to deterministic bounds. Overall, we hope you understand that our analysis is quite far from "standard as in the PAC-Bayesian analysis, which is based on bounding the difference of the network before and after injecting randomness into the parameters".
>
> The idea of generalizing these conditions is novel and is an important step to explain the noise-resilience of these networks on testing data. Besides being refined and structured, most importantly, our approach is general and leaves scope for future work to use it as a hammer on different sets of conditions (hopefully one that doesn't assume large preactivation values on all units!).
>
>
> We hope our detailed response better explains the contribution of our work to answering the generalization puzzle,  in the context of the results in [1,2].
>
> [1] Arora et al., "Stronger generalization bounds for deep nets via a compression approach."
> [2] Neyshabur et al., "Exploring gen- eralization in deep learning."
> [3] Dziugaite et al., "Computing nonvacuous generalization bounds ... than training data."

---

> ### Author Response · Authors · 2018-11-08
> **Dependence on pre-activation values is necessary to some extent**
>
> We provide some context as to why the dependence on pre-activation values is not outrageous, and is to some extent necessary:
> 	a) Here's our intuition: the larger the pre-activation values, the less likely is it that, under parameter perturbations, the hidden units jump the non-linearity in the ReLU; in other words, the network is more likely to behave "linearly" under small perturbations. Roughly speaking, the more locally linear the network is, the simpler is the fit that the network has found, and hence better the generalization.
> 	b) The assumption that only a small proportion of the hidden units  do not jump the non-linearity under perturbations has been made in prior works e.g., "Interlayer Smoothness" in [1] or condition C2 in [2], and *these have been verified in practice*. Overall, it is intuitively reasonable  that a generalization bound depends on a quantity that characterizes this behavior. Currently, for our bound to be small, one would need that none of the hidden units jump the non-linearity, which as we admitted in the paper, does not reflect reality completely.  Since our framework is quite general, with an even more careful analysis, in the future, one might be able to apply our framework for the case where this assumption is relaxed to better reflect reality (i.e., all but a small proportion of hidden units have a sufficiently large pre-activation value).

---

> ### Author Response · Authors · 2018-11-15
> **Uploaded revision**
>
> Hi! Based on Reviewer 1's feedback, we uploaded a revision with Appendix G that now describes and compares the noise-resilience conditions assumed in our work vs. the ones assumed in prior work. We believe that in addition to our earlier responses to your review, this section might better highlight how noise-resilience is studied in our paper.
>
> Overall, we hope our comments
> i) clarify the main contribution of this paper, which lies in showing how noise-resilience of the network generalizes from training data to test data.
> ii) convince you that our analysis is not a standard application of PAC-Bayes theorems (and is on the contrary, quite nuanced and novel)
> iii) justify the title.
>
> We are eager to know if you have any questions remaining; if your concerns have been clarified, we sincerely hope it helps you re-evaluate our paper and update your score.

---

> ### Author Response · Authors · 2018-11-19
> **Added plots of the bounds -- our bound works better for larger D, small H**
>
> Dear Reviewer,
>
> We want to let you know that, like you've suggested, we've added Figure 2 in the main paper, demonstrating the value of our bound for different values of D, for H=40. We want to highlight that our bound has weaker dependence on depth and does better than other product-of-spectral-norm-based bounds for sufficiently deep, not-so-wide networks. We hope this helps you better appreciate the contribution and significance of our work.

---

> ### Comment · AnonReviewer3 · 2018-11-22
> **how to compare this bound with previous results**
>
> Thanks for the authors' update.
>
> I still do not quite understand what benefit this new result provides compared with existing ones. For example, Neyshabur’18 and Bartlett’17 have the bound of the order (spectral norm product)*sqrt(D^3 H rank/m) by ignoring log factors, and Arora'18 has the bound of the order (max function output)*sqrt(D^3 H^2 /m). This paper (Theorem 4.1) has the bound of the order sqrt(D^7 H max(1/(H pre-act^2), max Jacobian norm)/m). It seems that the order 1/pre-act can be even significantly larger than the spectral norm product and the max function output, which leads to an overall larger bound than existing ones. The empirical result of Arora'18 is not provided, which should be a lot better than Neyshabur’18 and Bartlett’17, hence the proposed bound as well. Moreover, the poly(depth, width) dependence is also stronger than the existing ones. I do not think using the 5% and median pre-act values are fair comparisons with other bounds, which could have been tighter as well if they also use analogous worst-case exempted results.
>
> The analysis of the PAC-Bayes result in terms of the original function (Theorem 3.1) might be of independent interest here. But since the derived result for network functions is worse than existing ones (the dependence on depth/width and pre-act parameters), I do not see their significance in better understanding the generalization performance of neural nets here.

---

> > ### Author Response · Authors · 2018-11-22
> > **a) Unfair to compare with compressed network bound and b) despite pre-activations, our bound performs better for large D, small H**
> >
> > Dear Reviewer,
> >
> > Thanks for the response. We understand your concern is about i) a lack of comparison with Arora et al., and ii) how big the numerical value of our bound can be in comparison with Arora et al., and about the larger explicit polynomial dependence on depth. We have two concrete points to address your concern and we hope it helps you appreciate our result better:
> >
> > 1. First, we would like to remind you, as we have stated at multiple points throughout our paper and in our earlier responses here, the bound of Arora et al., is NOT on the original network but on a compressed network (as has been noted by them in  Remark (1) under page 4 of their arxiv version https://arxiv.org/pdf/1802.05296.pdf)
> > While they introduce a lot of interesting noise-resilience properties and show how a network can be compressed using those properties, their final bound which is small, holds only for the compressed network. Extending the benefits of noise-resilience in the form of a generalization bound on the original network is another non-trivial part of the puzzle, which is what we accomplish. It would be quite unfair to compare our bound with their bound on a compressed network because -- as we have stated everywhere -- our goal is to be able to say something about the original network.
> >
> > The reason we care about a bound on the original network and not on the compressed network is that a bound on the compressed network could potentially tell us very, very little about the original network. For example,  one can provide a compression bound  by simply getting rid of all the parameters in the original network, and training a much smaller network from scratch on the given training dataset. Of course, a generalization bound on the smaller network will be small; but does it say anything at all about the original network?
> >
> >
> > 2. We have been careful in stating everywhere in our paper that the 5% bound and median bound are just hypothetical quantities.  Our main claim through the experiments is that our bound has better asymptotic behavior  w.r.t depth -- at least for networks of small width -- as is evident from the reported slope of our actual bound named "Ours" vs Neyshabur+ '17 and Bartlett+ '17 Figure 2 a).  In fact, we report the actual value of our bound vs these bounds for a really deep network and show that across multiple runs, the distribution of our bound concentrates over smaller quantities (Figure 2 b).  Essentially, we have identified a regime (large D, small H) where, DESPITE the dependence on pre-activation values (which needs to be improved) our bound -- not just the hypothetical variations -- does better than existing bounds on the original network in practice. We hope that the existence of this regime in practice helps you appreciate the usefulness of our approach of generalizing noise-resilience, and the promise it holds in terms of providing bounds on the original network.
> >
> > The reason our bound does better in this regime is that when H is small,  the pre-activations tend to be large enough, and more importantly, when D is large, the product of spectral norms is exponentially larger than our terms including the extra D^2 in our bound.
> >
> > We'd love to know if this addresses your concerns, or if you have further questions.

---

> > > ### Comment · AnonReviewer3 · 2018-11-25
> > > **Still not clear why the proposed bound is better than existing ones**
> > >
> > > Thanks for the authors’ update and clarification. I do agree that the result that state the bound in terms of the original network is important (unlike [1]), and the derived deterministic PAC-Bayesian type of generalization bound may be of independent interest. But since the major claim of this paper is about the generalization bound for neural nets rather than the deterministic PAC-Bayesian bound, I tend to judge from a view of the former instead of the latter. I do not think the derived generalization bound is tighter than existing ones (e.g., [1,2]) in the scenarios of interesting/practical settings.
> > >
> > > 1. The network with a small width is not an interesting setting in general. Both practice and recent theoretical efforts show that over-parameterization is more interesting in general, which can help both optimization and generalization.
> > >
> > > 2. The claim that the derived result has better performance in increasing depths is too vague to see from the experiment results (e.g., Fig 2). It is ok to have the 5% and median plots as a way to see how the bound performs in the non-worst-case scenarios, but it is not fair to compare with [1,2]. I think only looking at the general bound (e.g., blue line in Fig. 2) is a fair game. There is no significant trend that the derived bound increases slower for a larger value of depth compared with [1,2].
> > >
> > > On the other hand, if I understand it correctly, the numerical results are obtained when there are no explicit constraints on weight matrices. The product of norms is indeed an issue in this case. However, it has been shown that using unit spectral norm weight matrices has as good empirical performance as those without such constraints in real tasks [4,5] (they have orthogonal weights). In the latter case, the product of spectral norms is simply 1, where I believe the bounds for [2,3] can be significantly lower without sacrificing the performance. It is not clear how the pre-act value will differ then, but it seems it will still be significantly larger than 1. In addition, when we only compare the polynomial dependence of the bound on the depth and width, the derived bound has a universally worse dependence on depth and the dependence on width is better only when the pre-act values are large over the entire parameter space and data (which seems highly impossible in practice).
> > >
> > > [1] Arora et al. Stronger generalization bounds for deep nets via a compression approach, 2018.
> > > [2] Bartlett et al. Spectrally-normalized margin bounds for neural networks, 2017.
> > > [3] Neyshabur et al. A PAC-bayesian approach to spectrally-normalized margin bounds for neural networks, 2018.
> > > [4] Xie et al. All you need is beyond a good init: Exploring better solution for training extremely deep convolutional neural networks with orthonormality and modulation, 2017.
> > > [5] Huang et al. Orthogonal weight normalization: Solution to optimization over multiple dependent stiefel manifolds in deep neural networks, 2017.

---

> > > > ### Author Response · Authors · 2018-11-25
> > > > **Part 1/2: Factual concerns**
> > > >
> > > > Thank you for the detailed response.
> > > >
> > > > Below we first address the factual concerns you have..
> > > >
> > > > ===================
> > > > In your review you say "you do not think the derived generalization bound is tighter than existing ones (e.g., [1,2])", we suppose this is a typo and you mean [2,3]? We've compared our results only with [2,3]; as we said a comparison with [1] is extremely unfair.
> > > > =====================
> > > >
> > > > On your factual concerns about our plots:
> > > >
> > > >  While it may not be visually apparent, in Figure 2 (a), the maximum - minimum y value of the blue line is 11.66 - 8.9 = 2.28 while for the line corresponding to [3] is 7.58-3.75 = 3.83. (Note that the y value corresponds to the log of the bound). The amount by which our bound increases with depth is definitely smaller than the amount by which [2,3] increase; even a seemingly small difference in the rates of the increase results in an exponential difference of the actual bound. For these two lines (not the hypothetical versions!), the rates translate to 1.57^D vs 2.15^D specifically and we have mentioned this in the paper. Furthermore, Fig 2 (b) clearly demonstrates the tipping point where ours improves over [2,3]. We hope this clears up any question about the vagueness/validity of our claim that for large D and small H our bound does better.
> > > >
> > > > Next, the hypothetical versions of our bound are plotted for the sake of comparison with our own bound to demonstrate that the pre-activation values are indeed the limiting factor in our bound. In the discussion in the paper which begins "We also plot hypothetical variations of our bound..." we clearly state
> > > >
> > > >  ".... perform orders of magnitude better than our actual bound (note that these two hypothetical bounds do not actually hold good) ... This indicates that the only bottleneck in our bound comes from the dependence on the smallest pre-activation magnitudes, and if this particular dependence is addressed, our bound has the **potential** to achieve tighter guarantees for even smaller D such as D = 8."
> > > >
> > > > We have been careful and transparent in presenting these hypothetical variations and made sure not to draw any explicit comparisons with [2,3] here.
> > > >
> > > > In short, we have NOT made any unfair comparisons!
> > > >
> > > > ============
> > > >
> > > > The point about the effectiveness of constrained spectral norm sounds quite interesting! Thanks for sharing it.
> > > >
> > > > However, we *strongly disagree* that it makes our result seem any less interesting: the fact that such a constrained-spectral-norm scenario works in practice, does not void the theoretical question of
> > > > "What is a generalization bound on deep networks where the spectral norm each matrix has not been constrained to be 1 and typically lies around 2.1-3?".  The fact that our bound might show no improvements in your scenario does not invalidate whatever claim we make about (small H, large D, unconstrained spectral norm)
> > > >
> > > > We understand that it is a worthwhile exercise to compare the polynomial dependence on depth/width and we agree that our bound has worse polynomial dependence on depth if we ignore the spectral norm terms. But it is not clear to us, from a theoretical point of view, why one would choose to ignore the existence of an exponential depth factor, and any possible improvement over that factor at the cost of extra polynomial dependence.
> > > >
> > > > ===============

---

> > > > ### Author Response · Authors · 2018-11-25
> > > > **Part 2/2: Subjective concerns about our contributions and their significance**
> > > >
> > > > Deriving a generalization bound on the original network is important as bounds on modified networks have limited explanatory powder. That has been the main premise and motivation of this paper, and we are happy to learn about your agreement with us on this!
> > > >
> > > > We are also glad you effectively agree that a comparison with [1] is unfair.
> > > > ============
> > > > Over to the subjective points about your claim of the paper, we believe it is simplistic to state that "the major claim of this paper is about the generalization bound for neural nets rather than the deterministic PAC-Bayesian bound".
> > > >
> > > > The claim of the paper is two-fold: informally, "a) here is a new method to use train-time noise-resilience of the network to derive a bound on the original network by generalizing noise-resilience and b) here's one particular way of characterizing noise-resilience (in terms of jacobians and pre-activations) and generalizing it gives us spectral-norm independent bounds; additionally, here's a particular regime where our bound can do better despite dependence on pre-activations." The claim of the paper is not "here's a bound on the original network" (which would only be Theorem 4.1).
> > > >
> > > > While the dependence on the pre-activation is something you find bothering -- and we do agree that is very, very reasonable -- the limitation of the dependence on pre-activations in Thm 4.1 is a limitation in how we characterize noise-resilience and not in how we generalize noise-resilience.
> > > >
> > > > You might still ask "why is 'generalizing noise-resilience' interesting? Why should I care about it if at this point, I do not know if it can help me provide stronger bounds for "practically relevant" deep networks (i.e., large H, not so large D)?"
> > > >
> > > > First, while it is true that we do not have stronger bounds for (large H, small D) networks,  the theoretical question of "why do overparametrized networks generalize well?" applies even for the (small H, large D regime).
> > > >
> > > > Next, most of the really strong (both non-vacuous and vacuous) bounds that we know so far apply only on modified networks. A BIG gap in these bounds is essentially about how to carry over the benefits of these bounds to the original network. Unfortunately, it might not be obvious how "big"  a gap this is, because to the best of our knowledge, research so far has not explicitly focused on closing this gap. We believe that the pursuit of closing the gap and providing a bound on the original network is a highly significant and non-trivial pursuit as otherwise these existing papers would have achieved that.
> > > >
> > > > So far, it seems like one has had to somehow modify the network -- either by dropping/modifying many of its parameters [1], or by adding noise to reduce the dependence of the parameters on the training data, or by doing both! [2,3] -- thereby "cheating" the actual question at hand about the original network, only to provide a strong generalization bound on a modified network. Our paper fills this significant conceptual gap here by providing the idea & specific technique of generalizing noise-resilience (Thm 3.1) and further illustrating its promise by showing how it can extend the benefits of noise-resilience to the original network's bound in a specific case -- even if it may not be a practically popular case.
> > > >
> > > >
> > > > Effectively, we provide a novel conceptual answer to a big piece in the puzzle and clearly demonstrate its benefits in a specific regime -- we think this will be valuable to the community and therefore worth publishing. Furthermore, our conceptual answer is quite general [i.e., Thm 3.1 is a general framework] and might inspire researchers to think about ways in which the multitudes of existing bounds on modified networks can be extended to their original networks.
> > > >
> > > >
> > > >
> > > >
> > > >
> > > > [1] Arora et al., Stronger generalization bounds for deep nets via a compression approach
> > > > [2] Zhou et al., Compressibility and Generalization in Large-Scale Deep Learning
> > > > [3] Dziugate and Roy, Computing Nonvacuous Generalization Bounds for Deep (Stochastic) Neural Networks with Many More Parameters than Training Data

---

> > > > ### Author Response · Authors · 2018-11-29
> > > > **Response to Reviewer 3's comment from  29 Nov (that is missing here?)**
> > > >
> > > > We received an email notification from openreview with Reviewer 3's comment but we can't  find it here on the website. The following is the comment we received:
> > > >
> > > > =============================
> > > > Comment: Thanks for the authors feedbacks. It is great to discuss the problems.
> > > >
> > > > As I have discussed with the author, I do think that the deterministic PAC-Bayesian bound itself maybe of interest if one can apply it to derive a stonger gengeralization bound. If the authors can demonstrate such superiority of the deterministic PAC-Bayesian bound by another example, I will further appreciate this result.
> > > >
> > > > However, my concern is the current derived theoretical result is not ease to interpret and there are quantities that heavily depend on empirical values (that can be very large). The product of norms of may not be good, but it provides an explicit way to control the capacity of the networks so that we can have guaranteed bounds. Also as I mentioned earlier, empirical studies have already shown that by explicitly controlling the spectral norms of weight to be (nearly) 1, the performance of the network is not affected so that the product of the spectral norm is not an issue (i.e., close 1). I am not sure how the pre-activation will be in such scenarios, but it seems highly likely that the pre-activation is still large. Removing the product of norms and introducing some empirical quantities may not be always good, especially such quantities are very sensitive to data and can results in even worse bounds than the product of norms.
> > > >
> > > > In summary, I do repect the authors that they provide a different angle to view the problem. On the other than, I do think that what is needed for the generalization bound of neural nets is not a new result that can be vacuous and can not be guarantted to push the edge of better understanding/interpreting the bound. I have updated my score to reflect my such a concern.
> > > >
> > > > =============================
> > > > OUR RESPONSE
> > > >
> > > > We thank the reviewer for their response, for increasing their score and for appreciating our new perspectives on the problem.
> > > >
> > > >
> > > > >>>> Also as I mentioned earlier, empirical studies have already shown that by explicitly controlling the spectral norms of weight to be (nearly) 1, the performance of the network is not affected so that the product of the spectral norm is not an issue (i.e., close 1).
> > > >
> > > > We apologize for repeating ourselves a bit here, we are in disagreement with your point that the existence of spectral-norm-controlled networks makes our bounds and our claimed conceptual contributions & specific numerical improvements less interesting. If we understand your argument right, this argument is similar to saying that "all extremely large vacuous bounds on extremely overparamterized networks are less interesting because there are relatively smaller overparameterized networks that generalize almost as well and on which a VC dimension bound would be smaller than the large vacuous bounds on the larger networks." The fact that extremely overparameterized networks exist and generalize well demands theoretical explanation, and this question is independent of other networks that may be either smaller or whose norms maybe controlled explicitly.
> > > >
> > > > >>>> "The product of norms of may not be good, but it provides an explicit way to control the capacity of the networks so that we can have guaranteed bounds"
> > > >
> > > > It is not clear to us why the quantities in our bound "can't" be explicitly controlled. During training, one could potentially add regularizers that minimize the norm of the layers' outputs, the Jacobians norms of the layers, and maximize the pre-activation values.
> > > > Of course, this all maybe highly non-trivial and way beyond the scope of the paper, but we want to establish that our quantities are in no way different from the spectral norms of the matrices in terms of how and whether they "can be controlled" or not. For a better comparison, we believe the quantities in our bound are just as "controllable/optimizable" as the quantities in Arora et al.,
> > > >
> > > > But more importantly, even if it is the case that our quantities can somehow not be controlled, we believe that evaluating the quality of ageneralization bound in terms of "does it contain quantities that can be explicitly controlled?" is an orthogonal goal to the theoretical question of "what properties of deep networks -- trained with SGD, without any explicit regularization/norm control -- will help us understand why they generalize well?"
> > > >
> > > > >>>> "If the authors can demonstrate such superiority of the deterministic PAC-Bayesian bound by another example, I will further appreciate this result. "
> > > >
> > > > We understand and appreciate your request. We'd love to think about this to improve future versions of this paper. But we're afraid there's not much time left in the rebuttal period for us to provide a concrete answer to this, nor do we think we have the option to update the paper at this point.

---

> ### Author Response · Authors · 2018-11-27
> **Summary of discussions with Reviewer 3**
>
> Over the course of this discussion we've done our best to address the different concerns raised by Reviewer 3. We think it'll be useful to have a quick summary of these. We thank them for their response so far and hope to continue the conversation until the rebuttal deadline so that as many of their concerns are addressed as possible.
>
> =========
> Summary of their Nov 2 comment and our Nov 8 response
> =========
>
> Concern: Our approach does not tighten the error bound from a more refined/structured way
> Our response: Our general PAC-Bayesian approach to generalizing noise-resilience involves carefully and iteratively generalizing a sequence of conditions without incurring a product-of-spectral norm term.
>
> Concern:  It is not clear where the noise resilience shows up from the analysis or the result/the title and the way the authors explain as noise resilience is somewhat misleading. More detailed explanation will help.
> Our response: We provided a detailed explanation of our contribution which we believe was misunderstood, and how it is about generalizing noise-resilience from training data to unseen data
>
> Concern:  The analysis seems to be standard as in the PAC-Bayesian analysis,
> Our response: Our PAC-Bayesian analysis is novel, non-trivial and far from standard analyses which do not generalize noise-resilience.
>
> Concern:  Lack of a comparison of our bound and existing ones to see the quantitative difference of the results.
> Our response: Added plots on 19 Nov.
>
> The reviewer has acknowledged in their 22 Nov response that our PAC-Bayes result about generalizing noise-resilience (Theorem 3.1) "might be of independent interest here." and in their 29 Nov response that "I do repect the authors that they provide a different angle to view the problem.".
>
> ======
> Summary of Nov 22 comment-response
> =======
> Concern: Lack of comparison with Arora et al., '18
> Our response: Unfair to compare with bound on the compressed network from Arora et al., Bound on a compressed network does not have full explanatory power. Important to study how one can extend the benefits of noise-resilience enjoyed by bounds on stochastic/compressed network to the original network.
>
> In their Nov 24th comment, the reviewer agreed that a bound on the original network is important.
>
> Concern: Using the 5% and median pre-act values are not fair comparisons with other bounds.
> Our response: We have been careful and transparent in presenting these hypothetical variations and we never compared them with the older bounds.
>
>
> ======
> Summary of Nov 24 comment-Nov 25 response
> =======
> Concern: The regime where we claim improvement is not practically relevant.
> Our response: The question of why (Large D, small H)-networks can generalize well is still a question that needs to be answered and the question holds theoretical value.
>
> Concern: The claim about improvements in the said regime is vague from the plots.
> Our response: We reported the exact numerical increase in the plots to justify our claim, and referred the Reviewer to Fig 2 (b)
>
> Concern: Training the network by explicitly controlling spectral norms to be 1 works pretty well. Our bound when applied on these networks won't show any improvement.
> Our response: The question of why networks with uncontrolled spectral norms can generalize well is still a question that needs to be answered and the question holds theoretical value.
>
>
> Concern: Explicit polynomial dependence on depth is worse.
> Our response: We agree but why should one ignore the exponential dependence on depth or any improvements on it?
>
> Concern: The major claim of this paper is about the generalization bound for neural nets rather than the deterministic PAC-Bayesian bound.
> Our response: Our claim is two-fold, with the general framework of generalizing noise-resilience one half of it. We argued why generalizing noise-resilience is interesting, and an important, highly non-trivial contribution to understanding generalization in deep learning.
>
> =====
> Summary of Nov 29 comment-Nov 29 response
> =====
> Concern: The terms in our bound, unlike the product of spectral norms, do not provide an explicit way to control the capacity of the networks so that we can have guaranteed bounds.
> Our response: It is not clear to us why the quantities in our bound "can't" be explicitly controlled, or why it would be harder to do so when compared to the equally nuanced terms present in bounds like in Arora et al.,  Even if we can't control them, the metric of "does the bound contain quantities that can be explicitly controlled?" is orthogonal to the metric of "does this bound help explain deep network generalization in some way?"
>
> Concern: Demonstrate the superiority of  the deterministic PAC-Bayesian bound by another example
> Our response:  This will certainly help improve future versions of this paper and we'll work on it. But we don't have the option of updating the paper, or much time left in the rebuttal period to think about this to provide a concrete answer.

---

### Official Review · AnonReviewer2 · 2018-11-03
**An honest work**

**Rating:** 7
**Confidence:** 3

**Review:**

The fact that a number of current generalization bounds for (deep) neural networks are not expressed on the deterministic predictor at stake is arguably an issue. This is notably the case of many recent PAC-Bayesian studies of neural networks stochastic surrogates (typically, a Gaussian noise is applied to the network weight parameters). The paper proposes to make these PAC-Bayesian bounds deterministic by studying their "noise-resilience" properties. The proposed generalization result bounds the margin of a (ReLU) neural network classifier from the empirical margin and a complexity term relying on conditions on the values of each layer (e.g., via layer Jacobian norm, the layer output norm, and the smallest pre-activation value).

I have difficulty to attest if the proposed conditions are sound. Namely, the authors genuinely admit that the empirically observed pre-activation values are not large enough to make the bound informative (I must say that I truly appreciate the authors' candor when it comes to analyzing their result). That being said, the fact that the bounds does not scale with the spectral norm of the weight matrices, like previous PAC-Bayesian result for neural networks, is an asset of the current analysis.

I must say that I had only a quick look to it the proofs, all of them being in the supplementary material along most of the technical details. Nevertheless, it appears to me as an honest, original and rigorous theoretical study, and I think it deserves to be presented to the community. It can bring interesting discussion and suggest new paths to explore to explain the generalization properties of neural networks.

Minor comment: For the reader benefit, Theorem F.1 in page 7 should quickly recall the meaning of some notation, even if it's the "short version" of the theorem statement.

====
update: The bound comparison added value to the paper. It strengthens my opinion that this work deserves to be published. I therefore increase my score to 7.

---

> ### Author Response · Authors · 2018-11-08
> **Yes, it is important to derive bounds on the original network!**
>
> Thank you for your positive response! We are glad you agree that many of the current generalization bounds for deep networks apply only to a compressed/stochastic network; indeed,  even though these bounds provide valuable intuition about generalization, we believe that an extremely important and non-trivial piece of the puzzle is to extend the benefits of these bounds (or at least some of its benefits -- in this case the lack of a product-of-spectral-norm dependence) over to the original network. And we achieve this through an approach that "generalizes noise resilience".
>
> With regards to your suspicion about the proposed "conditions", the only pesky condition in our result is the one involving the pre-activation values. The other bounds on the other quantities certainly hold favorably in practice as seen in our plots. We must also note that these conditions themselves are not the main contribution of our paper (and we have stated this point in "Our Contribution" in Page 3); the main contribution lies in how we generalize these conditions assumed about the network on the training data, to test data (without ever incurring a product-of-spectral-norms dependence). The conditions themselves are in fact philosophically similar to conditions examined and verified in prior work [1,2]; in essence, they dictate how the parts of the weight matrices activated by a particular datapoint, interact with each other.
>
> Even as far as the condition involving the pre-activation values are concerned, it appears in our analysis to ensure that the hidden units don't jump their non-linearity under parameter perturbations; the assumption that only a small proportion of the hidden units  do not jump the non-linearity under perturbations has been made in prior works, although in a more relaxed form e.g., "Interlayer Smoothness" in [1] or condition C2 in [2], and *these have been verified in practice*. Intuitively, we believe that this assumption allows one to argue that the network is "linear" in a small local neighborhood in the parameter space, and this local linearity helps imply that the network has lesser complexity.
>
> Again, we thank the reviewer for appreciating our contributions. We hope that the community finds our approach of generalizing noise-resilience useful. Our framework is general in that one could think of designing different sets of conditions that imply noise-resilience of the network, and argue how these conditions would generalize; with a better understanding of the source of noise-resilience in deep networks, we might identify better sets of conditions which can be generalized this way to obtain tighter bounds on the original network.
>
> We will take note of the reviewer's comment about Theorem F.1!
>
> [1] Arora et al., "Stronger generalization bounds for deep nets via a compression approach."
> [2] Neyshabur et al., "Exploring gen- eralization in deep learning."

---

> ### Author Response · Authors · 2018-11-15
> **Revision uploaded**
>
> As you suggested, we have recalled some of the notation in  the text preceding Theorem F.1 (which by the way is now Theorem 4.1 as it should be, thanks to Reviewer 4).
> Thanks for your suggestion!

---

> ### Author Response · Authors · 2018-11-24
> **Thank you**
>
> Thank you for increasing your score and for taking into consideration our discussion with Reviewer 4! We thank Reviewer 4 too for their constructive feedback and active interest in helping us improve the quality of the paper.

---

### Official Review · AnonReviewer1 · 2018-11-04

**Rating:** 7
**Confidence:** 2

**Review:**

This paper provides new generalization bounds for deep neural networks using the PAC-Bayesian framework. Recent efforts along these lines have proved bounds that
either apply to a classifier drawn from a distribution or to a compressed form of the trained classifier. In contrast, the paper uses PAC Bayesian bounds to
provide generalization bounds for the original trained network. At this same time, the goal is to provide bounds that do not scale exponentially in the depth of the
network and depend on more nuanced parameters such as the noise-stability of the network. In order to do that the paper formalizes properties that a classifier must
satisfy on the training data. While these are a little difficult to understand in general, in the context of ReLU networks these boil down to bounding the l2-norms
of the Jacobian and the hidden layer outputs on each data point. Additionally, the paper also requires the pre-activations to be sufficiently large, which as the authors
acknowledge, is an unrealistic assumption that is not true in practice. Despite that, the paper makes an important contribution towards our current understanding of
generalization of deep nets. It would have been helpful if the authors had a more detailed discussion on how their assumptions relate to the specific assumptions in the papers
of Arora et al. and Neyshabur et al. This would help when comparing the results of the paper with existing ones.

---

> ### Author Response · Authors · 2018-11-15
> **Thank you! Added detailed discussion on the conditions from prior work**
>
> Dear Reviewer,
>
> Thanks for your positive feedback!
>
> We have uploaded a revised version with Appendix G where we have added a one-page discussion relating our noise-resilience conditions and the conditions in prior work. We hope this provides you better context to understand our assumptions. Happy to provide more details if needed.

---

### Official Review · AnonReviewer4 · 2018-11-12
**Interesting paper - can be improved significantly**

**Rating:** 8
**Confidence:** 5

**Review:**

This paper presents a PAC-Bayesian framework that bounds the generalization error of the learned model. While PAC-Bayesian bounds have been studied before, the focus of this paper is to study how different conditions in the network (e.g. behavior of activations) generalize from training set to the distribution. This is important since prior work have not been able to handle this issue properly and as a consequence, previous bounds are either on the networks with perturbed weights or with unrealistic assumptions on the behavior of the network for any input in the domain.

I think the paper could have been written more clearly. I had a hard time following the arguments in the paper. For example, I had to start reading from the Appendix to understand what is going on and found the appendix more helpful than the main text. Moreover, the constraints should be discussed more clearly and verified through experiments.

I see Constraint 2 as a major shortcoming of the paper. The promise of the paper was to avoid making assumptions on the input domain (one of the drawbacks in Neyshabur et al 2018) but the constraint 2 is on any input in the domain. In my view, this makes the result less interesting.

Finally, as authors mention themselves, I think conditions in Theorem F.1 (the label should be 4.1 since it is in Section 4) could be improved with more work. More specifically, it seems that the condition on the pre-activation value can be improved by rebalancing using the positive homogeneity of ReLU activations.

Overall, while I find the motivation and the approach interesting, I think this is not a complete piece of work and it can be improved significantly.

===========
Update: Authors have addressed my main concern, improved the presentation and added extra experiments that improve the quality of the paper.  I recommend accepting this paper.

---

> ### Author Response · Authors · 2018-11-12
> **Constraint 2 is NOT a shortcoming, and provably holds!**
>
> Dear Reviewer, thanks for your precise summary of the paper's approach and your thoughts about it!
>
> We strongly disagree with your remark that Constraint 2 is  "a major shortcoming of the paper". Here's why:
>
> Constraint 2 is not restrictive and is in fact a very natural/intuitive constraint of the properties in the network -- and it provably holds good. At a high level, all the constraint says is the following:
>
> **For a given point x** for which the first r-1 sets of properties are bounded (say the first 3 layers have small l2 norm), the r-th property is noise-resilient (i.e., under noise injected into the parameters, the 4th layer's l2 norm does not suffer much change under parameter perturbation).
>
> This is a pretty natural constraint **which provably holds** for networks because of how the output of a particular layer depends only on the output of the preceeding layers.
>
> We make NO assumption of the form that something about the network holds good for ALL inputs in the domain. As you can see in Theorem 3.1, we say "if W satisfies T_r(W, x, y) > Delta_r^* ... for all (x,y) in S" which means that these properties are bounded only for the training data.
>
> We hope this clears the misunderstanding surrounding the constraint and convinces you that this is not at a drawback at all!
>
> The drawback that we acknowledge is regarding the dependence on the pre-activations, which we hope to improve upon in the future. But as it is, we believe the paper makes a conceptual contribution in terms of a new methodology of generalizing noise-resilience, and accomplishes a PAC-Bayes based product-of-spectral-norm independent bound in specific settings where it wasn't possible.
>
> As you've suggested, we will improve the discussion of the constraints; thanks for your comment!

---

> ### Author Response · Authors · 2018-11-15
> **Uploaded revision**
>
> Hi again!
>
> First of all, a quick note: we updated the label of Theorem F.1 to 4.1. Thanks for your note!
>
> Next, we'd like to get in touch with you again to know if we clarified your concern regarding Constraint 2. (By the way, please let us know in case we misunderstood your concern.)
>
> We'd like to reiterate, like we state throughout the text of the main paper, we do not make any assumption that holds on all input datapoints. The lack of such an assumption is the main strength/contribution of the paper.  We'd also like to point out that the mathematical statement of Constraint 2 and the text following it, and the mathematical statements of Theorem 3.1 and 4.1, all reflect this fact!
>
> In the light of this discussion, we respectfully encourage you to reevaluate the paper & update your score. Thank you!

---

> > ### Comment · AnonReviewer4 · 2018-11-16
> > **Thanks for clarification + some feedback**
> >
> > Thanks a lot for clarifying constraint 2. I think my confusion was because you have not mentioned the constraints in the Theorem 3.1 statement but used it in the proof of the theorem (and of course because I did not read the proof of Theorem 4.1 carefully). I have spent more time reading your paper and here is some feedback:
> >
> > 1- I find Theorem 3.1 interesting and useful. First of all, please clearly mention the assumptions in the statement of  theorem 3.1, i.e.  constraint 1 and 2.
> >
> > 2- There is too much notation in the paper. I understand that there is no easy way to figure out how to reduce the notation but this complexity hides the result of the paper and not many readers are willing to spend hours figuring out the notation. I suggest to put the neural net notation after the Theorem 3. With very simple notation, you should be able to write the assumptions and Theorem 3. I think this is the most interesting part of the paper and it worth spending time to present it properly.
> >
> > 3- I believe Theorem 4.1 is needed to demonstrate how Theorem 3.1 can be useful but the limitations of Theorem 4.1 (which are not related to Theorem 3.1) should be discussed clearly. You already mentioned the main limitation which is the dependence of the bounds on the inverse of smallest pre-activation. I have two suggestions:
> > a) Even though it is mentioned indirectly in the discussion, I think you should clearly mention early in the discussion that this limitation is due to the fact that the proof does not allow activations to flip. This helps the reader to have a better understanding of this limitation and potentially build on your work.
> >
> > b) Most plots show the quantities vs depth. Please fix the number of layers and plot the quantities vs "#hidden units per layer" as well (up to at least 2K hidden units per layer). Please also report the numerical value of the generalization bound on a network with 1K hidden units and 10 layers. If you have time, compare it to at least one of the other generalization bounds. To be clear, I am not going to evaluate your generalization bound based on these plots but what matters is that these plots help the reader to have a clearer picture.
> >
> > I am looking forward to the revision and then I will decide about the final score (up to 8 if all the suggestions are applied).

---

> > > ### Author Response · Authors · 2018-11-16
> > > **Thank you for the suggestions!**
> > >
> > > Dear Reviewer,
> > >
> > > Thanks for considering our clarification and accepting it. Also, thanks for studying the paper more carefully and providing concrete, valuable feedback. We will work on them!
> > >
> > > Currently, there are plots for dependence on width, upto 1280 hidden units, present in Figure 3 and 4. We will present more plots as soon as possible.

---

> > > ### Author Response · Authors · 2018-11-17
> > > **Incorporated suggestions 1,2 and 3(a)**
> > >
> > > Hi! We wanted to let you know that we've uploaded a revision with suggestions 1 2 and 3(a) incorporated. We are still working on 3b.
> > >
> > > 1. We're glad you find the theorem interesting. Indeed, we believe that the generality and the novelty in this theorem leaves a lot of opportunity for exploration by the both the deep learning theory community and the learning theory community.
> > >
> > > 2. We moved the network-related notations to Section 4. In Section 3, we completely rephrased the description of "INPUT-DEPENDENT PROPERTIES OF WEIGHTS"  and the description following Constraint 2, without using neural network notations. We also modified it to read better. We hope that the rewritten version of this discussion, and the additional text we've squeezed into Theorem 3.1 can help parse the notation more easily. However, we think it's hard to get rid of the other notations involving T, r, \rho etc., which are integral to describing the abstract setup.  Having said that, we are happy to consider further suggestions here! We really appreciate your above suggestions in this context and believe it helps reduce the burden on the reader.
> > >
> > > 3 (a) Again, this is a good point and we have incorporated it as follows:
> > > In the last paragraph of "Our Contributions" we say:
> > > "Intuitively, we make this assumption to ensure that under sufficiently small parameter perturbations, the activation states of the units are guaranteed not to flip."
> > > and again after Thm 4.1, we modified the paragraph at the end of page 7, and added the line:
> > > "Specifically, using the assumed lower bound on the pre-activation magnitudes we can ensure that, under noise, the activation states of the units do not flip; then the noise propagates through the network in a tractable, “linear” manner. Improving this analysis is an important direction for future work."

---

> > > ### Author Response · Authors · 2018-11-19
> > > **Incorporated all suggestions + demonstrated that our bound works better for sufficiently large D, small H**
> > >
> > > Hi again!
> > >
> > > We want to let you know that we've incorporated all your suggestions and presented some additional experiments too.
> > >
> > > Specifically, in the main paper, we have demonstrated the value of our bound for H=40, and varying depth and compared with spectral-norm-bounds Neyshabur et al., '18 and Bartlett et al., 17. We argue that for this H, our bound should perform asymptotically better and show that our bound does better for D=25.
> > >
> > > Due to space constraints, we had to present some of the plots in the appendix.
> > >
> > > >>>>>>>>> Please fix the number of layers and plot the quantities vs "#hidden units per layer" as well (up to at least 2K hidden units per layer).
> > >
> > > The plots in Appendix Figure 5 show the quantities and the overall bound (including existing bounds) for H=40 until 2000, for depth D=8.
> > > Additionally, Figure 6 shows a similar plot for depth D=14, for H=40 until 1280.
> > >
> > >
> > > >>>>>>>> Please also report the numerical value of the generalization bound on a network with 1K hidden units and 10 layers.
> > >
> > > You can find the plots in Appendix Figure 4 for no. of units H=1280, where we show both the individual quantities and the actual bound for different depths uptil D=14.
> > >
> > >
> > > >>>>>>> If you have time, compare it to at least one of the other generalization bounds.
> > > Compared our bounds with both Neyshabur et al., '18 and Bartlett et al., 17 which have pretty similar orders of magnitudes with each other. Please refer to Figure 2 in the main paper.
> > >
> > > We are eager to hear back from you if you have any feedback or further questions, and would love to know your updated review.

---

### Author Response · Authors · 2018-11-19
**Added plots of the bounds -- our bound works better for large D, small H**

We added experiments in the paper demonstrating the actual values of the bound in comparison with existing product-of-spectral-norm based bounds. We want to emphasize that our bound shows weaker dependence on depth, and performs asymptotically better with depth. Specifically, we show an improvement over two popular, existing bounds for $D=28$ and $H=40$. We argue that for larger depth, our bound promises greater improvements over product-of-spectral-norm-based bounds.

Note: The paper was originally within 8 pages, but is now 8.5 pages because of the additional plots & their accompanying discussion.

---

> ### Comment · AnonReviewer4 · 2018-11-19
> **Thanks for adding the plot - some suggestions**
>
> Thanks for adding the plot. I think it is very helpful and improves the quality of the paper. I understand that revisions take time and energy but I think there are two issues with the current Figure 2:
>
> More important:
>
> I agree with your conclusion that for sufficiently large D, your bound becomes lower than others. However, I find table (b) in Figure 2 a bit misleading. The main reason is that your bound is very sensitive to the value of pre-activations and hence if you train the same model with different random seeds, your bound gives very different values on each of the trained model. As a result, one cannot rely on reporting a single number here. Another thing that is a bit mysterious is that the slopes in figure (a) suggest that other bounds should be around 10^11 at depth 26 if they increase with the same rate but then their value is around 10^14 in table (b). So what happens between depth 13 and depth 26?
>
> I can think of three solutions here: 1) remove the table 2) report the average of 10 runs in table (b) 3) remove the table but extend the plot (a) to depth 30.
>
>
> Less important:
>
> I requested evaluating the bound for a network with 1K hidden units in each layer because that is the number which is typically used in practice. I still believe 40 hidden units is too low and it would be better to have at least 256 hidden units but this is not very important and I'm not going to insist on this.

---

> > ### Author Response · Authors · 2018-11-19
> > **We will update Table b**
> >
> > Thanks for engaging in a discussion with us and for providing prompt responses -- we really appreciate it!
> >
> > We are glad you agree with the asymptotic benefits of our bound.
> >
> > Your concern about Table b is understandable. The change in values is likely due to the fact that we used different training hyperparameters for D=26 (we will be sure to highlight the difference in the main text in the next revision, if the table persists). Training the networks beyond D=12 or 13 using vanilla SGD was tricky, and we realized we had to experiment with larger depths to convince the readers of the asymptotic benefits, so we had to pick a different D and resort to tuning the hyperparameters differently.
> >
> > We appreciate your different suggestions about Table b and we will work on it.
> >
> > As for H=1000, as we said we show plots for H=1280 in Figure 4 including the individual terms in the bound and the overall bound.   The goal of the experiments in the main paper was to identify and showcase the specific regime where we can hope the pre-activation values to not spoil the benefits of generalizing noise resilience. Improving the dependence on the pre-activation is crucial to achieve reasonable bounds for larger widths.

---

> > ### Author Response · Authors · 2018-11-20
> > **Updated Figure 2 (b)**
> >
> > Hi! We replaced the table reporting a single value with a distribution of values from 12 different runs  instead of just reporting averages which we think can be misleading here. Note that we have done this for D=28 instead of 26 as before. We hope we have addressed your above concerns through our previous response below and with the updated figure!

---

> > > ### Comment · AnonReviewer4 · 2018-11-20
> > > **Thanks**
> > >
> > > Thanks for updating the figure. At this point, all my concerns are addressed properly and hence I updated the score.

---

> > > > ### Author Response · Authors · 2018-11-21
> > > > **Thank you!**
> > > >
> > > > Thank you for your quick responses, for your useful suggestions, and  for updating your score!

---

### Author Response · Authors · 2019-02-12
**Updates in Camera Ready**

- In order to make the discussion of our framework in Section 3 simpler, we got rid of some notation.   The earlier version of our paper had some functions "T_1, T_2, ... T_r" and a constraint on them, which are no longer present.

- We have also improved our ReLU network bound by a factor of depth, D. In the earlier version, we had generalized O(D^2) noise-resilience related conditions from training data to test data, but now we generalize only O(D) such conditions, which helps us save a factor of D. Note that this update does not affect the structure of the proofs of generalization because we have abstracted most of it in Theorem 3.1. The only major change is in how we instantiate our framework for ReLU networks:
we add some extra conditions on the spectral norms of the Jacobians, and in "Main Lemma" Lemma E.1 we additionally analyze how these spectral norms respond to parameter perturbations.

---

### Meta-Review · Area_Chair1 · 2018-12-12
**ICLR 2019 decision**

**Confidence:** 4
**Recommendation:** Accept (Poster)

**Metareview:**

Existing PAC Bayes analysis gives generalization bounds for stochastic networks/classifiers. This paper develops a new approach to obtain generalization bounds for the original network, by generalizing noise resilience property from training data to test data.  All reviewers agree that the techniques  developed in the paper (namely Theorem 3.1) are novel and interesting.  There was disagreement between reviewers on the usefulness of the new generalization bound (Theorem 4.1) shown in this paper using the above techniques. I believe authors have sufficiently addressed these concerns in their response and updated draft. Hence, despite the concerns of R3 on limitations of this bound and its dependence on pre-activation values, I agree with R2 and R4 that the techniques developed in the paper are of interest to the community and deserve publication. I suggest authors to keep comments of R3 in mind while preparing the final version.